# The psychological, computational, and neural foundations of indebtedness

Xiaoxue Gao [1,2] ✉, Eshin Jolly [3], Hongbo Yu [4], Huiying Liu[5], Xiaolin Zhou [1,2,6] ✉ & Luke J. Chang [3] ✉

Receiving a favor from another person may induce a negative feeling of indebtedness for the beneficiary. In this study, we explore these hidden costs by developing and validating a conceptual model of indebtedness across three studies that combine a large-scale online questionnaire, an interpersonal game, computational modeling, and neuroimaging. Our model captures how individuals perceive the altruistic and strategic intentions of the benefactor. These inferences produce distinct feelings of guilt and obligation that together comprise indebtedness and motivate reciprocity. Perceived altruistic intentions convey care and communal concern and are associated with activity in insula, ventromedial prefrontal cortex and dorsolateral prefrontal cortex, while inferred strategic intentions convey expectations of future reciprocity and are associated with activation in temporal parietal junction and dorsomedial prefrontal cortex. We further develop a neural utility model of indebtedness using multivariate patterns of brain activity that captures the tradeoff between these feelings and reliably predicts reciprocity behavior.

Giving gifts and exchanging favors are ubiquitous behaviors that provide a concrete expression of a relationship between individuals or groups[1,2]. Altruistic favors convey concern for a partner's well-being and signal a communal relationship such as a friendship, romance, or familial tie[3–5]. These altruistic favors are widely known to foster the beneficiary's positive feeling of gratitude, which can motivate reciprocity behaviors that reinforce the communal relationship[6–9]. Yet in daily life, favors and gifts can also be strategic and imply an expectation of reciprocal exchanges, particularly in more transactive relationships[2,4,5,10–12]. Accepting these favors can have a hidden cost, in which the beneficiary may feel indebted to the favor-doer and motivated to reciprocate the favor at some future point in time[13–21]. These types of behaviors are widespread and can be found in most domains of social interaction. For example, a physician may preferentially prescribe medications from a pharmaceutical company that treated them to an expensive meal[22,23], or a politician might vote favorably on policies that benefit an organization, which provided generous campaign contributions[24]. However, very little is known about the psychological, computational and neural mechanisms underlying this hidden cost of indebtedness and how it ultimately impacts the beneficiary.

Immediately upon receipt of an unsolicited gift or favor, the beneficiary is likely to engage in a mentalizing process to infer the benefactor's intentions[25–27]. Does this person care about me? Or do they expect something in return? According to appraisal theory[28–33], these types of cognitive evaluations can evoke different types of feelings, which will ultimately impact how the beneficiary responds. Psychological Game Theory (PGT)[34–36] provides tools for modeling these higher order beliefs about intentions, expectations, and fairness in the context of reciprocity decisions[26,27,37,38]. Actions that are inferred to be motivated by altruistic intentions are more likely to be rewarded,

[1]Shanghai Key Laboratory of Mental Health and Psychological Crisis Intervention, School of Psychology and Cognitive Science, East China Normal University, Shanghai 200062, China. [2]School of Psychological and Cognitive Sciences, Peking University, Beijing 100871, China. [3]Department of Psychological and Brain Sciences, Dartmouth College, Hanover, NH 03755, USA. [4]Department of Psychological and Brain Sciences, University of California Santa Barbara, Santa Barbara, CA 93106-9660, USA. [5]Mental Health Education Center, Zhengzhou University, Zhengzhou 450001 Henan, China. [6]PKU-IDG/McGovern Institute for Brain Research, Peking University, Beijing 100871, China. ✉e-mail: xxgao@psy.ecnu.edu.cn; gxx114455@gmail.com; xz104@psy.ecnu.edu.cn; luke.j.chang@dartmouth.edu

while those thought to be motivated by strategic or self-interested intentions are more likely to be punished[26,27,37,38]. These intention inferences can produce different emotions in the beneficiary[39]. For example, if the benefactor's actions are perceived to be altruistic, the beneficiary may feel gratitude for receiving help, but this could also be accompanied by the feeling of guilt for personally burdening the benefactor[40–43]. Both feelings motivate reciprocity out of concern for the benefactor, which we refer to as "communal concern" throughout the paper[44,45]. In contrast, if the benefactor's intentions are perceived to be strategic or even duplicitous, then the beneficiary is more likely to feel a sense of obligation[13,14,21,46,47]. Obligation can also motivate the beneficiary to reciprocate[13,14,21,46,47], but unlike communal concern, it arises from external pressures, such as social expectations and reputational costs[48,49] and has been linked to feelings of pressure, burden, anxiety, and resentment[49–51]. Indebtedness has often been considered a unitary construct, defined singularly as either the feeling of guilt for personally burdening the benefactor[40–43] or as a sense of obligation to repay[13,14,21,46,47]. However, in everyday life, inferences about a benefactor's intentions are often mixed and we argue that indebtedness is a superordinate emotion that includes feelings of guilt for burdening the benefactor and social obligation to repay the favor.

In this work, we propose a conceptual model to capture how feelings of indebtedness arise and influence reciprocal behaviors (Fig. 1). Specifically, we posit that there are two distinct components of indebtedness–guilt and the sense of obligation, which are derived from appraisals about the benefactor's altruistic and strategic intentions respectively. The guilt component of indebtedness, along with

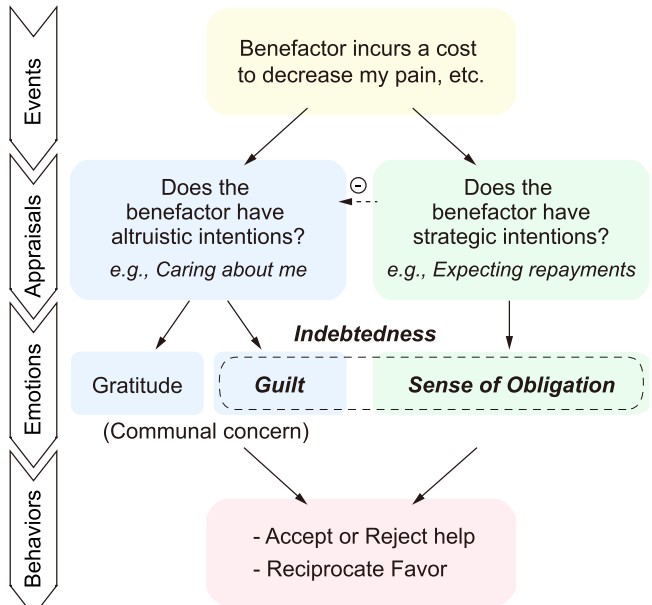

**Fig. 1 | Conceptual model of indebtedness.** We propose that there are two distinct components of indebtedness, guilt and the sense of obligation, which are derived from appraisals about the benefactor's altruistic and strategic intentions and can differentially impact the beneficiary's reciprocity behaviors. Following an event in which a benefactor provides help to the beneficiary (Yellow), the beneficiary is likely to appraise the benefactor's intentions. The higher the perception of the benefactor's strategic intention, the lower the perception of the benefactor's altruistic intention. The guilt component of indebtedness, along with gratitude, arises from appraisals of the benefactor's altruistic intentions (i.e., perceived care from the help) and reflects communal concern (Blue). In contrast, the obligation component of indebtedness results from appraisals of the benefactor's strategic intentions (e.g., second-order belief of the benefactor's expectation for repayment; Green). Both feelings of communal concern and obligation motivate the beneficiary's reciprocal behaviors (e.g., accept or reject the help and reciprocity after receiving help; Pink).

gratitude, arises from appraisals of the benefactor's altruistic intentions (i.e., perceived care from the help) and reflects communal concern. In contrast, the obligation component of indebtedness results from appraisals of the benefactor's strategic intentions (e.g., second-order belief of the benefactor's expectation for repayment). Both feelings of communal concern and obligation motivate the beneficiary's reciprocal behaviors. We find support for this conceptual model in Study 1, in which participants describe memories of past emotional experiences in a large-scale online questionnaire, using regression analysis and topic modeling.

In Study 2, we move beyond self-report and focus specifically on how the guilt and obligation components of indebtedness arise and influence behaviors in the context of an interpersonal game. In this study, participants receive electrical shocks and anonymous benefactors (co-players) can choose to provide aid to the participants by spending money to reduce the duration of their pain experience. The participants, in turn, have the opportunity to accept or reject this help and also to reciprocate the benefactor's help by sharing some of their own money back. We experimentally manipulate the participants' beliefs about the benefactors' intentions by providing information about whether or not the co-players are aware that the participants have the opportunity to repay after receiving help. We find evidence supporting the hypothesis that appraisals of altruistic intentions produce guilt as well as gratitude (i.e., communal concern) while appraisals of strategic intentions lead to obligation. Building on previous models of other-regarding preferences[37,38,52], we develop computational models to predict reciprocity and help-acceptance decisions by quantifying the tradeoff between the latent motivations of self-interest, communal concern (consisting of guilt & gratitude), and obligation based on appraisals induced by the interpersonal task (Eq. 1).

In Study 3, we provide further validations of the conceptual model by examining the brain processes associated with the two components of indebtedness by scanning an additional cohort of participants playing the interpersonal game using functional magnetic resonance imaging (fMRI). We find that the processes of communal concern (i.e., guilt & gratitude) are associated with activity in the insula, ventromedial prefrontal cortex (vmPFC), and dorsolateral prefrontal cortex (dlPFC), while the processing of obligation is more associated with activity in the temporal parietal junction (TPJ) and dorsomedial prefrontal cortex (dmPFC). Finally, we construct a neural utility model of indebtedness by applying our computational model directly to multivariate brain patterns to demonstrate that neural signals reflect the tradeoff between these feelings and can predict participants' trial-to-trial reciprocity behavior.

## Results
### Indebtedness is a mixed feeling comprised of guilt and obligation
In Study 1, we explore support for our conceptual model in self-reported experiences of Chinese participants collected via an online questionnaire. First, participants ($n = 1619$) described specific events, in which they either accepted or rejected help from another individual and rated their subjective experiences of these events. A regression analysis revealed that both self-reported guilt and obligation ratings independently and significantly contributed to increased indebtedness ratings ($\beta_{\text{guilt}} = 0.70 \pm 0.02$, 95%CI = [0.66, 0.73], $t(1988) = 40.08$, $p < 0.001$, $\beta_{\text{obligation}} = 0.40 \pm 0.02$, 95%CI = [0.36, 0.44], $t(1988) = 2.31$, $p = 0.021$, linear regression, two-tailed, FDR corrected; Fig. 2a-I; Table S1). Second, participants were asked to attribute sources of indebtedness in their daily lives. While 91.9% participants stated that their feelings of indebtedness arose from feeling guilt for burdening the benefactor, 39.2% of participants reported feeling obligation based on the perceived ulterior motives of the benefactor (Fig. 2a-II; Fig. S1a). Third, participants were asked to describe their own personal definitions of indebtedness. We applied Latent Dirichlet Allocation (LDA)

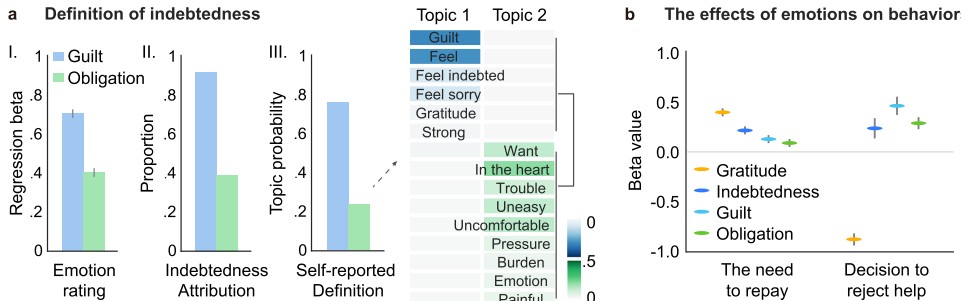

**Fig. 2 | Subjective experiences of indebtedness in Study 1. a** Contributions of guilt and obligation to indebtedness in Study 1 ($n = 1619$ participants). (I) Regression coefficients indicating the independent contributions of guilt and obligation ratings in predicting indebtedness ratings from recalling an event. Error bars reflect +/− SE of the estimate from the regression. (II) Proportion of participants attributing guilt and obligation as source of indebtedness. (III) Topic probabilities obtained from topic modeling of the emotional words in self-reported definition of indebtedness. Emotional words in each of the two topics extracted from self-

reported definition of indebtedness are listed in the right panel. The shade of background color underlying each word represents the probability of this word in the current topic. **b** The regression coefficients indicating the independent contributions of emotions on the self-reported need to repay after receiving help ($n = 1598$ help-acceptance events) and decisions to reject help ($n = 1598$ help-acceptance events and 395 help-rejection events). Data are presented as the regression coefficient +/− SE of the estimate from regression.

based topic modeling[53] to the emotion-related words extracted from the 100 words with the highest weight/frequency in the definitions of indebtedness based on annotations from an independent sample of raters ($n = 80$). We demonstrate that indebtedness is comprised of two latent topics (Fig. S1b, c). Topic 1 accounted for 77% of the variance of emotional words, including communal-concern-related words such as "guilt," "feel," "feel sorry," "feel indebted," and "gratitude". In contrast, Topic 2 accounted for 23% of the emotional word variance, including words pertaining to burden and negative bodily states, such as "uncomfortable," "uneasy," "trouble," "pressure," and "burden" (Fig. 2a-III). These results support the relationship between indebtedness and the feelings of guilt and obligation posited by our conceptual model.

Next, we examined how these distinct feelings relate to participants' self-reported responses to the help (Fig. 2b). Participants described events in which they chose to accept help and reported their experienced emotions. We found that self-reported indebtedness ($\beta = 0.20 \pm 0.04$, 95%CI = [0.13, 0.27], $t(1592) = 5.60$, $p < 0.001$, linear regression, two-tailed, FDR corrected), guilt ($\beta = 0.12 \pm 0.04$, 95%CI = [0.04, 0.19], $t(1592) = 2.98$, $p = 0.004$), obligation ($\beta = 0.09 \pm 0.04$, 95% CI = [0.01, 0.16], $t(1592) = 2.27$, $p = 0.024$), and gratitude ($\beta = 0.38 \pm 0.04$, 95%CI = [0.30, 0.45], $t(1592) = 9.86$, $p < 0.001$) all independently contributed to participants' reported need to repay after receiving help. Participants also described events, in which they chose to reject help and reported their anticipated counterfactual emotions had they instead accepted the benefactor's help[54]. Decisions to reject help were negatively associated with gratitude ($\beta = -0.87 \pm 0.06$, 95%CI = [−0.99, −0.74], $z(1986) = -13.65$, $p < 0.001$, logistic regression, two-tailed, FDR corrected), but positively associated with indebtedness ($\beta = 0.23 \pm 0.10$, 95%CI = [0.04, 0.42], $z(1986) = 2.40$, $p = 0.017$), guilt ($\beta = 0.46 \pm 0.09$, 95%CI = [0.29, 0.65], $z(1986) = 5.06$, $p < 0.001$), and obligation ($\beta = 0.28 \pm 0.06$, 95%CI = [0.16, 0.40], $z(1986) = 4.70$, $p < 0.001$). These results, based on subjective experiences, indicate that while gratitude, guilt, and obligation all contribute to reciprocating favors, only gratitude appears to be associated with increasing the likelihood of accepting help. The guilt and obligation components of indebtedness instead appear to be associated with increasing the likelihood of rejecting help.

**Benefactor's intentions cause diverging components of indebtedness**

We next sought to more specifically examine how indebtedness impacts behavior in the context of a laboratory-based task involving interactions between participants in Study 2a ($n = 51$, Fig. 3). In this

task, participants were randomly paired with a different anonymous same-gender co-player (benefactor) in each trial and were instructed that they would receive 20 s of pain stimulation in the form of a burst of medium intensity electrical shocks. The participant was informed that each benefactor had been endowed with 20 yuan (~$2.7 USD) and made a decision about how much to spend from this endowment to reduce the duration of pain experienced by the participant (i.e., benefactor's cost) during a separate lab visit. Unbeknownst to the participant, each benefactor's cost was predetermined by a computer program (Table S2). After seeing how much money the benefactor chose to spend, the participant reported how much they believed this benefactor expected them to reciprocate (i.e., second-order belief of the benefactor's expectation for repayment). In half of the trials, the participant had to passively accept the benefactor's help; in the other half, the participant could freely decide whether to accept or reject the benefactor's help. Finally, at the end of each trial, participants decided how much of their own 25 yuan endowment (~$3.4 USD) they wanted to allocate to the benefactor as reciprocity for their help. We experimentally manipulated participants' beliefs about the benefactors' intentions by providing additional information regarding the benefactors' expectations of reciprocation. Each participant was instructed that before making decisions, some benefactors knew that the participant would be endowed with 25 yuan and could decide whether to allocate some endowments to them as reciprocity (i.e., Repayment possible condition), whereas the other benefactors were informed that the participant had no chance to reciprocate after receiving help (i.e., Repayment impossible condition). In fact, participants could reciprocate in both conditions during the task. After the task, all trials were displayed again in a random order and participants recalled how much they believed the benefactor cared for them (i.e., perceived care), as well as their feelings of indebtedness, obligation, guilt, and gratitude in response to the help they received for each trial. To ensure incentive compatibility, five trials were randomly selected to be enacted and participants received the average number of shocks and money based on their decisions at the end of the experiment. We ran an additional version of this experiment (Study 2b, $n = 57$), in which we further systematically varied the exchange rate of how much it cost the benefactor to reduce the participant's duration of pain (i.e., help efficiency). However, we did not observe any significant interaction effect between efficiency and any of other experimental variables in Study 2b (Table S6) and chose to combine these two studies for all Study 2 analyses ($n = 108$, Table S3; see Supplementary Notes and Tables S4 to S6 for separate results).

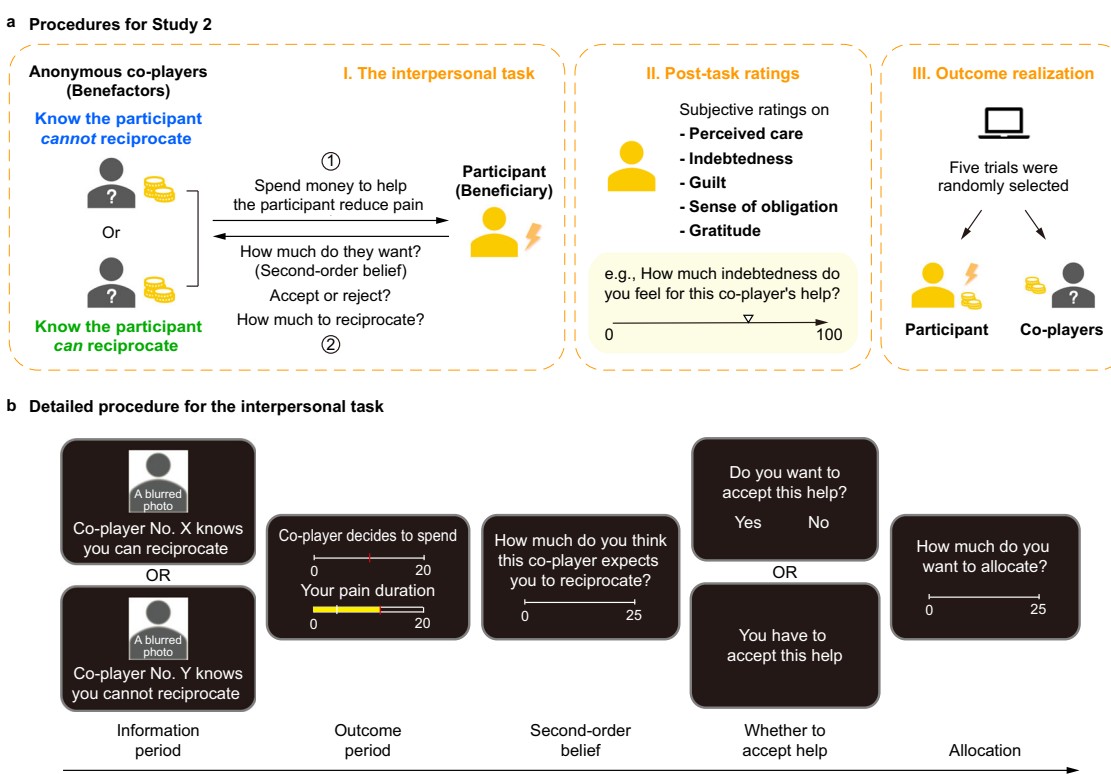

**Fig. 3 | Experimental procedures for Study 2. a** General procedures. In the interpersonal task (I), the participant was instructed that anonymous co-players (benefactors) made single-shot decisions to help reduce the duration of the participant's pain, and the participant, in turn, (1) reported how much they believed the benefactor expected them to reciprocate for their help (i.e., second-order belief), (2) decided whether to accept help, and (3) decided how much money to return to the benefactor. After the interpersonal task, participants recalled how much they believed the benefactor cared for them (i.e., perceived care), as well as their feelings of indebtedness, obligation, guilt, and gratitude in response to the help they received for each trial (II. Post-task ratings). At the end of the experiment, five trials in the interpersonal task were randomly selected to determine the participant's final amount of pain and payoff, and the selected benefactor's final payoffs (III. Outcome realization). **b** Detailed procedure for the interpersonal task. In each round, a different anonymous same-gender benefactor (represented by a blurred photo and a participant ID) decided how much of their endowment to

spend (i.e., benefactor's cost) to reduce the participant's pain duration. The more the benefactor spent, the more the duration of the participant's pain decreased. Participants indicated how much they thought the benefactor expected them to reciprocate (i.e., second-order belief). In half of the trials, the participant had to passively accept the benefactor's help; in the other half, the participant could freely decide whether to accept or reject the benefactor's help. Finally, at the end of each trial, the participant decided how much of their own endowment they wanted to allocate to the benefactor as reciprocity for their help. Unbeknownst to participants, benefactors' decisions (i.e., benefactor's cost) were pre-determined by the computer program (Table S2). We manipulated the perception of the benefactors' intentions by providing additional information about whether the benefactor knew the participant could (i.e., Repayment possible condition), or could not (i.e., Repayment impossible condition) reciprocate after receiving help. In fact, participants could reciprocate in both conditions during the task.

Our experimental manipulation successfully impacted participants' appraisals of the benefactors' hidden intentions behind their help. Participants reported increased second-order beliefs of the benefactors' expectations for repayment ($\beta = 0.53 \pm 0.03$, 95%CI = [0.47, 0.59], $t(107.04) = 15.71$, $p < 0.001$, linear mixed model, two-tailed) and decreased perceived care ($\beta = -0.31 \pm 0.02$, 95%CI = [−0.35, −0.27], $t(106.44) = -13.89$, $p < 0.001$) (Fig. 4a, Table S3) when the participant believed the benefactor knew they could reciprocate (Repayment possible) compared to when they could not reciprocate (Repayment impossible). Both of these effects were amplified as the benefactor spent more money to reduce the participant's duration of pain (Fig. 4b, c, Table S3; second-order belief: $\beta = 0.22 \pm 0.02$, 95%CI = [0.18, 0.26], $t(103.83) = 13.13$, $p < 0.001$; perceived care: $\beta = -0.08 \pm 0.01$, 95%CI = [−0.10, −0.06], $t(98.59) = -6.64$, $p < 0.001$). In addition, perceived care was negatively associated with second-order beliefs ($\beta = -0.44 \pm 0.04$, 95%CI = [−0.56, −0.32], $t(89.56) = -11.29$, $p < 0.001$) controlling for the effects of experimental variables (i.e., extra information about benefactor's intention, cost, and efficiency).

The belief manipulation not only impacted the participants' appraisals, but also their feelings. Our conceptual model predicts that

participants will feel indebted to benefactors who spent money to reduce their pain, but for different reasons depending on the perceived intentions of the benefactors. Consistent with this prediction, participants reported feeling indebted in both conditions, but slightly more in the Repayment impossible compared to the Repayment possible condition (Fig. 4a, Fig. S2a, Table S3, $\beta = -0.09 \pm 0.03$, 95%CI = [−0.15, −0.03], $t(105.81) = -2.98$, $p = 0.003$). Moreover, participants reported feeling greater obligation ($\beta = 0.30 \pm 0.03$, 95%CI = [0.24, 0.36], $t(106.82) = 9.28$, $p < 0.001$), but less guilt ($\beta = -0.25 \pm 0.02$, 95%CI = [−0.29, −0.21], $t(106.30) = -10.30$, $p < 0.001$), and gratitude ($\beta = -0.27 \pm 0.02$, 95%CI = [−0.31, −0.23], $t(106.35) = -13.18$, $p < 0.001$) in the Repayment possible condition relative to the Repayment impossible condition (Fig. 4a, Fig. S2b–d, Table S3). Similar to the appraisal results, these effects were magnified as the benefactor's cost increased (Fig. S2b–d, Table S3; obligation: $\beta = 0.11 \pm 0.01$, 95%CI = [0.09, 0.13], $t(100.74) = 8.85$, $p < 0.001$; guilt: $\beta = -0.05 \pm 0.01$, 95%CI = [−0.07, −0.03], $t(96.94) = -4.28$, $p < 0.001$; gratitude: $\beta = -0.06 \pm 0.01$, 95%CI = [−0.08, −0.04], $t(99.31) = -4.20$, $p < 0.001$).

We conducted two separate types of multivariate analyses to characterize the relationships between appraisals and emotions. First, exploratory factor analysis (EFA) on the subjective appraisals

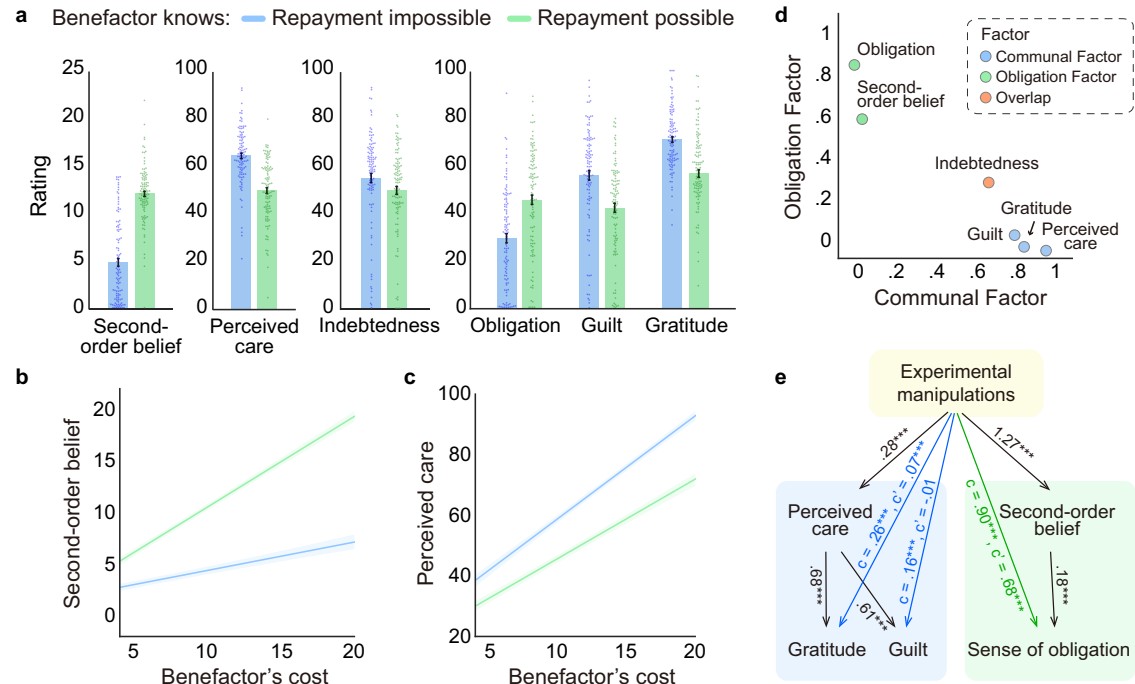

**Fig. 4 | Appraisals and emotional responses to benefactor's help with different intentions. a** Participants' appraisals (i.e., second-order belief of how much the benefactor expected for repayment and perceived care) and emotion ratings (indebtedness, obligation, gratitude, and guilt) in Repayment impossible and Repayment possible conditions. Each dot represents the average rating in the corresponding condition for each participant ($n = 108$ participants). Data are presented as mean values +/− SEM. **b, c** Participant's second-order beliefs of how much the benefactor expected for repayment and perceived care plotted as functions of extra information about benefactor's intention (Repayment impossible vs. Repayment possible) and benefactor's cost. Data are presented as mean values +/− SEM. SEMs were generated via bootstrapping that respected the repeated measurements within each participant ($n = 24$ trials for each condition in Study 2a/28 trials for each condition in Study 2b, 108 participants). **d** Factor analysis showed that participants'

appraisals and emotions could be explained by two independent factors, which appeared to reflect two distinct subjective experiences. The Communal Factor reflects participants' perception that the benefactor cared about their welfare and resulted in emotions of gratitude and guilt (Blue), while the Obligation Factor reflects participants' second-order beliefs about the benefactor's expectation for repayment and the sense of obligation (Green). **e** Simplified schematic representation of mediation analysis. See full model in Fig. S3c. Results showed that second-order beliefs and perceived care appraisals differentially mediated the effects of the experimental manipulations on emotional responses. Second-order beliefs mediated the effects of the experimental manipulations on the sense of obligation (Green), while perceived care mediated the effects of experimental manipulations on gratitude and guilt (Blue).

and emotion ratings in Study 2 revealed that 66% of the variance in ratings could be explained by two factors (Fig. 4d, and Fig. S2e; Fig. S3a, b). The Communal Factor reflected participants' perception that the benefactor cared about their welfare and resulted in emotions of guilt and gratitude, while the Obligation Factor reflected participants' second-order beliefs about the benefactor's expectation for repayment and the sense of obligation. Interestingly, indebtedness moderately loaded on both factors supporting its mixed relationship to guilt and obligation. Second, a mediation analysis revealed that second-order beliefs and perceived care appraisals differentially mediated the effects of the experimental manipulations on emotional responses (total indirect effect = 0.59 ± 0.04, 95%CI = [0.51, 0.67], $Z = 14.49$, $p < 0.001$, two-tailed; model performance: $\chi^2 = 9.68$, $df = 4$, CFI = 1.00, TLI = 0.997, RMSEA = 0.023, SRMR = 0.004; Fig. 4e and Fig. S3c). Second-order beliefs mediated the effects of the experimental manipulations on obligation (Indirect effect = 0.22 ± 0.03, 95%CI = [0.16, 0.29], $Z = 7.18$, $p < 0.001$), while perceived care mediated the effects of the experimental manipulations on guilt (Indirect effect = 0.17 ± 0.01, 95%CI = [0.15, 0.20], $Z = 13.23$, $p < 0.001$) and gratitude (Indirect effect = 0.19 ± 0.01, 95%CI = [0.17, 0.22], $Z = 13.72$, $p < 0.001$). Together, these results provide further support for the predictions of our conceptual model that indebtedness is comprised of two distinct feelings. The guilt component arises from the belief that the benefactor acts from altruistic intentions (i.e., perceived care), while the

obligation component arises when the benefactor's intentions are perceived to be strategic (e.g., expecting repayment).

## Beneficiary's behaviors are influenced by benefactor's intentions

Next, we examined participants' behaviors in response to receiving help from a benefactor. Specifically, we were interested in how much participants would reciprocate after receiving the favor and also whether they might outright reject the benefactor's help given the opportunity. We found that participants reciprocated more money as a function of the amount of help received from the benefactor, $\beta = 0.63 ± 0.02$, 95%CI = [0.59, 0.67], $t(109.12) = 25.60$, $p < 0.001$, linear mixed model, two-tailed (Fig. 5a, Table S3). This effect was slightly enhanced in the Repayment impossible condition relative to the Repayment possible condition, $\beta = −0.03 ± 0.01$, 95%CI = [−0.05, −0.01], $t(130.29) = −2.99$, $p = 0.003$. A mixed-effect logistic regression revealed that when given the chance, participants were more likely to reject help in the Repayment possible condition when they reported more obligation (rejection rate = 0.37 ± 0.10), compared to the Repayment impossible condition (rejection rate = 0.30 ± 0.03), $\beta = 0.27 ± 0.08$, 95%CI = [0.11, 0.43], $z(2788) = 3.64$, $p < 0.001$, two-tailed (Fig. 5b, Table S3). Moreover, as the benefactor's cost increased, participants were less likely to reject the help ($\beta = −0.65 ± 0.13$, 95%CI = [−0.90, −0.40], $z(2788) = −5.16$, $p < 0.001$). No significant interaction effect

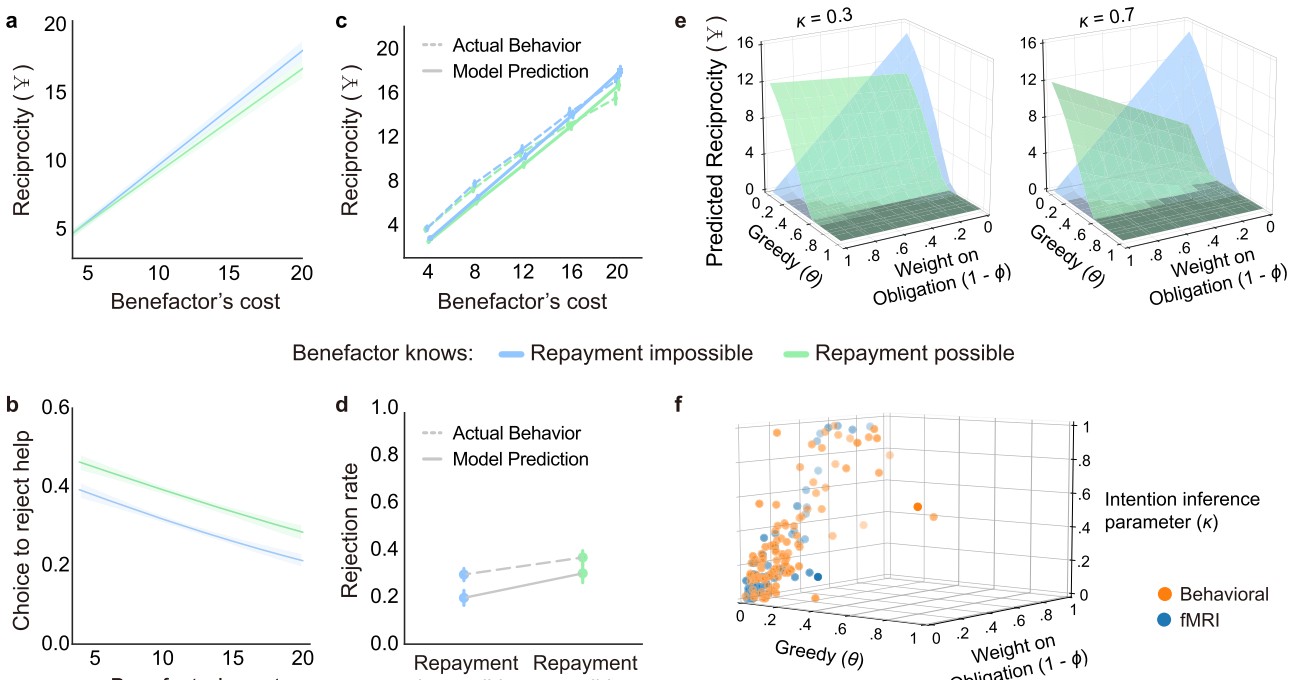

**Fig. 5 | Computational models of indebtedness. a** Participants' reciprocity behavior in each trial plotted as a function of information about benefactor's intention (Repayment impossible vs. Repayment possible) and benefactor's cost. **b** Participants' decisions to accept or reject help in each trial plotted as a logit function of information about benefactor's intention and benefactor's cost. For a and b, data are presented as mean values +/− SEM. SEMs were generated via bootstrapping that respected the repeated measurements within each participant ($n = 12$ trials for each condition in Study 2a/14 trials for each condition in Study 2b, 108 participants). **c** The observed amounts of reciprocity after receiving help and predictions generated by the reciprocity model at each level of the benefactor's cost in Repayment impossible and Repayment possible conditions. **d** The observed rates of rejecting help and predictions generated by the help-acceptance model in Repayment impossible and Repayment possible conditions. For c and d, data are presented as mean values +/− SEM ($n = 108$ participants). **e** Model simulations for predicted reciprocity behavior in Repayment impossible and Repayment possible conditions at different parameterizations. The $y$-axis shows the average values of the predicted amount of reciprocity across all levels of benefactor's cost. The model predicted reciprocity changes as a function of the tradeoff between communal and obligation feelings based on $\phi$ and interacts with the intention inference parameter $\kappa$. Increased emphasis on obligation corresponds to increased reciprocity to favors in the Repayment possible condition, but decreased reciprocity in the Repayment impossible condition; this effect is amplified as $\kappa$ increases. **f** Best fitting parameter estimates of the computational model for reciprocity decisions for each participant ($n = 108$ and $n = 53$ participants for behavioral and fMRI studies, respectively).

between condition and benefactor's cost was observed ($\beta = 0.07 \pm 0.07$, 95%CI = [−0.07, 0.21], $z(2788) = 1.08$, $p = 0.279$).

## Computational models of how indebtedness impacts behavior

Building on our conceptual model of indebtedness, we developed two computational models using a Psychological Game Theoretic framework[34–36] to predict reciprocity and help-acceptance decisions that maximize the beneficiary's expected utility based on the competing latent motivations of self-interest, communal concern (i.e., guilt and gratitude), and obligation (Eq. 1).

$$U(D_B) = \theta_B * \pi_B + (1 - \theta_B) * \left( \phi_B * U_{Communal} + (1 - \phi_B) * U_{Obligation} \right) \quad (1)$$

The central idea is that upon receiving a favor $D_A$ from a benefactor $A$, the beneficiary $B$ chooses an action $D_B$ that maximizes his/her overall utility $U$. This utility is comprised of a mixture of values arising from self-interest $\pi$ weighted by a greed parameter $\theta$, and feelings of communal concern $U_{Communal}$ and obligation $U_{Obligation}$, which are inferred from the appraisals of $D_A$ and weighted by the parameter $\phi$. Larger $\phi$ values reflect the beneficiary's higher sensitivity to feelings of communal concern relative to obligation. $U_{Communal}$ reflects a linear combination of guilt and gratitude components (see Methods).

The *reciprocity model* (Model 1.1) predicts the amount of money reciprocated to the benefactor, while the *help-acceptance model*

(Model 2.1) predicts binary decisions to accept or reject help. Though the two models are conceptually similar, the values of $U_{Communal}$ and $U_{Obligation}$ are computed slightly differently due to differences in the types of data (i.e., continuous vs. binary decisions) and how appraisals are inferred. It is important to note that in the reciprocity model, we are unable to distinguish between the separate motivations of guilt and gratitude because both positively contribute to reciprocity. In contrast, based on the findings from Study 1, we divide up the parameter space for the help-acceptance model such that $\phi > 0$ indicates a preference for gratitude and motives accepting the help, while $\phi < 0$ indicates a preference for guilt and motives rejecting the help (see Methods).

For both models, we define $U_{Obligation}$ as the appraisal of the amount of money that $B$ believes $A$ expects them to return (i.e., $B$'s second-order belief $E_B''$) normalized by $B$'s endowment size $\gamma_B$.

$$U_{Obligation} = \begin{cases} -\left(\frac{E_B'' - D_B}{\gamma_B}\right)^2 & \text{Reciprocity model} \\ -\frac{E_B''}{\gamma_B} & \text{Help − acceptance model} \end{cases} \quad (2)$$

where $E_B''$ is operationalized as $D_A$ in the Repayment possible condition and zero in the Repayment impossible condition.

$$E_B'' = \begin{cases} 0 & \text{Repayment impossible condition} \\ D_A & \text{Repayment possible condition} \end{cases} \quad (3)$$

In contrast, we define $U_{Communal}$ in terms of the appraisal of how much $B$ believes $A$ cares about their welfare (i.e., perceived care $\omega_B$).

$$U_{Communal} = \begin{cases} -\left(\frac{\omega_B * \gamma_B - D_B}{\gamma_B}\right)^2 & \text{Reciprocity model} \\ \omega_B & \text{Help} - \text{acceptance model} \end{cases} \quad (4)$$

We assume that $B$ infers perceived care $\omega_B$ proportional to how much $A$ spent $D_A$ from their endowment $\gamma_A$ and that this effect might be mitigated by the amount of money $B$ believes $A$ expects them to return (i.e., second-order belief $E_B''$).

$$\omega_B = \frac{D_A - \kappa_B * E_B''}{\gamma_A} \quad (5)$$

where $\kappa$ reflects the degree to which the perceived strategic intention $E_B''$ reduces the perceived altruistic intention $\omega_B$. See details for the models in Methods.

## The reciprocity model

We performed a rigorous validation of our reciprocity model (Model 1.1) across a variety of different types of evaluations. First, we were interested in how well our computational model for reciprocity captured trial-to-trial reciprocity decisions. Model parameters were estimated by minimizing the sum of squared error between the model predicted behaviors and participants' reciprocity decisions separately for each participant. Our computational model was able to successfully predict participants' continuous reciprocity decisions after receiving help ($r^2 = 0.79$, $\beta = 0.88 \pm 0.01$, 95%CI = [0.85, 0.91], $t(87.82) = 59.36$, $p < 0.001$, linear mixed model, two-tailed; Fig. 5c; Fig. S6a–c; Table S16) and significantly outperformed other plausible models, such as: (a) a model with linear formulations of utilities for self-interest, communal concern, and obligation (Model 1.2), (b) models that solely included the term for communal concern (Model 1.3) or obligation (Model 1.4) besides the self-interest term, (c) models with separate parameters for self-interest, communal concern, and obligation (Model 1.5 and Model 1.6), (d) a model that assumes participants reciprocate purely based on the benefactors helping behavior (i.e., tit-for-tat) (Model 1.7)[37,38], and (e) a model that assumes that participants are motivated to minimize inequity in payments (Model 1.8)[52,55] (Tables S10 and S11). Parameter recovery tests indicated that the parameters of the reciprocity model were highly identifiable (Pearson correlation between true and recovered parameters over the three parameters, two-tailed, reciprocity $r = 0.94 \pm 0.07$, 95%CI = [0.80, 1.08], $t(322) = 50.70$, $p < 0.001$; Table S14).

A simulation of the reciprocity model across varying combinations of the $\theta$, $\phi$ and $\kappa$ parameters revealed diverging predictions of the beneficiaries' response to favors in Repayment impossible and Repayment possible conditions (Fig. 5e). Not surprisingly, greedier individuals (higher $\theta$) are less likely to reciprocate others' favors. However, reciprocity changes as a function of the tradeoff between communal and obligation feelings based on $\phi$ and interacts with the intention inference parameter $\kappa$. Increased emphasis on obligation corresponds to increased reciprocity to favors in the Repayment possible condition, but decreased reciprocity in the Repayment impossible condition; this effect is amplified as $\kappa$ increases. We found that most participants had low $\theta$ values (i.e., greed), but showed a wide range of individual differences in $\kappa$ and $\phi$ parameters (Fig. 5f). Interestingly, the degree to which the perceived strategic intention reduced the perceived altruistic intention during intention inference $\kappa$, was positively associated with the relative weight on obligation $(1 - \phi)$ during reciprocity ($r = 0.79 \pm 0.05$, 95%CI = [0.70, 0.85], $t(106) = 13.14$, $p < 0.001$, Pearson correlation, two-tailed). This suggests that the participants who cared more about the benefactor's strategic

intentions also tended to be motivated by obligation when deciding how much money to reciprocate.

Beyond just simply predicting behaviors, we conducted additional validations to assess how well our reciprocity model's predictions of second-order beliefs ($E_B''$; Eq. 3) and perceived care ($\omega_B$; Eq. 5) were able to capture trial-to-trial variations in participants' self-reported ratings of appraisals and feelings. Regression analyses showed that the reciprocity model's representations of $E_B''$ and $\omega_B$ were associated with trial-to-trial variations in self-reported values of second-order beliefs of the benefactor's expectation for repayment ($\beta = 0.68 \pm 0.03$, 95%CI = [0.62, 0.74], $t(106.93) = 21.48$, $p < 0.001$, linear mixed model, two-tailed; Fig. S5a, b) and perceived care ($\beta = 0.72 \pm 0.03$, 95%CI = [0.66, 0.77], $t(107.30) = 26.76$, $p < 0.001$; Fig. S5c, d), respectively. Moreover, $\kappa$ appeared to successfully capture individual differences as participants who reported an overall higher level of perceived care were also observed to have a higher overall level of $\omega_B$ ($r = 0.27 \pm 0.09$, 95%CI = [0.09, 0.44], $t(106) = 2.92$, $p = 0.004$, Pearson correlation, two-tailed).

We further assessed if the reciprocity model's representations of perceived care ($\omega_B$) and second-order belief ($E_B''$) appraisals corresponded to self-reported communal and obligation feelings. Supporting our predictions, the reciprocity model's predictions of $\omega_B$ significantly predicted self-reported guilt ratings ($\beta = 0.47 \pm 0.03$, 95% CI = [0.42, 0.53], $t(105.5) = 17.21$, $p < 0.001$, linear mixed model, two-tailed) as well as the Communal Factor scores obtained from EFA in Fig. 4d ($\beta = 0.81 \pm 0.03$, 95%CI = [0.75, 0.87], $t(107.58) = 25.81$, $p < 0.001$), while the model predictions of $E_B''$ significantly predicted self-reported obligation ratings ($\beta = 0.38 \pm 0.03$, 95%CI = [0.32, 0.44], $t(106.20) = 12.67$, $p < 0.001$) and the Obligation Factor scores ($\beta = 0.64 \pm 0.06$, 95%CI = [0.56, 0.71], $t(106.03) = 15.97$, $p < 0.001$).

## The help-acceptance model

Next, we evaluated how well the help-acceptance model (Model 2.1) was able to capture participants' trial-to-trial decisions of whether or not to accept the benefactor's help. We estimated the parameters by maximizing the log-likelihood of the predicted probability of the chosen option (accept or reject) separately for each participant. Overall, we found that our model was able to predict participants' decisions to accept or reject help (accuracy = 80.37%; Fig. 5d; Fig. S6d–f; Table S17). The help-acceptance model outperformed models with separate parameters for self-interest, communal concern, and obligation (Model 2.4 and 2.5), but did not significantly outperform models that solely included terms for communal concern (Model 2.2) or obligation (Model 2.3) (Tables S12 and S13). This likely stems from a slight instability in the parameterization of the model (see Methods and Discussion), which is confirmed by the moderate level of identifiability indicated by the parameter recovery tests (Pearson correlation between true and recovered parameters over the four parameters, two-tailed, $r = 0.43 \pm 0.40$, 95%CI = [−0.35, 1.21], $t(430) = 9.92$, $p < 0.001$; and Table S15).

## Communal concern and obligation involve distinct neural processes

Next, in Study 3 ($n = 53$), we explored the neural bases of indebtedness and examined whether the processing of communal concern and obligation involve differential brain processes as suggested by our conceptual model. Participants completed the same task as Study 2 while undergoing fMRI scanning, except that they were unable to reject help. First, we successfully replicated all of the behavioral results observed in Study 2 (see detailed statistics in Tables S1 and S7, and Figs. S7 and S8). In addition, we found that the two-factor EFA model we estimated using the self-report data in Study 2 generalized well to the independent sample in Study 3 using confirmatory factor analysis (CFA; Fig. S7g), with comparative fit indices exceeding the >0.9 acceptable threshold (CFI = 0.986, TLI = 0.970) and the root mean

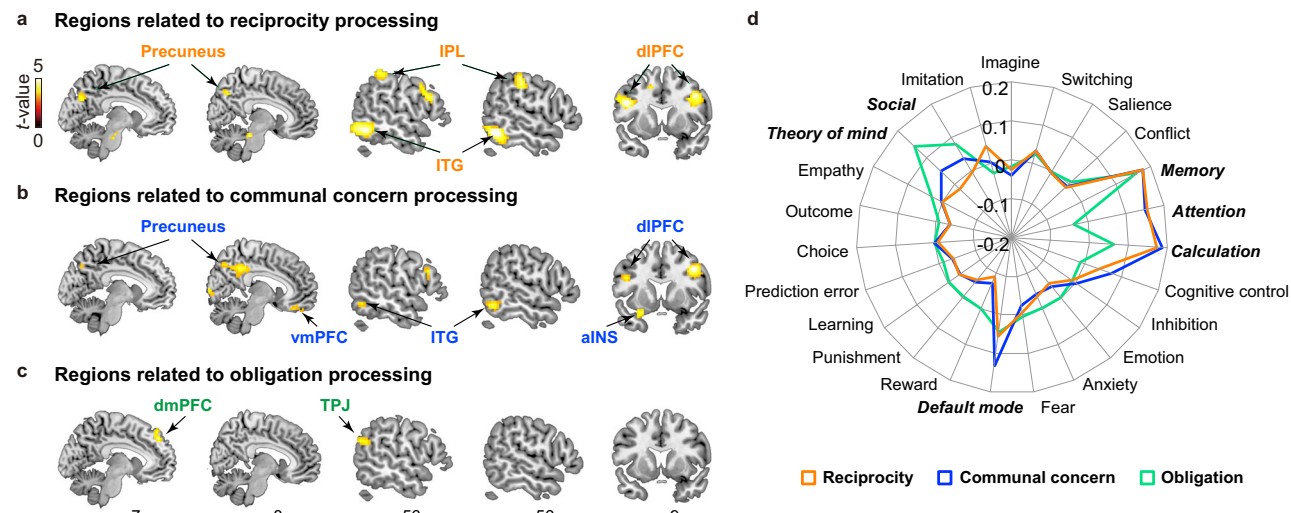

**Fig. 6 | Neural processes associated with reciprocity, communal concern and obligation. a** Brain regions responding to trial-by-trial levels of reciprocity. **b** Brain regions responding parametrically to trial-by-trial communal concern, which depended on the perceived care from the help ($\omega_B$). **c** Brain regions identified in the parametric contrast for obligation ($E_B''$), the responses of which monotonically increased in the Repayment possible condition relative to the Repayment impossible condition. For **a**–**c**, $n = 53$ participants. The brain maps were thresholded using cluster correction FWE $p < 0.05$ with a cluster-forming threshold of $p < 0.001$ [121]. **d** Results of meta-analytical decoding for the neural correlates of reciprocity, communal concern and obligation, respectively. IPL inferior parietal lobule, dlPFC dorsolateral prefrontal cortex, ITG inferior temporal gyrus, vmPFC ventromedial prefrontal cortex, aINS anterior insula, dmPFC dorsomedial prefrontal lobe, TPJ temporal parietal junction.

square error of approximation and the standardized root mean squared residual were within the reasonable fit range of <0.08 (RMSEA = 0.079, SRMR = 0.019)[56–58].

Second, we performed univariate analyses to identify brain processes during the Outcome period (Fig. 3b), where participants learned about the benefactor's decision to help. Using a model-based fMRI analytic approach[59], we fit three separate general linear models (GLMs) to each voxel's timeseries to identify brain regions that tracked different components of the computational model. These included trial-by-trial values for: (1) the amount of reciprocity, (2) communal concern, which depended on the perceived care from the help ($\omega_B$), and (3) obligation, which depended on the second-order belief of the benefactor's expectation for repayment ($E_B''$) (see Methods). We found that trial-by-trial reciprocity behavior correlated with activity in bilateral dlPFC, bilateral inferior parietal lobule (IPL), precuneus, and bilateral inferior temporal gyrus (ITG) (Fig. 6a, Table S18). Trial-by-trial communal feelings tracked with activity in the anterior insula, vmPFC, precuneus, bilateral dlPFC, and bilateral ITG (Fig. 6b; Table S18). The processing of obligation was associated with activations in dmPFC and left TPJ (Fig. 6c, Table S18).

To aid in interpreting these results, we performed meta-analytic decoding[60] using the Neurosynth database[61]. Reciprocity-related activity was primarily associated with "Attention," "Calculation," and "Memory" terms. Communal feelings related activity was similar to the reciprocity results, but was additionally associated with "Default mode" term. Obligation activity was highly associated with terms related to "Social," "Theory of mind (ToM)," and "Memory" (Fig. 6d). Together, these neuroimaging results reveal differential neural correlates of feelings of communal concern and obligation and support the role of intention inference in the generation of these feelings proposed by our conceptual model. The processing of communal feelings was associated with activity in vmPFC, an area in default mode network that has been linked to gratitude[62–64], positive social value, and kind intention[65,66], as well as the insula, which has been previously related to guilt[54,67,68]. In contrast, the processing of obligation was associated with activations of theory of mind network, including dmPFC and TPJ, which is commonly observed when representing other peoples' intentions or strategies[66,69,70].

## Neural utility model of indebtedness predicts reciprocity behavior

Finally, we sought to test whether we could use signals directly from the brain to construct a utility function and predict reciprocity decisions (Fig. 7a). Using brain activity during the Outcome period of the task (Fig. 3b), we trained two whole-brain models using principal components regression with 5-fold cross-validation[71–73] to predict the appraisals associated with communal concern ($\omega_B$) and obligation ($E_B''$) separately for each participant. These whole-brain patterns were able to successfully predict the model representations of these feelings for each participant on new trials, though with modest effect sizes (communal concern pattern: average $r = 0.21 \pm 0.03$, 95%CI = [0.15, 0.27], $t(52) = 7.00$, $p_{perm} < 0.001$, one-sample permutation $t$-test, two-tailed; obligation pattern: average $r = 0.10 \pm 0.03$, 95%CI = [0.04, 0.16], $t(52) = 3.33$, $p_{perm} = 0.004$; Fig. 7a). Moreover, these patterns appear to be capturing distinct information as they were not spatially correlated, $r = 0.03 \pm 0.04$, 95%CI = [−0.05, 0.11], $p = 0.585$. These results did not simply reflect differences between the Repayment possible and Repayment impossible conditions as the results were still significant after controlling for this experimental manipulation (communal concern: average $r = 0.18 \pm 0.02$, 95%CI = [0.14, 0.22], $t(52) = 9.00$, $p_{perm} < 0.001$; obligation: average $r = 0.04 \pm 0.02$, 95% CI = [−0.02, 0.08], $t(52) = 2.00$, $p_{perm} = 0.024$). Furthermore, we were unable to successfully discriminate between these two conditions using a whole brain classifier (accuracy = 55.0 ± 1.25%, permutation $p = 0.746$).

Next, we assessed the degree to which our brain models could account for reciprocity behavior. We used cross-validated neural predictions of communal concern ($\omega_B$) and obligation ($E_B''$) feelings as inputs to our computational model of reciprocity behavior instead of the original terms:

$$U(D_B) = \theta_B * \pi_B + (1 - \theta_B)$$
$$* \left( \phi_B * \overrightarrow{\beta_{map}} \cdot \overrightarrow{Communal_{map}} + (1 - \phi_B) * \overrightarrow{\beta_{map}} \cdot \overrightarrow{Obligation_{map}} \right)$$

(6)

**a  Multivariate patterns for model components**

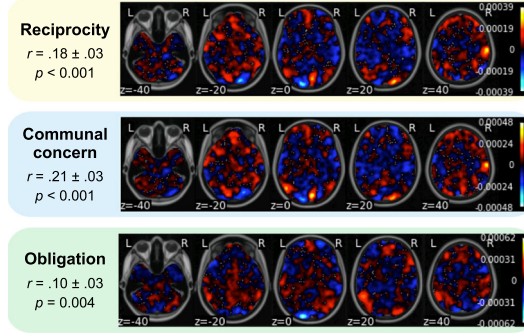

**b  Performance of the neural utility model**

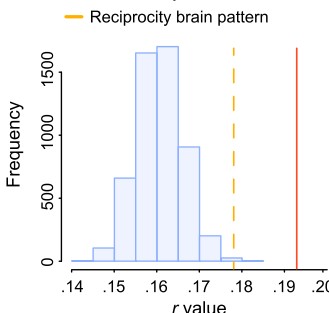

**c  Individual differences in spatial alignment of multivariate patterns**

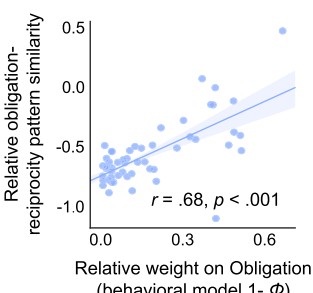

**Fig. 7 | Neural utility model of indebtedness. a** Unthresholded multivariate patterns used to predict the amounts of reciprocity (Yellow), trial-by-trial communal concern ($\omega_B$, Blue) and obligation ($E_B''$, Green) separately. **b** We assessed the importance of the participant-specific model parameters estimated from the neural utility model (i.e., $\phi$) by generating a null distribution of predictions after permuting the estimated $\phi$ parameter across participants 5,000 times. The red line indicates the performance of our neural utility model ($r$ value of prediction), and the orange line indicates the performance of the whole-brain model trained to directly predict reciprocity. The subject-specific weightings were important in predicting behavior as our neural utility model significantly outperformed a null distribution of average prediction accuracy after randomly shuffling the participant weights (Blue). **c** The relationship between the relative weight on obligation ($1 - \phi$) derived from behavior and a neurally derived metric of how much obligation vs. communal feelings influenced reciprocity behavior (Pearson correlation, two-tailed, $r = 0.68 \pm 0.09$, 95%CI = [0.51, 0.81], $t(51) = 6.69$, $p < 0.001$, $n = 53$ participants). Data are presented as regression line +/− 95% confidence interval for the regression.

where $\overrightarrow{\beta_{map}}$ refers to the pattern of brain activity during the Outcome period (Fig. 3b) of a single trial and $\overrightarrow{Commual_{map}}$ and $\overrightarrow{Obligation_{map}}$ refer to the multivariate brain models predictive of each participant's communal concern and obligation utilities respectively. We were able to reliably predict reciprocity behavior with our computational model informed only by predictions of communal and obligation feelings derived purely from brain responses (average $r = 0.19 \pm 0.02$, 95%CI = [0.15, 0.23], $t(52) = 9.50$, $p_{perm} < 0.001$; AIC = 317.70 ± 5.00; Fig. 7b). As a benchmark, this model numerically outperformed a whole-brain model trained to directly predict reciprocity (average $r = 0.18 \pm 0.03$, 95%CI = [0.12, 0.24], $t(52) = 6.00$, $p_{perm} < 0.001$; AIC = 317.54 ± 5.00; Fig. 7a), but this difference did not reach statistical significance ($t(52) = 1.64$, $p = 0.108$, Cohen's $d = 0.23$, 95%CI = [−0.002, 0.022], permutation-based paired $t$-test, two-tailed).

We performed several additional validations of the neural utility model to demonstrate its overall performance. First, we found that the parameter $\phi$ estimated from the neural utility model, which reflects the tradeoff between communal concern and obligation, strongly correlated with the same parameter estimated from the behavioral computational model across participants ($r = 0.88 \pm 0.04$, 95%CI = [0.80, 0.93], $t(51) = 13.3$, $p < 0.001$, Pearson correlation, two-tailed). Second, we assessed the individual specificity of $\phi$ derived from the neural utility model, to test how uniquely sensitive individuals are to communal concern versus obligation. To do so, we generated a null distribution of predictions after permuting the estimated $\phi$ parameter across participants 5,000 times. We found that the participant-specific weightings were highly important in predicting behavior as our neural utility model significantly outperformed null models using randomly shuffled $\phi$ parameters, $p_{perm} < 0.001$ (Fig. 7b). Third, we tested how well our neural-utility model reflected the trade-off between an individual's feelings of communal concern and obligation estimated from the behavioral model. We hypothesized that the relative influence of a particular feeling on behavior should be reflected in the spatial alignment of their corresponding brain patterns[74] (see Methods). Our results support this hypothesis. Participants who cared more about obligation relative to communal concern (higher behavioral $1 - \phi$) also exhibited greater spatial alignment between their obligation and reciprocity brain patterns relative to communal concern and reciprocity patterns (Fig. 7c; $r = 0.68 \pm 0.09$, 95%CI = [0.51, 0.81], $t(51) = 6.69$,

$p < 0.001$, Pearson correlation, two-tailed). These results provide evidence at the neural level indicating that individuals appear to trade-off between feelings of communal concern and obligation when deciding how much to reciprocate after receiving help from a benefactor.

## Discussion

Gift-giving, favor-exchanges, and providing assistance are behavioral expressions of relationships between individuals or groups. While favors from friends and family often engender reciprocity and gratitude, they can also elicit guilt in a beneficiary who may feel that they have burdened a benefactor. Favors in more transactive relationships, however, can evoke a sense of obligation in the beneficiary to repay the favor. In this study, we sought to develop a conceptual model of indebtedness that outlines how appraisals about the intentions behind a favor are critical to the generation of these distinct feelings, which in turn motivates how willing individuals are to accept or reject help and ultimately reciprocate the favor.

We provide a systematic validation of this conceptual model of indebtedness across three separate experiments by combining a large-scale online questionnaire, behavioral measurements in an interpersonal game, computational modeling, and neuroimaging. First, we used an open-ended survey to capture lay intuitions about indebtedness based on regression analysis of past emotional experiences and topic modeling based-text analysis of self-reported definitions. Overall, we find strong support that the feeling of indebtedness can be further separated into two distinct components—guilt for burdening the favor-doer and obligation to repay the favor. Using topic modeling on lay definitions of indebtedness, we find that guilt and gratitude appear to load on the same topic, while feeling words pertaining to burden and negative body states load on a separate topic. Second, we used a laboratory task designed to elicit indebtedness in the context of an interpersonal interaction and specifically manipulated information intended to shift the benefactor's perceptions of the beneficiary's intentions underlying their decisions. Although our manipulation was subtle, we find that it was able to successfully change participants' appraisals about how much the beneficiary cared about them and their beliefs about how much money the benefactor expected in return. Consistent with appraisal theory[28–33], these shifts in appraisals influenced participants' subjective feelings and ultimately their behaviors. Intentions perceived to be altruistic led to increased guilt and gratitude, while intentions viewed as more strategic increased feelings of

obligation. While all three feelings increased reciprocity decisions, the guilt and obligation components of indebtedness increased the probability of rejecting help when that option was available to the participant.

One contribution of this work is the use of computational modeling to predict reciprocity and help-acceptance decisions in our interpersonal task based on our conceptual model of indebtedness. The majority of empirical research on indebtedness[21,46,47,75] and other emotions[76,77] has relied on participants' self-reported feelings in response to explicit questions regarding social emotions, which has significant limitations, such as its dependence on participants' ability to introspect[78,79]. Formalizing emotions using computational models is critical to advancing theory, characterizing their impact on behavior, and identifying associated neural and physiological substrates[39,80,81]. However, the application of computational modeling to the study of social emotions is a relatively new enterprise[39,54,82,83]. Previous research has had success modeling belief-dependent utility using Psychological Game Theory[36,37] in interactive social contexts. Building on this work, we model participants' appraisals and emotions[28–33] based on the state of the game to predict two different types of decisions (i.e., reciprocity & help-acceptance)[39]. The current work contributes to a growing family of game theoretic models of social emotions such as guilt[34,54], gratitude[84], and anger[85,86], and can be used to infer feelings in the absence of self-report, providing avenues for investigating other social emotions.

We provide a rigorous validation of our computational models using behaviors in the interpersonal game, self-reported subjective experiences, and neuroimaging. First, we can accurately predict participants' reciprocity and help-acceptance decisions. Second, we observed that the model predictions of second-order belief and perceived care in the reciprocity model accurately captured participant's trial-to-trial self-reported appraisal and feeling ratings. Third, our brain imaging analyses demonstrate that each feeling reflects a distinct psychological process, and that intention inference plays a key role during this process. Consistent with previous work on guilt[54,67,68,87] and gratitude[62–64], our model representation of communal concern correlated with increased activity in the insula, dlPFC, and default mode network including the vmPFC and precuneus. Obligation, in contrast, captured participants' second order beliefs about expectations of repayment and correlated with increased activation in regions routinely observed in mentalizing including the dmPFC and TPJ[66,69,70].

We provide an even stronger test of our ability to characterize the neural processes associated with indebtedness by deriving a "neural utility" model. Previous work has demonstrated that it is possible to build brain models of preferences that can predict behaviors[88,89] and the hidden motives behind the behaviors[90]. Here, we trained multivoxel patterns of brain activity to predict participants' communal and obligation utility. We then used these brain-derived representations of communal concern and obligation to predict how much money participants ultimately reciprocated to the beneficiary. Remarkably, we found that this neural utility model of indebtedness was able to predict individual decisions entirely from brain activity and numerically outperformed (but not significantly) a control model that provided a theoretical upper bound of how well reciprocity behavior can be predicted directly from brain activity. Importantly, the neural utility model was able to accurately capture each participant's preference for communal concern relative to obligation. We observed a significant drop in our ability to predict behavior when we randomly shuffled the weighting parameter across participants. In addition, we find that the more the pattern of brain activity predicting reciprocity behavior resembled brain patterns predictive of communal concern or obligation, the more our behavioral computational model weighted this feeling in predicting behavior, demonstrating that these distinct appraisals/feelings are involved in motivating reciprocity decisions.

This work advances our theoretical understanding of social emotions. First, we highlight the complex relationship between gratitude and indebtedness. We propose that feeling cared for by a benefactor, which we call communal concern[44,45], is comprised of both guilt and gratitude. Each emotion diverges in valence, with gratitude being positive[6–9], and guilt being negative[40–42,44,54], but both promote reciprocity behavior. When faced with the offer of help, anticipated gratitude should motivate the beneficiary to accept help in order to establish or promote a relationship[6,7], whereas anticipated guilt should motivate the beneficiary to reject help out of concern to protect the benefactor from incurring a cost[44,54,91]. Although we observed support for this prediction, our interpersonal task was not designed to explicitly differentiate guilt from gratitude, which limited the ability of our reciprocity model to capture the specific contributions of guilt and gratitude to communal concern and likely impacted identifiability of the parameters of the help-acceptance model. Future work might continue to refine the relationship between these two aspects of communal concern both in terms of behaviors in experiments and computations in models[54,62–64,67,68,87].

Second, our conceptual model provides a framework to better understand the role of relationships and contexts in generating feelings of indebtedness within a single individual. Different types of relationships (see Clark and Mills's theory of communal and exchange relationships[4,5], and Alan Fiske's Relational Models Theory[92]) have been theorized to emphasize different goals and social norms which can impact social emotions[93–95]. For example, communal relationships prioritize the greater good of the community and are more conducive to altruistic sharing, which can be signaled by altruistic favors[3–5]. In contrast, exchange relationships are more transactional in nature[2,4,5,10–12] and emphasize maintaining equity in the relationship, which can be signaled by strategic favors[92]. Our conceptual model proposes that perceptions of the benefactor's intentions directly impact the feelings experienced by the beneficiary (e.g., guilt & obligation). Although we deliberately attempted to minimize aspects of the relationship between the benefactor and beneficiary by making players anonymous to control for reputational effects, future work might experimentally manipulate these relationships to directly test the hypothesis that relationship types differentially moderate the responses of gratitude and subcomponents of indebtedness.

Third, we present evidence exploring the relationship between indebtedness and guilt. Guilt and indebtedness are interesting emotions in that they are both negatively valenced, yet promote prosocial behaviors. In previous work, we have operationalized guilt as arising from disappointing a relationship partner's expectations[39,54,55,96], which is conceptually related to the feeling of obligation in this paper. This feeling results from disappointing a relationship partner or violating a norm of reciprocity and is a motivational sentiment evoked by social expectations reflecting a "sense of should" that is associated with other negative affective responses such as feelings of pressure, burden, anxiety, and even resentment[49–51]. In other work, we have investigated how guilt can arise from causing unintended harm to a relationship partner[68,97]. This is conceptually more similar to how we frame guilt here, which arises from the feeling that one has unnecessarily burdened a relationship partner even though the help was never explicitly requested by the beneficiary. We believe that continuing efforts to refine mathematical models of emotions across a range of contexts, will eventually allow the field to move beyond relying on the restrictive and imprecise semantics of linguistic labels to define emotion categories (e.g., guilt, gratitude, indebtedness, obligation, feeling, motivation, etc.).

Our study has several potential limitations, which are important to acknowledge. First, although we directly and conceptually replicate our key findings across multiple samples, all of our experiments recruit experimental samples from a Chinese population. It is possible that there are cultural differences in the experience of indebtedness, which

may not generalize to other parts of the world. For example, compared with Westerners who commonly express gratitude when receiving benevolent help, Japanese participants (East Asian population) often respond with "Thank you" or "I am sorry", indicating their higher experience of guilt after receiving favors[40,41]. Cultural differences may perhaps reflect how the two components of indebtedness are weighted, with guilt being potentially more prominent in East Asian compared to Western populations, reflecting broader cultural differences in collectivism and individualism. Second, our computational models may oversimplify the appraisal and emotion generating processes. These models operationalize the appraisals of perceived care and second-order belief using information available to each participant in the task (i.e., benefactor's helping behavior and manipulation about the participant's ability to reciprocate), which may not generalize to other experimental contexts without modification. Although our computational models performed well in capturing participants' behaviors in this task, we emphasize the importance of continued refinement. Third, future research is needed to extend our conceptual model by differentiating different types of help-receiving events (e.g., help when moving to a new apartment vs. help during a period of sickness) and manipulating other related contexts, such as gift-receiving[23] and help-seeking[17].

In summary, in this study, we develop a comprehensive and systematic conceptual model of indebtedness and validate it across three studies combining a large-scale online questionnaire, an interpersonal game, computational modeling and neuroimaging. A key aspect to this work is the emphasis on the role of appraisals about the intentions behind a favor in generating distinct feelings of guilt and obligation, which in turn motivates how willing beneficiaries are to accept or reject help and ultimately reciprocate the favor. Together, these findings highlight the psychological, computational, and neural mechanisms underlying the hidden costs of receiving favors[22–24].

## Methods

### Participants

For Study 1, participants (1808 graduate and undergraduate students) were recruited from Zhengzhou University, China to complete an online questionnaire. Participants were excluded if they filled in information irrelevant to the question or experiment in the essay question (e.g., this question is boring, or I don't want to answer this question; 189 participants), leaving 1619 participants (self-reported gender: 812 females, $18.9 \pm 2.0$ (SD) years). While 98.7% of participants reported the events of receiving help, 24.4% of participants reported the events of rejecting help within the past 1 year, resulting in 1991 effective daily events (1598 help-acceptance events and 395 help-rejection events). To extract the words related to emotions and feelings in the definition of indebtedness, 80 additional graduate and undergraduate students (45 females, $22.6 \pm 2.58$ years) were recruited from different universities in Beijing to complete an online word classification task (see Supplementary Methods). No data were excluded from the analysis in the task.

For Study 2a (behavioral study), 58 graduate and undergraduate Chinese Han students were recruited from Zhengzhou University, China, and 7 participants were excluded due to equipment malfunction, leaving 51 participants (self-reported gender: 33 females, $19.9 \pm 1.6$ years) for data analysis. For Study 2b (behavioral study), 60 graduate and undergraduate Chinese Han students were recruited from Zhengzhou University, China, and 3 participants were excluded due to failing to respond in more than 10 trials, leaving 57 participants (45 females, $20.1 \pm 1.8$ years) for data analyses.

For Study 3, 57 right-handed healthy graduate and undergraduate Chinese Han students from Beijing, China took part in the fMRI scanning. Four participants with excessive head movements (>2 mm) were excluded, leaving 53 participants (self-reported gender: 29 females, $20.9 \pm 2.3$ years) for data analysis.

For all experiments, none of the participants reported any history of psychiatric, neurological, or cognitive disorders. No statistical method was used to predetermine sample size. All experiments were carried out in accordance with the Declaration of Helsinki and were approved by the Ethics Committee of the School of Psychological and Cognitive Sciences, Peking University. Informed written consent was obtained from each participant prior to participating.

### Experimental procedures

**Study 1—Online questionnaire.** All participants completed the same questionnaire on the Questionnaire Star platform (https://www.wjx.cn/) online using their mobile phones. The questionnaire consisted of two parts. Each participant was asked to recall a daily event in which they received help (part 1) or rejected help (part 2) from others, and to answer the questions regarding their appraisals (second-order belief and perceived care), emotions (gratitude, indebtedness, guilt, and obligation), behaviors (the need to reciprocate and whether to reject help) and other details of this event (e.g., the relationship with the benefactor, the participant's benefit, and the benefactor's cost). To explore how participants defined indebtedness, we asked participants to answer the following two questions about the definition of indebtedness after recalling the event: (1) In the context of helping and receiving help, what is your definition of indebtedness? (Fill-in-the-blanks test) (2) In daily life, what do you think is/are the source(s) of indebtedness? Multiple-choice question with four options "Negative feeling for harming the benefactor", "Negative feeling for the pressure to repay caused by other's ulterior intentions", "Both" and "Neither". See the online questionnaire in Supplementary Methods.

**Study 2—Interpersonal task.** During Session 1 (the main task), each participant played multiple single-shot rounds of the interpersonal task (Fig. 3) as a Receiver. In each trial, the participant was paired with an anonymous same-gender Decider (co-player), and was informed that the co-player in each trial was distinct from the ones in any other trials and only interacted with the participant once during the experiment. In each round, the participant had to receive a 20-second pain stimulation with the intensity of 6. The participant was instructed that each co-player: (a) had come to the lab before the participant, (b) had been endowed with 20 yuan (-\$2.7 USD), and (c) had decided whether and how much to spend from this endowment to help the participant reduce the duration of pain (i.e., benefactor's cost, $D_A$). The more the benefactor spent, the shorter the duration of the participant's pain experience. The maximum pain reduction was 16 s to ensure that participants had some amount of pain on each trial. Unbeknownst to the participant, the benefactor's costs were predetermined by the computer program (Table S2).

Each trial began by informing the participant which Decider from Stage 1 was randomly selected as the co-player for the current trial (Information period, 4 s), with the blurred photo and the participant ID of the co-player, and extra information regarding the co-player's intention to help (see below). The co-player's decision on how much they chose to spend to help the participant was presented (Outcome period, 5 s). Next, the participant indicated how much they thought this co-player expected them to reciprocate (i.e., Second-order belief of the co-player's expectation for repayment; continuous rating scale from 0 to 25 using mouse, step of 0.1 yuan, < 8 s). In half of the trials, the participant had to passively accept the co-player's help (force-accept situation). The sentence "You have to accept this help" was presented on the screen (4 s). In the other half, the participant could decide whether or not to accept the co-player's help (free-choice situation, < 4 s). The order of options was counterbalanced across trials. If the participant accepted the help, the co-player's cost and the participant's pain reduction in this trial would be realized according to the co-player's decision; if the participant did not accept the help, the co-player would spend no money and the duration of participant's pain

stimulation would be the full 20 s. At the end of each trial, the participant was endowed with 25 yuan (~3.4 USD) and decided how much they wanted to allocate to the co-player as reciprocity in this trial from this endowment ($D_B$, continuous choice from 0 to 25 using mouse, step of 0.1 yuan, < 8 s). We focused on two types of behaviors in this help-receiving context: (1) the participant's amounts of allocation when they passively accept the co-player's help (i.e., reciprocity decisions), and (2) the participant's decisions of whether to accept or reject help in free-choice situation (i.e., help-acceptance decisions).

We manipulated the perceived intention of the co-player (i.e., the benefactor) by providing participants with extra information regarding the co-player's expectation of reciprocity (i.e., extra information about benefactor's intention) below the co-player's subject id at the beginning of each trial. Each participant was instructed that before making decisions, some co-players were informed that the participant would be endowed with 25 yuan and could decide whether to allocate some endowments to them as reciprocity (i.e., Repayment possible condition). The other co-players were informed that the participant had no chance to reciprocate after receiving help (i.e., Repayment impossible condition). In fact, participants could reciprocate in both conditions during the task. The endowment of the co-player ($\gamma_A$) was always 20 yuan, and the endowment of the participant ($\gamma_B$) in each trial was always 25 yuan. The endowment of the participant was always larger than the endowment of the co-player to make the participant believe that the co-player expected repayments in Repayment possible condition.

In Study 2a, we manipulated the participant's beliefs about the benefactor's intentions (condition: Repayment possible vs. Repayment impossible) and the benefactor's cost in a within-subject design. In Study 2b, to disentangle the effect of the benefactor's cost and participant's benefit (i.e., pain reduction), we additionally manipulated the exchange rate between the co-player's cost and participant's pain reduction (i.e., efficiency; the efficiency was always 1.0 in Study 2a) in a within-subject design.

During Session 2 of the interpersonal task, all of the trials from Session 1 were displayed again in a random order. The participant was asked to recall how much they believed the benefactor cared about them, as well as their feelings of indebtedness, obligation, guilt, and gratitude at the time point when they had received the help of the co-player, but had not indicated decisions of accept/reject help and the amount of reciprocity. Ratings were conducted on a scale from 0 to 100, with 0 represented "not at all" and 100 represented "extremely intense". At the end of the experiment, five trials in Session 1 were randomly selected to be realized. See Supplementary Methods for additional details about procedures and experimental designs.

### Study 3—FMRI study

Each participant came to the scanning room individually. The two sessions of the task in the fMRI study were identical to Study 2a, except that participants always had to accept their co-player's help. Session 1 (the main task; Fig. 3b) was conducted in the fMRI scanner, while Sessions 2 was conducted after participants exited the scanner. Before and after each period, a fixation cross was presented for a variable interval ranging from 2 to 6 s, which was for the purpose of fMRI signal deconvolution. See Supplementary Methods for additional details about procedures and experimental designs.

### Data analyses in Study 1 (online questionnaire)
**Validating conceptual model with emotion ratings.** We first attempted to validate the conceptual model using the emotional ratings for daily-life events of receiving and rejecting help obtained from online-questionnaire in Study 1. We conducted between-participant linear regressions predicting indebtedness ratings from guilt and obligation ratings (see Supplementary Methods).

**Validating conceptual model with self-reported appraisals.** Next, we summarized participants' self-reported sources of their feelings of indebtedness. We calculated the frequency that participants selected each of the four options in the question "In daily life, what do you think is/ are the source(s) of indebtedness?" in Study 1 (Fig. S1A), as well as how often that participants attributed "Negative feeling for harming the benefactor" and "Negative feeling for the pressure to repay caused by other's ulterior intentions" as the sources of indebtedness (i.e., the frequency of choosing each single option plus the frequency of choosing "Both of the above").

**Validating conceptual model with topic modeling.** We also attempted to validate the conceptual model by applying topic modeling to participant's open-ended responses describing their own definition of indebtedness in Study 1. We conducted word-splitting, counted the frequency that each participant used each word and transformed these frequencies using Term Frequency-Inverse Document Frequency (TF-IDF)[98,99]. The 100 words with the highest weight/frequency in the definitions of indebtedness were extracted (Table S20). These 100 words were then classified by an independent sample of participants ($n = 80$) into levels of appraisal, emotion, behavior, person and other. We conducted Latent Dirichlet Allocation (LDA) based topic modeling on the emotional words of indebtedness using collapsed Gibbs sampling. We selected the best number of topics by comparing the models with topic numbers ranging from 2 to 15 using 5-fold cross validation. Model goodness of fit was assessed using perplexity[100]. We found that the two-topic solution performed the best (Fig. S1c). See Supplementary Methods for additional details about topic modeling.

**Validating conceptual model with self-reported behaviors.** We next sought to test the predictions of the conceptual model using the self-reported behaviors from Study 1. First, we used data from Part 1 of the questionnaire (help-receiving events) and used linear regression to predict self-reported need to reciprocate from self-reported feelings of indebtedness, guilt, obligation and gratitude. Second, we combined the data of the events associated with receiving (Part 1) and rejecting help (Part 2) and used logistic regression to classify rejecting from accepting behavior using self-reported counterfactual ratings of indebtedness, guilt, and obligation, and gratitude.

### Data analyses in Study 2 (interpersonal task)
**The effects of experimental manipulations on participants' appraisal, emotional and behavioral responses.** To test the effects of experimental manipulations on beneficiary's appraisals (i.e., second-order belief and perceived care), emotions (i.e., gratitude, indebtedness, guilt, and obligation) and behaviors (reciprocity and help-acceptance decisions), we conducted linear mixed effects analyses for each dependent variable separately with the benefactor's cost, extra information about benefactor's intention (Repayment possible vs. Repayment impossible) and their interaction effect as fixed effects and the participant ID as a random intercept and slope (see Supplementary Methods).

**Relationships between appraisals and emotions.** To reveal the relationships between appraisals (i.e., second-order belief and perceived care) and emotions (i.e., indebtedness, guilt, obligation, and gratitude), we estimated the correlations between these variables at both within-participant (Fig. S3a, Table S8) and between-participant levels (Fig. S3b, Table S9). Given the strong correlations between appraisals and emotions (Fig. S3a, b, Tables S8 and S9), we conducted a factor analysis to examine the relationship between appraisals and emotions[101]. We first applied EFA in Study 2 to identify the number of common factors and the relationships between appraisals and emotions. Next, we conducted CFA using the data of Study 3 to test the two-factor model built by

Study 2 in an independent sample. Finally, to test whether the two appraisals mediated the observed effects of experimental variables on emotional responses, we conducted a multivariate mediation analysis using structural equation modeling. See Supplementary Methods for additional details about EFA, CFA, and mediation analysis.

**Computational modeling.** We built separate computational models predicting participant's reciprocity decisions and help-acceptance decisions based on the conceptual model of indebtedness (see Table S19 for all model object definitions). The utility of each behavior $U(D_B)$ was modeled based on the competing latent motivations of self-interest, communal concern (guilt and gratitude), and obligation using Eq. 1.

For reciprocity decisions, self-interest $\pi_B$ was defined as the percentage of money kept by the participant out of their endowment $\gamma_B$. For help-acceptance decisions, self-interest $\pi_B$ for accepting help was defined as the percentage of pain reduction from the maximum amount possible, which depended on how much the benefactor spent to help $D_A$ and the exchange rate between the benefactor's cost and the participant's benefit $\mu$ (i.e., help efficiency).

$$\pi_B = \begin{cases} \frac{\gamma_B - D_B}{\gamma_B} & \text{Reciprocity model} \\ \frac{D_A * \mu}{\max(D_A * \mu)} & \text{Help} - \text{acceptance model} \end{cases} \quad (7)$$

For each trial, we modeled the participant's second-order belief $E_B''$ of how much they believed the benefactor expected them to reciprocate (Eq. 3) based on the amount of help offered by the benefactor $D_A$ and whether the benefactor knew repayment was possible. In the Repayment impossible condition, participants knew that the benefactor did not expect them to reciprocate, so we set $E_B''$ to zero. However, in the Repayment possible condition, the benefactor knew that the participant had money that they could spend to repay the favor. In this condition, we modeled the $E_B''$ as proportional to the amount of money the benefactor spent to help the participant.

The appraisals of perceived care $\omega_B$ (Eq. 5) were defined as a function of the benefactor's cost $D_A$ and second-order belief $E_B''$. Specifically, we assumed that the perceived care from help increased as a linear function of how much the benefactor spent $D_A$ from his/her endowment $\gamma_A$. In other words, the more the benefactor spent, the more care the participant would perceive from the help. However, we assume that this effect is mitigated by the second-order belief of the benefactor's expectation for repayment $E_B''$. That is, when faced with a specific amount of benefactor's cost, if the participant thought this benefactor expected more repayment, the less care the participant would perceive from the help. Here, the parameter $\kappa$ ranges from [0, 1] and represents the degree to which the perceived strategic intention $E_B''$ reduces the perceived altruistic intention $\omega_B$. This creates a nonlinear relationship between $\omega_B$ and $E_B''$ such that the relationship is negative when $\kappa$ is close to one, positive when $\kappa$ is close to zero, and uncorrelated in the current dataset with $\kappa = 0.32 \pm 0.01$, $\beta = -0.03 \pm 0.03$, 95%CI = [−0.09, 0.03], $t(112.32) = -1.23$, $p = 0.222$ (Fig. S4).

Furthermore, our conceptual model proposed that the feelings of obligation and communal concern (guilt and gratitude) stem from the appraisals of benefactor's strategic intention (i.e., second-order belief) and altruistic intention (i.e., perceived care from the help), respectively. This hypothesis was supported by the results of the mediation analysis (Fig. 4e). Thus, we modeled the utilities of obligation and communal concern (i.e., $U_{Obligation}$ and $U_{Communal}$) as the functions of $E_B''$ and $\omega_B$, respectively (Eqs. 2 and 4).

**Predicting reciprocity decisions.** To predict continuous reciprocity decisions, we assume that participants were motivated to meet the expectation of the benefactor due to the sense of obligation, and thus maximized $U_{Obligation}$ by minimizing the difference between the

amount they reciprocated $D_B$ and their second-order belief of how much they believed the benefactor expected them to return $E_B''$, scaled by the participant's endowment size $\gamma_B$ (Eq. 2). We note that our mathematical operationalization of obligation here is more akin to how we have previously modeled guilt from disappointing others in previous work[34,39,54,55] (see also Discussion). We also assumed that participants were motivated to reciprocate in response to the benefactor's perceived care due to guilt and gratitude (i.e., communal concern), and thus maximized $U_{Communal}$ by minimizing the difference between the benefactor's reciprocity $D_B$ and their perception of how much they believed the benefactor cared about them $\omega_B$, scaled by the participant's endowment size $\gamma_B$ (Eq. 4).

Based on our conceptual model (Fig. 1), we defined $U_{Communal}$ as a mixture of feelings of gratitude $U_{Gratitude}$ and guilt $U_{Guilt}$, in which the parameter $\delta_B$ ranged from [0,1] and reflected how much gratitude contributed to communal concern in comparison to guilt.

$$U_{Communal} = \delta_B * U_{Gratitude} + (1 - \delta_B) * U_{Guilt} \quad (8)$$

We note that both guilt and gratitude positively contribute to reciprocity and our interpersonal task was not designed to explicitly differentiate the effects of guilt and gratitude, which precluded our ability to estimate the specific value of $\delta_B$ for predicting reciprocity decisions due to a lack of identifiability. Thus, in this paper we can only make inferences about the broader $U_{Communal}$, which may reflect guilt and/or gratitude. However, our help-acceptance model does attempt to differentiate the contributions of guilt and gratitude to decisions of whether or not to accept help as discussed below.

We modeled the utility $U$ associated with the participants' reciprocity decisions $D_B$ after receiving help in Eq. 1, where $\phi$ is a free parameter constrained between [0, 1] that captures the trade-off between feelings of communal concern and obligation. The reciprocity model (Model 1.1, Eq. 9) selects the participant's decision $D_B$ associated with the highest utility.

$$U(D_B) = \theta_B * \frac{\gamma_B - D_B}{\gamma_B} - (1 - \theta_B)$$
$$* \left( \phi_B * \left( \frac{\omega_B * \gamma_B - D_B}{\gamma_B} \right)^2 + (1 - \phi_B) * \left( \frac{E_B'' - D_B}{\gamma_B} \right)^2 \right) \quad (9)$$

**Predicting help-acceptance decisions.** We created a separate model (Model 2.1, Eq. 10) to predict help-acceptance decisions. $U_{Obligation}$ was defined as a linear function of $E_B''$ (Eq. 2). $U_{Communal}$ was defined as a linear function of $\omega_B$ (Eq. 4). We modeled the utility of accepting and rejecting help as:

$$\begin{cases} U(Accept) = \theta_B * \pi_B + (1 - \theta_B) * \left( \phi_B * U_{Communal} - (1 - |\phi_B|) * U_{Obligation} \right) \\ \qquad = \theta_B * \frac{D_A * \mu}{\max(D_A * \mu)} + (1 - \theta_B) * \left( \phi_B * \omega_B - (1 - |\phi_B|) * \frac{E_B''}{\gamma_B} \right) \\ U(Reject) = 0 \end{cases} \quad (10)$$

In this model, $U(Reject)$ was set to zero, because the participant's emotional responses would not change if the participant did not accept help. Increased obligation reduces the likelihood of accepting help to avoid being in the benefactor's debt[13,14,102]. In contrast, $U_{Communal}$ has a more nuanced influence on behavior, with guilt decreasing the likelihood of accepting help to avoid burdening a benefactor[34,54], and gratitude motivating accepting help to build a communal relationship[6,7]. However, because $U_{Communal} = U_{Guilt} = U_{Gratitude} = \omega_B$ in this formulation, there is no variability in the design for the model to be able to disentangle the effect of gratitude from that of guilt. To address this complexity, we constrain $\phi$ to be within the interval of [−1, 1], and explicitly divide up the parameter space such that $\phi > 0$ indicates a preference for gratitude and motives the

participants to accept the help, while $\phi < 0$ indicates a preference for guilt and motives the participants to reject the help.

$$\begin{cases} \phi_B > 0 & Gratitude \\ \phi_B < 0 & Guilt \end{cases} \qquad (11)$$

Regardless of whether the participant is motivated primarily by guilt or gratitude, participants can still have a mixture of obligation captured by $1 - |\phi|$, which ranges from [0, 1]. Unfortunately, if participants are equally sensitive to gratitude and guilt, $\phi$ will reduce to zero and the weight on obligation increases, which decreases the model fit and leads to some instability in the parameters.

See Supplementary Methods for the methods of model fitting, model comparison and parameter recovery for reciprocity decisions and help-acceptance decisions.

### Data analyses in Study 3 (FMRI Study)

**Univariate fMRI analyses.** FMRI data preprocessing (see Supplementary Methods) and univariate analyses were conducted using Statistical Parametric Mapping software SPM12 (Wellcome Trust Department of Cognitive Neurology, London). We used a model-based fMRI analytic approach[59] to identify brain regions that parametrically tracked different components of the computational model for reciprocity during the Outcome period of the task (5 s; Fig. 3b), where participants learned about the benefactor's decision to help. To ensure that each hypothesis tested had maximum variance, we chose to separately test each hypothesis using a separate model to minimize issues with multicollinearity. GLM 1 identified reciprocity related brain responses based on the parametric modulator of participant's reciprocity behavior $D_B$. GLM 2 identified brain responses related to communal concern based on the parametric modulator of the participant's appraisal of perceived care $\omega_B$. GLM 3 identified brain responses related to obligation, which we modeled as a linear contrast of the participant's second-order belief of the benefactor's expectation for repayment $E_B$'[54]. For whole brain analyses, all results were corrected for multiple comparisons using cluster correction $p < 0.05$ with a cluster-forming threshold of $p < 0.001$, which attempts to control for family wise error (FWE) using Gaussian Random Field Theory[103]. See Supplementary Methods for details of univariate fMRI analyses.

**Meta-analytical decoding.** To reveal the psychological components associated with the processing of reciprocity, communal concern and obligation, we conducted meta-analytic decoding using the Neurosynth Image Decoder[61] (http://neurosynth.org). This allowed us to quantitatively evaluate the spatial similarity[60] between any Nifti-format brain image and selected meta-analytical images generated by the Neurosynth database (see Supplementary Methods).

**Neural utility model of indebtedness.** We constructed a neural utility model by combining our computational model for reciprocity with multivariate pattern analysis (MVPA) with our open source Python NLTools package[104] version 0.3.14 (https://nltools.org/)[105]. First, using principal components regression with 5-fold cross-validation, we trained two separate multivariate whole-brain models predictive of communal concern ($\omega_B$) and obligation ($E_B$') terms in our behavioral model separately for each participant[71–73]. For each whole-brain model, we extracted the cross-validated prediction accuracy ($r$ value) for each participant, conducted $r$-to-$z$ transformation, and then conducted a one-sample permutation $t$-test to evaluate whether each model was able to significantly predict the corresponding term. We used the cross-validated models to generate predictions for each trial for each participant and then used brain-predicted communal concern ($\omega_B$) and obligation ($E_B$') feelings as inputs to our computational model of reciprocity behavior (Model 1.1, Eq. 9) instead of the original terms. We estimated the $\theta$ values (i.e., weight on greed) and $\phi$ weighting

parameters (i.e., relative trade-off between on communal concern and obligation) using the same procedure described in the behavioral computational modeling section. As a benchmark for our neural utility model, we used the same training procedure described above, but predicted trial-to-trial reciprocity behavior using principal components regression separately for each participant.

Finally, we were interested in evaluating how well we could estimate how much each participant had a relative preference for communal concern or obligation by computing the relative spatial alignment of their communal and obligation predictive spatial maps with their reciprocity predictive spatial map. We operationalized this relative pattern similarity as

$$relative\ pattern\ similarity = corr\left(\overrightarrow{Obligation_{map}}, \overrightarrow{Reciprocity_{map}}\right) \\ - corr\left(\overrightarrow{Commual_{map}}, \overrightarrow{Reciprocity_{map}}\right) \qquad (12)$$

We tested the Pearson correlation between this relative pattern similarity and the $(1 - \phi)$ parameters estimated by fitting the computational model (Eq. 1) directly to the participants' behaviors. See Supplementary Methods for additional details about neural utility model.

### Reporting summary

Further information on research design is available in the Nature Portfolio Reporting Summary linked to this article.

## Data availability

Behavioral data from all the three studies are available on GitHub (https://github.com/xiaoxuepsy/Indebtedness) and Zenodo (https://doi.org/10.5281/zenodo.8328235)[106]. The unthresholded first-level and second-level maps from the fMRI study are available on Open Science Framework (https://doi.org/10.17605/OSF.IO/K8RXH). Raw imaging data are available from the corresponding authors upon request due to privacy concern. Meta-analytic decoding was conducted using the meta-analytical images generated by the Neurosynth database (http://neurosynth.org)[61].

## Code availability

The codes used in the current study are available on GitHub https://github.com/xiaoxuepsy/Indebtedness and Zenodo (https://doi.org/10.5281/zenodo.8328235)[106].

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

## Acknowledgements

We thank Drs. Matthew Rushworth and Christian C. Ruff for their comments and suggestions on this article. In addition, we thank Ms. Wan Wang, Mr. Shuaiqi Li and Mr. Sensen Song for their assistances in data collection, Ms. Yunyan Duan for her advice in topic modeling, and Ms. Zhewen He for the preparation of the manuscript. Dr. Gao

thanks Dr. Can Tang, Dr. Li Zhang, Ms. Jiyan Lyu, and Ms. Kewalin Boonsatta for their support during the revision process. This work was supported by National Natural Science Foundation of China (71942001, X.Z. and X.G.; 32371094, 31900798, X.G.; 31630034, X.Z. and X.G.), Young Elite Scientists Sponsorship Program by China Association for Science and Technology (YESS20210176, 2021QNRC001, X.G.), the National Science Foundation of USA (CAREER 1848370, L.J.C.), the National Institute of Health (R01MH116026, L.J.C.), the Research Project of Shanghai Science and Technology Commission (20dz2260300, X.G. and X.Z.), the Fundamental Research Funds for the Central Universities (X.G. and X.Z.), and the STI 2030 - Major Projects 2021ZD0200500 (X.Z. and X.G.). We also acknowledge support from the Graduate School of Peking University to fund Dr. Gao's training at Dartmouth College.

## Author contributions

X.G., E.J., H.Y., X.Z., and L.J.C. designed the experiments. X.G. and H. L. implemented the study design and collected the data. X.G. and L.J.C. carried out the analyses. X.G., E.J., H.Y., X.Z., and L.J.C. wrote the paper. X.Z. and L.J.C. supervised the work. All authors provided critical revisions and approved the final paper for submission.

## Competing interests

The authors declare no competing interests.
