## [Peer Review File · Nature Communications]

Reviewers' Comments:

Reviewer #1:

Remarks to the Author:

This paper develops a theory of indebtedness, that is modeled using a value function over actions in response to a favor. The value function possesses self and other components. A straightforward self-benefit piece and a social cost comprised of a communal term (guilt) and an obligation term.

The paper's conclusions and appeals derives from three sources:

- (1) responses to an online questionnaire,
- (2) behavior in a task,
- (3) computational modeling of (2),
- (3) neuroimaging to provide evidence for their proposed theory.

General Impression

This paper enlists a voluminous amount of data and analysis in support of the proposed theory/model of indebtedness. The data are compelling but there are weaknesses in the exposition, most importantly in the description of the task that make it difficult to assess the paper. TO be clear, I found it difficult to understand in detail exactly what had been done or exactly how particular analyses were carried out. With that said, I found the paper rich and the ideas presented interesting and this paper can be of wide interest to the social neuroscience, decision neuroscience, and neuroimaging communities. I particularly liked the results presented in figure 4.

I have outlined specific complaints below, but the 'semantic envelop' of this review is that I really needed more clarity for each of the points.

Points.

Line 99. Equation 1. The Greek letters in the equation do not match those in the text.

Line 122. "predictions" seems to be too strong a term. it would be good to enumerate the predictions first, for example.

Line 194 (And figure 2C). First, it seems figure 2C deserves its own figure. Second, there are many details of the task that are omitted: 1) What exactly does the partner know about the partners? Is there actually another person for each trial? 2) How many trials are there? 3) Does the participant actually get shocked (is that the pain)? Is the endowment of the partner ($\gamma_{\{A\}}$?) always 20? Is the endowment of the participant ($\gamma_{\{B\}}$?) always 25?

Line 294. it is not clear that listing the parameters here helps since the equations are relegated to the methods.

Figure 4C. Why not show this separately for the two conditions.

Line 359. The definition of the linear contrast is unclear.

Line 674. "percentiles"? I believe this refers to Eq. 10, which is not well-explained. I think the first instance of $D_{\{B\}}(t)$ refers to the actual choice whereas the max is over the possible choices ($D_{\{B\}}(t)$). The error term that is squared seems to be the squared percentage error.

Eq. 12. What is the link between the difference between the utility of ACCEPT and that of REJECT and P? (I assume $D_{\{B\}}(t)$ is the participant's choice ACCEPT or REJECT.

In general it would be useful to see a summary of the parameter estimates for the models (e.g. distributions of the parameters).

Reviewer #2:

Remarks to the Author:

The authors present the results of three studies that aim to test a model of indebtedness, reflecting the trade-off between guilt (or communal concern) and the sense of obligation. Guilt is defined as the appraisal of the benefactor's altruistic intentions. Obligation is defined as the appraisal of the benefactor's strategic intentions. In the first study, they use an online questionnaire to characterize the subjective experience of indebtedness in Chinese participants. Participants describe their feelings during every-day life encounters that are then coded by independent raters. The results suggest two main factors – guilty and sense for obligation. Study 2 is a behavioral experiment in which benefactors spend some amount of their initial endowment to reduce pain experienced by the participants. In one condition (called strategic condition), the benefactors ostensibly were informed that the participant can reciprocate the favor. In the other condition (called altruistic condition) the benefactors were informed that the participants cannot reciprocate the favor. Participants reported feeling indebted to benefactors who spent money to reduce their pain, in both conditions, but slightly more in the altruistic compared to the strategic condition. Study 3 used the same paradigm as Study 2 and fMRI to regress the model parameters (assumed to reflect communal concern and sense of obligations) against neural activations. Moreover, they analyzed the brain activation during receiving the favor, interpreted as reciprocity. The results showed a large number of brain regions, including frontal, parietal, temporal areas. Further analyses (metaanalytic decoding based on Neurosync) indicated that reciprocity-related activation was linked to attention and memory processes, communal concern was associated to the default network, and sense of obligation was associated with terms related to "Social", "Theory of Mind" and "Memory". Finally, the authors submitted the brain activations associated with communal concern and sense of obligations to their model and showed that they can predict reciprocity behavior. The authors conclude that they developed a neural utility model of indebtedness that captures the tradeoff between feelings of communal concern and obligation when deciding how much to reciprocate after receiving help from a benefactor.

The manuscript use rich data sets and elaborate analyses to tackle an interesting and relevant research question. The paper is well written, and a computational model that disentangles the intentions behind altruistic behavior would be a welcome contribution to the literature of the field. However, I have conceptual and methodological concerns that dampen my enthusiasm.

Major concerns:

1. One main concern is related to the vague definitions of the concepts. Indebtedness is divided into guilt/ communal concern and sense of obligation. The Introduction section implies that guilt/ communal concern is a feeling while sense of obligation is "strategic". I think this distinction between "altruistic" vs "strategic/ self-serving" is not convincing, because guilt can also induce a strategy, namely the strategy to reduce the negative guilt-related emotions or arousal. In this case, behavior elicited by guilt would not be altruistic, but self-serving. On the other hand, it is plausible to assume that a sense of obligation can raise communal concern, which, based on the framework of the authors, would be "altruistic", rather than strategic and self-serving. In other parts of the manuscript (including the abstract) both constructs are labelled as feelings (e.g., "distinct feelings of guilt and obligation"; line 32), which makes the definition of the concepts even more confusing. Without a clear definition of the underlying concepts, it is hard to say what we learn from the proposed model.

2. As currently defined, it is unlikely that guilt/ communal concern and sense of obligation are two orthogonal concepts (see my first point). However, the assumption of orthogonality is an essential part of the proposed model ($\Phi * U_{\text{communal}} + (1 - \Phi) * U_{\text{obligation}}$). In Study 2, the variance inflation factor (VIF) was calculated and a PCA for dimensional reduction was performed, but in Studies 1 and 3 there are no analyses showing orthogonality of the concepts. In Studies 1 and 3, there were only marginal differences between the AIC of the model with self-reported guilt as only predictor and the model with both predictors (Table S1). This result indicates that model fit does not considerably improve by adding the second component (i.e., sense of obligation). In other,

words, guilt might be sufficient to model the data. In light of these results, I am not convinced that guilt and sense of obligation are conceptually and/or statistically orthogonal.

3. In Studies 2 and 3, the validity of the two components is inferred from self-reports that explicitly asked participants to indicate their feeling of guilt and obligations (e.g., "How much guilt do you feel for this Decider's decision?"; "How much afraid/pressure do you feel for the decider's expectation for repayment?"). These explicit questions might impose the demand to classify emotions in the two respective categories. It would be desirable to determine the validity (and assumed orthogonality) of the two components with more implicit measures.

4. The imaging results show activations in a large number of brain regions. The statistical threshold is set to voxel-level $P < 0.001$ (uncorrected) combined with cluster-level threshold $P < 0.05$ [family-wise error (FWE) corrected]. Given the lack of a priori neural hypotheses, I strongly recommend to present only FWE voxel-level results and to use only these FWE voxel-level corrected results to predict reciprocity behavior.

Minor points

1. In Studies 2 and 3, the partner was anonymous. Thus, reputation effects are minimized which might account for the stronger effects in the altruistic condition compared to the strategic condition. It would be good to discuss this point in the Discussion section.

2. The survey method in Study 1 is described sometimes as "large-scale experience sampling" and sometimes as an "online questionnaire". If a "large-scale experience sampling" is reported, it is not intuitively clear that this is a questionnaire and vice versa. This point should be clarified and described consistently.

3. Figure 2: The description of the paradigm (Fig. 2C) should be presented first (i.e., as panel A).

4. PCA is a data reduction method in which the "factors" (main components) are formed as linear combinations of the items. Measurement errors are not subtracted (based on the assumption that all items are free of measurement errors, which tends to be incorrect for most empirical data). As a result, the main components may actually contain error variance (e.g. Fabrigar, L. R., Wegener, D. T., MacCallum, R. C., & Strahan, E. J. (1999). Evaluating the use of exploratory factor analysis in psychological research. *Psychological Methods*, 4(3), 272–299. <https://doi.org/10.1037/1082-989X.4.3.272>). To overcome this problem, the authors might consider a confirmatory factor analysis to test the assumption of two components derived from Study 1.

5. Discussion (line 493): "One of the most notable contributions of this work is the development and validation of a computational model of indebtedness. The majority of research on emotions relies on self-reported subjective feelings, which has a number of limitations, such as its dependence on participants' ability to introspect." Given that the current study also used self reports to validate the two components (see Major concerns, point 3 above), this statement should be deleted or toned down accordingly.

6. The description of the sample in Study 1 is unnecessarily complicated "In total, the data of 1,619 (812 females, 18.9 ± 2.0 (SD) years), 51 (33 females, 19.9 ± 1.6 years), 57 (45 females, 20.1 ± 1.8 years), and 53 (29 females, 20.9 ± 2.3 years) healthy graduate and undergraduate students were included for Study 1". It should be explicitly stated how many participants have completed the questionnaire and how many have participated in the following procedures.

7. At page 16 of the SI it is stated that 6 LMMs were performed. Where these analyses controlled for multiple comparisons?

Reviewer #3:

Remarks to the Author:

There is a large body of psychological research on prosociality — people’s tendency to act (even altruistically at personal cost) to help others. Most of this research assumes that receiving help is an unproblematically good outcome for the beneficiary, so the puzzle is just why the benefactor is motivated to do it. However, as these authors note, receiving help is not a simple experience; indeed, the costs of receiving help are high enough that people often reject help altogether. The current project proposes that there are at least two distinct negative experiences associated with receiving help: feeling guilt for the burden on the benefactor, and feeling resentment for being coerced into reciprocation. They offer three rich and complementary sources of evidence for their claim: a survey; behaviour in a controlled interpersonal game, and patterns of neural activity.

The current paper tackles an important and under-explored topic with a rich and rigorous combination of methods. I think the work is an exciting contribution to the literature, and certainly deserved to be published. My comments below are intended to improve communication of the hypothesis and results.

(1) Study 1 and the conceptual model: Distinguish guilt from gratitude

The overall concern of this paper is how people react to receiving help, depending on the intention that the beneficiary attributes to the benefactor. Here, the authors measure 4 psychological reactions (indebtedness, gratitude, guilt, and “obligation”). Although I generally agree with their view, I disagree with two choices in how they present and describe it.

First, and most importantly, their theoretical model conflates gratitude and guilt. However, both conceptually and empirically, it seems important to keep gratitude and guilt separate. Conceptually, gratitude is a positive emotion and guilt is a negative emotion. In study 1, the survey, gratitude has a very weak association with the ‘guilt’ component in people’s emotional descriptions; and gratitude strongly predicts accepting help whereas guilt quite strongly predicts rejecting help. Furthermore, in the real world, gratitude and guilt are likely to occur in different kind of relationships (in Alan Fiske’s scheme, gratitude might be more likely after receiving help in communal sharing relationships, whereas guilt might be more likely after receiving help in authority ranking relationships); and thus be associated with different subsequent behaviour.

When the authors formalize their model, they currently do not have a separate term for gratitude versus guilt. As a result, they have an awkward workaround -- the effect of inferring the benefactor’s caring is allowed to influence decisions to accept or reject help either positively (to capture the contribution of gratitude) or negatively (to capture the contribution of guilt). A more straightforward and conceptually clear model would have two separate terms for these distinct emotions -- even if patterns of game play does not allow for separate estimation of these terms.

(1a) A better word for obligation?

Relatedly, the authors describe three main emotional responses to receiving help. Guilt and gratitude, described above, and “obligation”. Of course, the actual materials were in Chinese, so I am not sure what word exactly the participants saw. But the emotional experience the authors describe -- in Fig s1: “a negative feeling of pressure to repay caused by the other’s ulterior intentions” -- seems to me a better fit for “resentment”. “Obligation” is not commonly used as an emotion word in English, and doesn’t have the negative valence that their theory seems to imply - e.g. as the strongest motive for rejecting help. Indeed, feeling “obliged to reciprocate” does not predict rejecting help; whereas feeling “resentment” is a negative emotion and could lead to both rejecting help and reciprocating help, analogous to the argument that they make for “guilt”.

(1c) What is indebtedness?

Indebtedness does not appear in Fig 1, the theoretical model. The authors might think this is a fourth distinct emotional response (as suggested in Fig 2B); or is a cognitive appraisal of a factual situation (“I owe \$4”); or is a motivation to repay a perceived debt (suggested in Fig 4A and most of the text). Each of these is a slightly different view. Their formal model is closest to the third of these, because indebtedness is the parameter that is minimized to define the optimal repayment.

In summary, redescribing their conceptual model for better clarity: I think their conceptual model should actually be described like this:

receiving help can cause people to feel at least three different emotions, gratitude, guilt, and/or resentment ("obligation");

wanting to avoid feeling (the negative emotions) guilty or resentful can lead to rejecting help altogether;

if help is received, then gratitude, guilt, resentment, or a combination of them, can lead to a motive to repay or reciprocate the help, namely, indebtedness.

Their formal model is simpler than this conceptual model in various ways. For example, in their formal model, the degree to which participants attribute strategic (i.e. second-order belief) versus caring (i.e. perceived care) motivations is constrained to trade off linearly (governed by one parameter that varies across individuals, κ). Thus, the amount of guilt versus resentment a participant experiences is defined as anticorrelated. This over-simplification fits the two first principle components of individual differences in participants in the game, but seems unlikely in real life. That's fine; the authors can just explicitly acknowledge how the model is simplified for tractability.

(2) Distinguish experimental conditions from participant inferences

In the interactive game, benefactors paid money to relieve pain (from later electric shocks) for the participants (from the benefactors' perspective: anonymous beneficiaries). To influence participants' inferences of the benefactors' intentions, there were two conditions: when the participants could versus could not "reciprocate" by returning money to the benefactor. The authors argue that when the participant could not reciprocate, donations must be perceived as 'altruistic', whereas when the participant could reciprocate, donations might be perceived as 'strategic' (to elicit money in return); the authors name the conditions according to these predicted inferences.

However, as their own data show (Fig 3), participants' inferences are mixed. When returning money was possible, participants attributed a mixture of motives to benefactors -- both altruistic care and strategic coercion -- and actually felt more gratitude than obligation or guilt, on average (3a). And on the other hand, even when returning money was impossible, participants attributed non-zero expectations of reciprocation to the benefactor (3a), which increased with the benefactor's cost (3b).

All of this is perfectly consistent with the authors' theory -- but it is very confusing for a reader. So: please use different language to describe the experimental manipulation (e.g. \$ Return Allowed; \$ Return Impossible) and the resulting inferred intentions of the benefactor (e.g. Care, Coercion).

(3) Study 2 and the formal model

In study 2, the interpersonal game, the benefactor can pay money to reduce the electric shocks that will be received by the participant. There are two key manipulations: either the participant will be allowed to repay the money, or the participant will not be allowed to repay it; and either the participant is allowed to reject the help, or the participant is not allowed to reject it. The game has three outcome variables:

Self-reported emotions: how much gratitude, guilt, and "obligation", as well as indebtedness, the participant reported feeling on each trial.

Accept / reject decisions.

Repayment amounts.

Overall, I think the argument of this study is that all three of these outcomes can only be understood by taking into account participants' inference about the intent of the benefactor: whether the benefactor expected to be repaid, or genuinely cared about the participant's pain.

What sometimes this confusing is that the study uses three correlated but different proxies for

participants' inferences about benefactor intentions:

experimental condition: the benefactor's intentions are caring in the "no-repayment-possible" condition and strictly about repayment the "repayment-possible" condition.

self-report: participants were asked to explicitly report their inferences of benefactor expectations and caring.

expressions in a formal model. The benefactor's expectation of return is the money donated on the "repayment-possible" trials, and the perceived care is the money donated on "no-repayment-possible" trials; the formal model also includes some perceived care on the "repayment-possible" trials, by allowing that (variably across individuals) participants see donation on these trials as reflecting a mix of expectation of repayment and genuine care; this individual difference is captured by a free parameter κ , that varies across individuals, and is fit separately to the accept/reject and repayment decisions.

Overall, the hypothesis that the intent of the benefactor influences the three outcomes (emotions, rejections, repayments) is tested with respect to some of the proxies for inferred intent, but not in all or the most rigorous possible ways. So, the argument of the paper could be clarified, by making all of this structure more accessible to a reader (this took days to figure out); and it could be improved by doing stronger versions of the tests.

(3a) Validating the conceptual model in the self-reported emotions

In Fig 3a, the authors show that self reported emotions (outcome I) depend on experimental condition (proxy A) (and dependence on benefactor cost is shown in Fig S2). That is good!

But it would also be useful to test the conceptual model using the continuous data available in self-report (proxy B). Self-reports inference of benefactor intentions should predict, on a trial by trial basis, the reported level of gratitude, guilt, and 'obligation' on each trial. Probably the resulting correlations will be much stronger than the effects of condition shown here, because (according to the authors' own theory) there is lots of variability within conditions. (Note: in this analysis, make sure to carefully distinguish between variability across trials within participants, and variability across participants -- those two sources of variability are not distinguished in Fig 3 currently). Also, I recommend you replace the principle components analysis with the mediation models in Fig S3; or maybe a modified mediation model, combining across studies, and more fully testing the conceptual causal hypothesis: Experimental design -> two inferred intentions -> three emotional responses -> indebtedness.

(3b) Correlation of inferred intentions in self-report versus the formal model

As one measure of the validity of the formal model, the authors say that the terms for the inferred benefactor intentions in the formal model (proxy C) correlate with participants' self report of benefactor intentions (proxy B). Although these correlations look very high, this is actually a weak test of the model, because the shared variance is determined by the experimental design.

For example: the expected repayment ("second order belief") in the formal is the amount the benefactor paid on trials when repayment is possible. Participants reports of expected repayment are correlated with this model -- participants don't report benefactors expecting to be repaid more than they paid (e.g. paid 1pt expects to be repaid 20pts), and on average report lower expectations for repayment when repayment is impossible (though there is tons of variability here, as noted above). A stronger and more conservative test of the model would be to show that it correctly captures differences between participants, in inferred intentions, after accounting for the experimental design. There is a lot of variability in participant reported benefactor intentions within each trial type (figure S4A). The formal model of expected repayment does not -- by definition -- capture any of this structure. On the other hand, the formal model of perceived care is fit to participant behaviour using a free parameter (κ); so maybe the formal model's quantity for perceived care would correlate with self-reported perceived care, within trial types but across participants? That would be a better test of convergence between the model and the self-reported inferences.

(3c) Validating individual differences, across behaviours

The authors measure individual differences in three separate outcome variables, in the same individuals. It is not strictly necessary, but would be reassuring, to see that the estimates of individuals converge across these variables. For example, that a single person's kappa (which is related to estimates of caring motivations even in the presence of strategic incentives) is correlated, when modeled in accept/reject decisions, versus when modeled in reciprocity decisions.

(4) What do the neural data add?

First analysis shows that training of 4/5ths of the data (fit to estimates of expectation of return and perceived care), can modestly but significantly decode these values in the left out fifth of the data, within participants. One big contribution to these trialwise estimates is the trial type: whether it is a return possible or a return impossible type. These two conditions lead to big differences in the mean of estimates of expectation of return (high when return is possible) and perceived care (high when return is impossible). It is thus possible that what is being decoded, and cross-validated, is just this difference between conditions, rather than any more specific underlying representation. Maybe just the observation that these conditions can be decoded from the neural activity at the outcome phase is nevertheless an argument in favour of the authors' conceptual model.

The second model showed that these neural patterns could predict reciprocity behaviour -- here again, it would be nice to know whether these predictions are better than similar predictions made just in terms of the two conditions. Does the formal expression for expectation and perceived care fit the data (and predict behaviour) better than the qualitative condition difference?

For the third analysis, they look at individual differences. In principle, this ought to be the key analysis for testing the fit of their model. For example, if their model is capturing individual level variance then fits to the data would be better if using the model as fit to the participant's own responses, than the same structural form of the model but fit to a different participant's data.

In fact the analysis that they do present of individual differences is somewhat confusing. They show that the similarity between a participant's neural pattern predicted by feeling "obligation" (resentment) and the neural pattern predicted by behavioural reciprocity, predicts the level of the participant's reciprocity. This is a somewhat odd analysis, especially given that in these authors' conceptual model, reciprocity is likely to be caused by any combination of gratitude, guilt and obligation/resentment. So why should only the pattern similarity of between obligation and reciprocity predict behaviour? Indeed, it seems plausible that this was not the only similarity analysis conducted by these authors, and thus reflects some experimenter degrees of freedom in the choice of analyses presented in the final paper, not entirely accounted for in the calculation of frequentist statistics. (If not, and these analyses were preregistered or defined a priori, the authors should be more clear about this).

Again, to be clear, I do agree that the authors' neural data are consistent with their hypothesis that in order to understand the pattern of neural responses that occur upon receiving a generous offer of help, it is necessary to consider the participant/ recipient's inference of the beneficiary's intention and that this is the central contribution of the neural data. I just believe that the evidential basis of this claim, and it's strength could be communicated more effectively.

Minor point: How does the efficiency manipulation affect emotions?

They measured feelings of obligation, guilt and gratitude after benefactors gave different amounts, with different efficiency. How does efficiency affect these emotions? Do recipients feel less guilt when a small gift had a larger effect? And more guilt when a larger gift had a smaller effect? Do donations of low efficiency create more gratitude? There is some hint of these effects on self-reported emotions Fig S3 -- and perhaps they need to stay there, because of how complicated this paper is already. But, well, they are interesting questions.

Surveys. This is not "experience sampling" (which implies repeated measurements). It's just a survey.

-- Rebecca Saxe

Response to reviews

We thank the reviewers for their constructive suggestions, which we believed have helped us to greatly improve the clarity of the manuscript. Below we detail our responses to each of the reviewers' points. The revised parts are highlighted in blue in the Manuscript File. The reference numbers are consistent with those in the main text. References not included in the main text are listed after the corresponding question.

Reviewer #1 (Remarks to the Author):

This paper develops a theory of indebtedness, that is modeled using a value function over actions in response to a favor. The value function possesses self and other components. A straightforward self-benefit piece and a social cost comprised of a communal term (guilt) and an obligation term.

The paper's conclusions and appeals derives from three sources:

- (1) responses to an online questionnaire,*
- (2) behavior in a task,*
- (3) computational modeling of (2),*
- (3) neuroimaging to provide evidence for their proposed theory.*

General Impression

This paper enlists a voluminous amount of data and analysis in support of the proposed theory/model of indebtedness. The data are compelling but there are weaknesses in the exposition, most importantly in the description of the task that make it difficult to assess the paper. TO be clear, I found it difficult to understand in detail exactly what had been done or exactly how particular analyses were carried out. With that said, I found the paper rich and the ideas presented interesting and this paper can be of wide interest to the social neuroscience, decision neuroscience, and neuroimaging communities. I particularly liked the results presented in figure 4.

I have outlined specific complaints below, but the 'semantic envelop' of this review is that I really needed more clarity for each of the points.

Response: We thank the reviewer for their careful observations and thoughtful comments on our paper, and we apologize for not clearly articulating the details of our experimental design and modeling methods. This was likely due to trying to balance methodological details and readability. We have now attempted to clarify these details throughout the manuscript beyond the specific points highlighted by the

reviewer. We have also tried to remove and simplify some of the analyses to streamline the manuscript and improve the overall clarity of the main points. In addition, to enable the readers to understand our tasks and analyses more clearly, we have substantially edited and condensed our methods and supplemental materials to have the majority of the experimental details located in the *Materials and Methods* section in the manuscript (see *Materials and Methods* on pp. 31 - 53).

Points.

Line 99. Equation 1. The Greek letters in the equation do not match those in the text.

Response: We have replaced Θ and Φ in the main text and Fig. 4 (the Fig. 5 now) with θ and ϕ to make them consistent with the Greek letters in equations.

Line 122. "predictions" seems to be too strong a term. it would be good to enumerate the predictions first, for example.

Response: We thank the reviewer for this suggestion. In this paper, we aimed to validate the predictions of both our theoretical model (paragraph 3 on pp. 4 - 5) and computational model (paragraph 4 on p. 5 and Eq. 1) across multiple studies. As suggested, we have now revised the paragraph 5 on pp. 5 - 6 to clarify the main hypotheses we aimed to test and the corresponding approaches we used in each of the three studies:

“We validate our conceptual and computational models of indebtedness across a series of studies. In Study 1, we explore lay intuitions of indebtedness using a large-scale online questionnaire to test the hypothesis that indebtedness is a mixed feeling comprised of both guilt and obligation. In Study 2, we evaluate how different components of indebtedness are generated and influence behaviors in an interpersonal game, in which benefactors (co-players) choose to spend some amount of their initial endowments to reduce the amount of pain experienced by the participants. We test the hypothesis that guilt and obligation arise from appraisals of the benefactor's intentions, and specifically that appraisals of altruistic intentions produce guilt while appraisals of strategic intentions lead to obligation. We then evaluate how well our computational model (Eq. 1) captures these appraisal/feeling components and can predict participants' decisions to reciprocate help in the interactive game. In Study 3, we test the hypothesis that the two components of indebtedness are associated with unique brain representations using functional magnetic resonance imaging (fMRI). We create a neural utility model of indebtedness by applying our computational model directly to multivariate brain patterns to demonstrate that neural signals reflect the tradeoff

between these feelings and can be used to predict participants' trial-to-trial reciprocity behavior. ”

Line 194 (And figure 2C). First, it seems figure 2C deserves its own figure. Second, there are many details of the task that are omitted: 1) What exactly does the partner know about the partners? Is there actually another person for each trial? 2) How many trials are there? 3) Does the participant actually get shocked (is that the pain?)? Is the endowment of the partner (γ_{A} ?) always 20? Is the endowment of the participant (γ_{B} ?) always 25?

Response: First, as suggested, we have now made Fig. 2C a separate figure (Fig. 3 now). We have also added an overview of the entire experiment in this figure to illustrate the general procedures for Study 2.

Second, we apologize for not clearly articulating the details of our task. We have added more details of the experimental procedures (pp. 9 - 10) and computational model (pp. 14 - 17) to the main text, moved our methods in the *Supplementary Materials* to the *Materials and Methods* section (pp. 33 - 38) and have attempted to streamline the details to improve readability.

We have also provided detailed response to each of the reviewer's specific questions.

(To be noted, as suggested by *Reviewer 3's Query 2*, to show the actual condition manipulation of our task rather than the inferred conditions of intentions, we have now replaced the original condition names of “Altruistic condition” and “Strategic condition” with “Repayment impossible condition” and “Repayment possible condition” in the manuscript.)

1) What exactly does the participant know about the partners? Is there actually another person for each trial?

We have added the following paragraph to the main text (pp. 9 - 10):

“In Study 2a (N = 51), participants were randomly paired with a different anonymous same-sex co-player (benefactor) in each trial and were instructed that they would receive 20 seconds of pain stimulation in the form of a burst of medium intensity electrical shocks. The participant was instructed that each benefactor was: (a) informed of the participant's situation, (b) endowed with 20 yuan (~ \$3.1 USD), and (c) could spend any amount of this endowment to help the participant reduce the duration of pain (i.e., benefactor's cost). The more the benefactor spent, the shorter the duration of the participant's pain experience. After seeing how much money the benefactor chose to spend, the participant reported how much they believed this benefactor expected them to reciprocate for their help (i.e.,

second-order belief of the benefactor's expectation for repayment). In half of the trials, the participant had to accept the benefactor's help; in the other half, the participant could freely decide whether to accept or reject the benefactor's help. Finally, at the end of each trial, the participant decided how much of their own 25 yuan endowment (~ \$3.8 USD) he/she wanted to allocate to the benefactor as reciprocity for their help. We manipulated the participant's beliefs about the benefactor's intentions by providing additional information regarding the benefactor's expectations of reciprocation. Each participant was instructed that before making decisions, some benefactors knew that the participant would be endowed with 25 yuan and could decide whether to allocate some endowments to them as reciprocity (i.e., ***Repayment possible condition***), whereas the other benefactors were informed that the participant had no chance to reciprocate after receiving help (i.e., ***Repayment impossible condition***). In fact, participants could reciprocate in both conditions during the task. After the task, participants recalled how much they believed the benefactor cared for them, as well as their feelings of indebtedness, obligation, guilt, and gratitude based on the help they received for each trial. At the end of the experiment, five trials of the interactive task were randomly selected to be realized and participants received the average number of shocks and money based on their decisions. Unbeknownst to participants, benefactors' decisions were pre-determined by a computer program (Table S2)."

Please see also the detailed procedures on pp. 33 - 38.

2) *How many trials are there?*

There were 48, 56 and 54 trials for Study 2a, Study 2b and Study 3, respectively. Please see methods on pp. 35-36, p. 37 and Table S2 for the detailed information about the experimental design.

3) *Does the participant actually get shocked (is that the pain?)?*

At the end of the experiment, five trials were randomly selected to be realized. The participant received the average pain stimulation in these five trials. The participant's final payoff was the average amount of endowment the participant left for him/herself across the chosen trials. The participant was instructed that the final payoff of each co-player was the amount of endowment the co-player left plus the amount of endowment the participant allocated to him/her. Participants were informed of this arrangement before the experiment began (p. 36).

4) *Is the endowment of the partner (γ_A) always 20? Is the endowment of the participant (γ_B) always 25?*

Yes, the endowment of the co-player (γ_A) was always 20 yuan, and the endowment of the participant (γ_B) in each trial was always 25 yuan. In each trial, the endowment of

the participant was always larger than the endowment of the co-player. This experimental manipulation was designed to make the participant believe that the co-player expected repayments in Repayment possible condition, since these co-players knew that the participant had the ability to reciprocate and was endowed with more money than the co-player (p. 35).

Line 294. it is not clear that listing the parameters here helps since the equations are relegated to the methods.

Response: We agree with the Reviewer and have now moved the model equations of these two terms and the corresponding explanations into the main text (pp. 14 - 17) to improve readability.

Figure 4C. Why not show this separately for the two conditions.

Response: As suggested, we now show the actual behaviors and the model predictions of reciprocity separately for Repayment possible and Repayment impossible conditions (p. 17, Fig. 5C).

Line 359. The definition of the linear contrast is unclear.

Response: Our model predicted that the participants' sense of obligation (indicated by E_B) was near zero in Repayment impossible condition and increased linearly in Repayment possible condition as the benefactor's cost increased. Therefore, we formulated the contrast as increasing linearly in the Repayment possible condition (i.e., [+1, +2, +3]), but flat in the Repayment impossible condition with a sufficient magnitude to ensure that the images were equally balanced in the contrast (i.e., -6). An alternative contrast would have been to keep the Repayment impossible condition at 0 and only model the linear contrast in the Repayment possible condition (i.e., [-1, 0, 1]). However, this would specify that the Repayment impossible condition responses would be equivalent to the mean of the increasing Repayment possible condition, which does not fit our observations of increased obligation in the Repayment possible condition (Fig. 4A and Fig. S2B). We have used this type of piecewise contrast specification in previous related work⁵⁴. We have added the following description in the *Materials and Methods* on p. 49 - 50:

“For GLM3, because our computational model's representation of second order beliefs E_B had a very non-normal distribution, we constructed a piecewise linear contrast. This entailed creating four separate regressors modeling different parts of the function during the Outcome phase: (1) Repayment impossible, (2) Repayment possible and low benefactor's cost (i.e., 4, 6, or 8), (3) Repayment possible and

medium benefactor's cost (i.e., 10, 12, or 14), (4) Repayment possible and high benefactor's cost (i.e., 16, 18, or 20). Subsequently, for each participant, we constructed a contrast vector of $c = [-6, 1, 2, 3]$. This piecewise linear contrast ensures that brain responses to the Repayment impossible trials are lower than all of the Repayment possible trials. We have successfully used this approach in previous work modeling guilt using similar psychological game theoretic utility models⁵⁴.

Line 674. "percentiles"? I believe this refers to Eq. 10, which is not well-explained. I think the first instance of $D_{\{B\}}(t)$ refers to the actual choice whereas the max is over the possible choices ($D_{\{B\}}(t)$). The error term that is squared seems to be the squared percentage error.

Response: We thank the Reviewer for pointing this out. This error has been corrected now on pp. 45:

“We estimated the model parameters for Eq. 1 by minimizing the sum of squared error of the percentages that the model's behavioral predictions deviate actual behaviors over all the trials that participants had to accept help using Matlab's `fmincon` routine. More formally, for each participant we minimized the following objective function:

$$SSE = \sum_{t=1}^n \left(\frac{D_B(t) - \max(U(D_B(t)))}{\gamma_B} * 100 \right)^2 \quad \text{Eq. 9}$$

with t indicating trial number. To avoid ending the fitting procedure at a local minimum, the model-fitting algorithm was initialized at 1000 random points in theta-phi-kappa parameter space for each participant.”

Eq. 12. What is the link between the difference between the utility of ACCEPT and that of REJECT and P? (I assume $D_{\{B\}}(t)$ is the participant's choice ACCEPT or REJECT.

Response: We thank the Reviewer for their careful read of our manuscript. A softmax function linking the $U(\text{Accept})$ with $P(\text{Accept})$ was accidentally omitted here. These functions have now been added on p. 47:

“We computed the probability of the decision of whether to accept or reject help using a softmax specification with inverse temperature parameter λ , which ranges from [0,1]. In each trial, the probability of the participant choosing to accept help is given by

$$P(\text{Accept}) = \frac{e^{U(\text{Accept})/\lambda}}{e^{U(\text{Accept})/\lambda} + e^{U(\text{Reject})/\lambda}} \quad \text{Eq. 12}$$

We then conducted maximum likelihood estimation at the individual level by minimizing the negative log likelihood of the decision that the participant made ($D_B = \text{Accept or Reject}$) over each trial t with 1000 different starting values:

$$LLE = - \sum_{t=1}^n \log(P(D_B(t))) \quad \text{Eq. 13''}$$

In general it would be useful to see a summary of the parameter estimates for the models (e.g. distributions of the parameters).

Response: We thank the reviewer for this suggestion, and now include descriptive statistics for all model parameters are presented in Table S11 and S12, and distributions of model parameters are now presented in Fig. S6. A sentence referring to these results is added on p. 17.

Reviewer #2 (Remarks to the Author):

The authors present the results of three studies that aim to test a model of indebtedness, reflecting the trade-off between guilt (or communal concern) and the sense of obligation. Guilt is defined as the appraisal of the benefactor's altruistic intentions. Obligation is defined as the appraisal of the benefactor's strategic intentions. In the first study, they use an online questionnaire to characterize the subjective experience of indebtedness in Chinese participants. Participants describe their feelings during every-day life encounters that are then coded by independent raters. The results suggest two main factors – guilty and sense for obligation. Study 2 is a behavioral experiment in which benefactors spend some amount of their initial endowment to reduce pain experienced by the participants. In one condition (called strategic condition), the benefactors ostensibly were informed that the participant can reciprocate the favor. In the other condition (called altruistic condition) the benefactors were informed that the participants cannot reciprocate the favor. Participants reported feeling indebted to benefactors who spent money to reduce their pain, in both conditions, but slightly more in the altruistic compared to the strategic condition. Study 3 used the same paradigm as Study 2 and fMRI to regress the model parameters (assumed to reflect communal concern and sense of obligations) against neural activations. Moreover, they analyzed the brain activation during receiving the favor, interpreted as reciprocity. The results showed a large number of brain regions, including frontal, parietal, temporal areas. Further analyses (metaanalytic decoding based on Neurosync) indicated that reciprocity-related activation was linked to attention and memory processes, communal concern was associated to the default network, and sense of obligation was associated with terms related to “Social”, “Theory of Mind” and “Memory”. Finally, the authors submitted the brain activations associated with communal concern and sense of obligations to their model and showed that they can predict reciprocity behavior. The authors conclude that they developed a neural utility model of indebtedness that captures the tradeoff between feelings of communal concern and obligation when deciding how much to reciprocate after receiving help from a benefactor.

The manuscript use rich data sets and elaborate analyses to tackle an interesting and relevant research question. The paper is well written, and a computational model that disentangles the intentions behind altruistic behavior would be a welcome contribution to the literature of the field. However, I have conceptual and methodological concerns that dampen my enthusiasm.

Major concerns:

1. One main concern is related to the vague definitions of the concepts. Indebtedness is divided into guilt/ communal concern and sense of obligation. The Introduction section implies that guilt/ communal concern is a feeling while sense of obligation is “strategic”. I think this distinction between “altruistic” vs “strategic/ self-serving” is not convincing, because guilt can also induce a strategy, namely the strategy to reduce

the negative guilt-related emotions or arousal. In this case, behavior elicited by guilt would not be altruistic, but self-serving. On the other hand, it is plausible to assume that a sense of obligation can raise communal concern, which, based on the framework of the authors, would be “altruistic”, rather than strategic and self-serving. In other parts of the manuscript (including the abstract) both constructs are labelled as feelings (e.g., “distinct feelings of guilt and obligation”; line 32), which makes the definition of the concepts even more confusing. Without a clear definition of the underlying concepts, it is hard to say what we learn from the proposed model.

Response: We thank the reviewer for raising this issue and agree that our terminology was imprecise and somewhat confusing. Unfortunately, this stems in part as a result of the concepts that we build on from previous work come from different domains and lack precise agreed upon definitions (e.g., emotion, neuroscience, economics, sociology, etc). We have now attempted to clarify our definitions of each concept, and also use more consistent language throughout the manuscript. We use feelings/emotions interchangeably and note the lack of consensus of these terms across disciplines. Our model attempts to break down these different aspects of indebtedness from the perspective of appraisal theory²⁸⁻³³. Emotions/feelings from an appraisal framework are adaptive responses that: (1) are elicited based on how an agent evaluates its situation (i.e., appraisals), and (2) function to motivate the agent's adaptive responses (i.e., behaviors). Under this appraisal-emotion-behavior framework, the current study aims to investigate how indebtedness arises and influences reciprocal behaviors.

Our model proposes that both guilt and obligation are feelings/emotions that the beneficiary may experience after receiving favors. Definitions of these two feelings were determined based on previous studies (p. 4). Both of these feelings contribute to indebtedness. However, the key element that determines whether the beneficiary feels guilt or obligation is how they appraise or evaluate the benefactor's intentions. We propose that the guilt component of indebtedness arises when the beneficiary believes the benefactor provided help with altruistic intentions, meaning that they expected nothing in return from the beneficiary. The obligation component of indebtedness, in contrast, arises when the beneficiary believes the benefactor has an ulterior motive to be repaid for providing the help (i.e., strategic intentions). We have now revised Fig. 1 to better illustrate our conceptual model.

We also may have inadvertently caused some confusion with the reviewer's understanding of our theoretical frame. Our model is not concerned with the strategic nature of the beneficiary, but rather, the beneficiary's *perception* of how strategic a benefactor appears to be. Benefactors that have no expectation of repayment are likely to be *perceived* as more altruistic and therefore engender more communal concern in a beneficiary. However, benefactors that expect repayment may instead appear to be *perceived* as more strategic and thus engender obligation in a beneficiary. We have

now changed the labels of our experimental manipulation to hopefully reduce some of this confusion (i.e., altruistic -> repayment impossible, & strategic -> repayment possible).

The central idea of our conceptual and computational models is that these feelings arise from different perceptions of the benefactor's intentions and serve as different motivations underlying the beneficiary's reciprocity behaviors. The reciprocity induced by guilt (as well as gratitude) reflects the beneficiary's communal concern for the benefactor^{44,45}, while the obligation-based reciprocity is likely driven by external pressures, such as social expectations and reciprocity norms^{48,49} (p. 4).

We model the likelihood of the benefactor taking different actions using expected utility theory, in which each action is the linear sum of the various costs and benefits associated with the outcome. We utilize the Psychological Game Theory framework³⁴⁻³⁶ to incorporate the psychological costs of these feelings of guilt or obligation into the utility functions associated with selecting each action. The costs and benefits of each action are solely considered from the perspective of a single agent and we assume that agents are self-serving in the sense that they are primarily concerned with selecting actions that maximize their own expected utility. However, it is important to note that this theoretical framework still allows considerations of others' outcomes to be modeled as a positive value. Thus, the contribution of this paper is specifying a new utility model that incorporates psychological appraisal theory into the costs and benefits of selecting an action (Eq. 1). This utility function provides an additional tool for modeling social preferences (e.g., inequity-aversion, guilt-aversion, intention-based reciprocity, etc). We hope that these ideas come across more clearly in the revised manuscript.

*2. As currently defined, it is unlikely that guilt/ communal concern and sense of obligation are two orthogonal concepts (see my first point). However, the assumption of orthogonality is an essential part of the proposed model ($\Phi * U_{communal} + (1 - \Phi) * U_{obligation}$). In Study 2, the variance inflation factor (VIF) was calculated and a PCA for dimensional reduction was performed, but in Studies 1 and 3 there are no analyses showing orthogonality of the concepts. In Studies 1 and 3, there were only marginal differences between the AIC of the model with self-reported guilt as only predictor and the model with both predictors (Table S1). This result indicates that model fit does not considerably improve by adding the second component (i.e., sense of obligation). In other, words, guilt might be sufficient to model the data. In light of these results, I am not convinced that guilt and sense of obligation are conceptually and/or statistically orthogonal.*

Response: We apologize for the confusion, but think it is important to make a clear distinction between our conceptual model, our computational model, and the statistics we use to evaluate the likelihood of these models given the data.

Fig. 1 outlines our conceptual model that there are two components of indebtedness - guilt and obligation, which are derived from appraisals about the benefactor's intentions and can differentially impact the beneficiary's reciprocity behaviors.

First, we use statistics (linear regression models) to evaluate evidence supporting our conceptual model in the self-reported data. In Study 1, we test our conceptual model by demonstrating that both ratings of guilt and obligation significantly contributed to the ratings of indebtedness, with acceptable level of multicollinearity between guilt and obligation ratings. To demonstrate that the parameter estimates for guilt and obligation are unlikely to be affected by multicollinearity, we examined the variance inflation factor (VIF). Results demonstrated an acceptable level of multicollinearity between guilt and obligation ratings (i.e., $VIF < 5$; Study 1: $VIF = 1.22$; Study 2: $VIF = 1.44$; Study 3: $VIF = 1.98$). This information is now added in *Materials and Methods* on p. 38, p. 41 and Table S1. More importantly, both self-reported guilt and obligation ratings *independently and significantly* contributed to increased indebtedness ratings (p. 7 and Table S1). We have now removed the nested model comparisons from the main text to simplify the exposition of the results, as they were redundant with the full regression models.

In Studies 2 and 3, we use an interactive task and statistical models to provide additional support for our conceptual model. Our experimental design successfully manipulated participant's self-reported appraisals and their corresponding feelings (Fig. 4). We completely agree with the reviewer, that participants are likely not making these ratings as if the constructs are orthogonal. However, our PCA analysis (now factor analysis based on the reviewer's minor points Query 4) demonstrates that self-reported guilt and obligation and their corresponding appraisals do indeed map onto two orthogonal factors (i.e., along each axis of the two factors depicted in Fig. 4D). Importantly, self-reported feelings of indebtedness covary with each of these independent factors consistent with our operationalization of indebtedness as a mixture of these feelings in our computational model. In addition, we test a path model to demonstrate that appraisals mediate changes in self-reported feelings following our causal manipulation of the benefactor's intentions (Fig. 4E).

Second, we performed a more rigorous test of our conceptual model by constructing a computational model of the proposed psychological processes (Eq. 1). Our computational model defines the participants' reciprocity behaviors as driven by mixtures of independent feelings of communal concern (guilt and gratitude) and obligation. These feelings arise as functions of the appraisals of the benefactor's intentions. We use statistical models to evaluate how well the computational model can explain participant's decision behavior in Studies 2 and 3. It is possible that our mathematical operationalization does not reflect the psychological processes underlying the decisions. To rule out this possibility, we evaluate how well different

aspects of our computational model map onto participant's self-reported appraisals and feelings as an additional validation (pp. 15 - 16).

As the reviewer notes, our computational model includes a mixture of independent constructs of communal concern and obligation, which have a nonlinear relationship depending on κ , and uncorrelated in the current dataset with $\kappa = 0.32 \pm 0.01$, $\beta = -0.03 \pm 0.03$, $t = -1.23$, $p = 0.222$ (p. 15 and Fig. S4A). More importantly, covariance between these model terms implies that there might be multiple configurations of parameters that can produce the same predicted behavior. This means that, in practice, the more that these constructs covary, the less identifiable our parameters will become. We demonstrate in our parameter recovery tests that we are able to accurately recover parameters that we simulate for reciprocity behavior ($r = 0.94 \pm 0.07$, $p < 0.001$; Table S9), confirming that this is not a problem for our reciprocity model. Furthermore, we found that our model significantly outperforms alternative models that only include a term for either communal concern or obligation (p. 16 and *Materials and Methods* on p. 47).

Taken together, these results consistently suggest that both subjective experiences of guilt and obligation significantly contributed to indebtedness, and influenced the beneficiary's reciprocal behaviors, with an acceptable level of collinearity. These findings support our conceptual model that the two components of indebtedness arise from different appraisals and motivate reciprocity from different concerns.

We recognize that this paper is dense and includes a lot of data, models, and statistical tests, and have now made substantial revisions on the ways to report our *Results* and *Materials and Methods* to better express the logic of our analyses. We hope that the reviewer finds our revised manuscript more clear based on his/her feedback.

3. In Studies 2 and 3, the validity of the two components is inferred from self-reports that explicitly asked participants to indicate their feeling of guilt and obligations (e.g., "How much guilt do you feel for this Decider's decision?"; "How much afraid/pressure do you feel for the decider's expectation for repayment?"). These explicit questions might impose the demand to classify emotions in the two respective categories. It would be desirable to determine the validity (and assumed orthogonality) of the two components with more implicit measures.

Response: We appreciate the reviewer's concerns and are deeply sympathetic to the general challenge that self-report remains the gold-standard measure for studying emotions despite completely relying on participants' ability to introspect and potential demand characteristics that may arise from self-report^{81,82}. We note that there is currently no objective method (e.g., face expressions, physiology, neural responses) to measure internal emotional experience states in general or social emotions more

specifically. We may be misunderstanding the reviewer’s use of the word “implicit”, but to our knowledge there is no implicit method of measuring social emotions.

We hope the reviewer can appreciate the important role of this work as a substantial step forward towards the ultimate goal of developing more objective ways to measure social emotions. We have created both a conceptual and computational model of a specific type of social emotion and a task to measure indebtedness using *behavior*. We think it is important to emphasize that our computational model *does not require self-report*. We validate our computational model using behavior and also self-report in Study 2. In Study 3, we demonstrate that it is possible to use this model to objectively measure these constructs using brain activity with our neural utility model. We believe that this single paper provides a substantial contribution towards this goal, which we hope will inspire future work (p. 25).

Here, the questions for self-reported ratings on guilt⁴⁰⁻⁴³ and obligation^{13,14,21,46,47} were designed according to the operational definitions built by previous research (pp. 31 - 32, and p. 36). The reviewer may also wonder if participants may feel other types of feelings related to indebtedness in Studies 2 and 3. This was precisely the goal of Study 1, to be very open ended and allow participants to respond using self-report without even the constraints of rating scales. We allowed participants to report their definitions of indebtedness without restriction and conducted topic modeling to explore lay intuitions of indebtedness⁵³. This data-driven approach provides evidence supporting the distinction between obligation and guilt proposed in our conceptual model. Topic modeling is one of many tools used to study natural language processing, but has not yet been widely used in the study of social emotions. It uses latent Dirichlet allocation to find latent factors of semantic concepts based on the co-occurrence of words in participant’s verbal descriptions without constraining participants’ responses using rating scales, which currently dominates emotion research¹⁰⁴. Using this method, in Study 1, emotional words were extracted from the 100 words with the highest weight/frequency in the definitions of indebtedness based on the annotation by an independent sample of participants (N = 80). Note, words beyond these 100 had TF-IDF weights < 0.01 (Fig. S1B), indicating that the words included in the current analysis explained vast majority of variance in the definition of indebtedness (pp. 39 - 40).

We hope our work will provide new avenues to objectively or even implicitly measure social emotions in future work. Previous opinion articles have suggested that formalizing emotions using computational models is critical to advancing theory, characterizing their impact on behavior, and identifying associated neural and physiological substrates^{39,83,84}. However, the application of computational modeling to the study of social emotions is a relatively new enterprise^{39,54,85,86}. Our model demonstrates how emotion appraisal theory²⁸⁻³³ can be integrated with psychological game theory^{36,37} to predict behavior³⁹. We model emotions as arising from appraisals about perceived care and beliefs about the beneficiary’s expectations, which both

ultimately increase the likelihood of the benefactor selecting actions to reciprocate the favor. This model contributes to a growing family of game theoretic models of social emotions such as guilt^{34,54}, gratitude⁸⁷, and anger^{88,89}, and can be used to infer feelings in the absence of self-report providing new avenues for investigating other social emotions (p. 25). We have updated text throughout the manuscript to make these points clearer. Please see the page numbers above for each point.

4. The imaging results show activations in a large number of brain regions. The statistical threshold is set to voxel-level $P < 0.001$ (uncorrected) combined with cluster-level threshold $P < 0.05$ [family-wise error (FWE) corrected]. Given the lack of a priori neural hypotheses, I strongly recommend to present only FWE voxel-level results and to use only these FWE voxel-level corrected results to predict reciprocity behavior.

Response: We agree with the reviewer and appreciate the opportunity to clarify that we only report results that survive family-wise error correction (FWE). We use the standard cluster correction approach available in most fMRI packages (e.g., SPM, & FSL) that attempts to control FWE using Gaussian Random Field Theory. This approach attempts to estimate the number of independent spatial resels or resolution elements in the data necessary to control for FWE. This calculation requires defining an initial threshold to determine the Euler Characteristic of the data. It has been demonstrated that an initial threshold of $p < 0.001$ does a reasonable job of controlling for false positives at 5% using this approach⁷¹. We have edited the text to clarify that we only report FWE corrected results (p. 20, Fig. 6 caption and p. 50).

Beyond the more traditional univariate neuroimaging analyses, we provide a much stronger test of our conceptual model with our neural utility model that does not require controlling for multiple comparisons. In this approach, we train whole-brain models capable of predicting guilt and obligation. We then construct a neural utility model by combining these predictions and demonstrate that we can predict participant's decision behavior directly from brain activity almost as well as computational model fit directly to behavior. Whole-brain multivariate models differ from univariate models in that they do not require controlling for multiple comparisons as we only estimate a single model, rather than a separate one for each voxel in univariate analyses. We intentionally did not perform any feature selection in this process as we did not want to insert any bias in the results. Applying a spatial feature selection (or mask) based on the univariate analyses as suggested by the reviewer would actually bias our results without carefully thinking through cross-validation. This is one of the examples discussed by Niko Kriegeskorte in his well-cited paper describing double-dipping in multivariate analyses (Kriegeskorte et al., 2009). We have recently written a paper outlining the pros/cons of different spatial feature selection strategies in multivariate analyses and argue that whole-brain

approaches as we have employed here are recommended when the underlying neural representation is hypothesized to be spatially diffuse (Jolly & Chang, 2021).

Kriegeskorte, N., Simmons, W., Bellgowan, P. *et al.* Circular analysis in systems neuroscience: the dangers of double dipping. *Nat Neurosci* **12**, 535–540 (2009). <https://doi.org/10.1038/nn.2303>

Jolly, E., & Chang, L. J. (2021). Multivariate spatial feature selection in fMRI. *Social Cognitive and Affective Neuroscience*. doi:10.1093/scan/nsab010

Minor points

1. *In Studies 2 and 3, the partner was anonymous. Thus, reputation effects are minimized which might account for the stronger effects in the altruistic condition compared to the strategic condition. It would be good to discuss this point in the Discussion section.*

Response: Having prior information about a player’s behavior could indeed influence perceptions of the player’s reputation and could influence appraisals about the player’s intentions, further complicating our experimental manipulation. This is precisely why we designed the experiment to minimize potential reputational effects for both players. Our experimental manipulation of the participants’ beliefs about whether the co-player knows they have an opportunity to repay or not allows us to manipulate the participant’s beliefs about the co-player’s intentions behind the help. Knowing information about the co-player’s reputation would make this manipulation more complicated. We have added corresponding discussion on pp. 27-28. We connect this work to studies of relationships in sociology and are currently considering how we could explicitly study the impact of different types of relationships on indebtedness in future studies.

“Second, our model provides a framework to better understand the role of relationships and contexts in generating feelings of indebtedness within a single individual. Different types of relationships (see Clark and Mills’s theory of communal and exchange relationships ^{4,5}, and Alan Fiske’s Relational Models Theory ⁹⁴) have been theorized to emphasize different goals and social norms which can impact social emotions ^{95,96}. For example, communal relationships prioritize the greater good of the community and are more conducive to altruistic sharing, which can be signaled by altruistic favors ³⁻⁵. In contrast, exchange relationships are more transactional in nature ^{2,4,5,10-12} and emphasize maintaining equity in the relationship, which can be signaled by strategic favors ⁹⁴. Our model proposes that perceptions of the benefactor’s intentions directly impact the feelings experienced by the beneficiary (e.g., guilt & obligation). Although we deliberately attempted to minimize aspects of the relationship between the benefactor and

beneficiary by making players anonymous to control for reputational effects, future work might experimentally manipulate these relationships to directly test the hypothesis that relationship types differentially moderate the responses of gratitude and subcomponents of indebtedness.”

2. *The survey method in Study 1 is described sometimes as “large-scale experience sampling” and sometimes as an “online questionnaire”. If a “large-scale experience sampling” is reported, it is not intuitively clear that this is a questionnaire and vice versa. This point should be clarified and described consistently.*

Response: We apologize for the inconsistency and have now replaced all instances of “experience sampling” with “online questionnaire”.

3. *Figure 2: The description of the paradigm (Fig. 2C) should be presented first (i.e., as panel A).*

Response: Since Fig. 2A-B and Fig. 2C belong to different studies (i.e., Study 1 and Study 2), we have now moved Fig. 2C to a separate figure (Fig. 3 now). This order of figures is consistent with the order of results in the main text.

4. *PCA is a data reduction method in which the “factors” (main components) are formed as linear combinations of the items. Measurement errors are not subtracted (based on the assumption that all items are free of measurement errors, which tends to be incorrect for most empirical data). As a result, the main components may actually contain error variance (e.g. Fabrigar, L. R., Wegener, D. T., MacCallum, R. C., & Strahan, E. J. (1999). Evaluating the use of exploratory factor analysis in psychological research. *Psychological Methods*, 4(3), 272–299. <https://doi.org/10.1037/1082-989X.4.3.272>). To overcome this problem, the authors might consider a confirmatory factor analysis to test the assumption of two components derived from Study 1.*

Response: We thank the reviewer for this suggestion. We have now replaced our principal component analysis with a factor analysis, and the results are quite similar (p. 13). We would like to note that this analysis uses data from Study 2. Study 1 did not use PCA, but rather Topic Modeling based on the word frequency in the responses to the open-ended responses from our online questionnaire. However, we appreciate the reviewer’s suggestion to consider also using confirmatory factor analysis. We now train an exploratory factor analysis (EFA) on the data from Study 2 and test the model’s generalizability to Study 3 using confirmatory factor analysis (CFA). This is conceptually similar to the cross-validation approach we used in some of our other modeling analyses. The Kaiser-Meyer-Olkin (KMO) Measure of Sampling Adequacy

¹⁰⁹ and Bartlett's test of sphericity ¹¹⁰ showed that the current data sets in Studies 2 and 3 were adequately sampled and met the criteria for factor analysis (Study 2: KMO value = 0.76, Bartlett's test $\chi^2 = 8801.85$, $df = 15$, $p < 0.001$; Study 3: KMO value = 0.77, Bartlett's test $\chi^2 = 2970.53$, $df = 15$, $p < 0.001$) (p. 42). Results showed that the fitness of this two-factor model is appropriate given conventional cutoffs (RMSEA = 0.079, SRMR = 0.019, CFI = 0.986, TLI = 0.970; Fig. S7G; p. 18). Taken together with results of mediation analyses, our results suggested that the guilt component of indebtedness, arises from the belief that the benefactor acts from altruistic intentions, while the obligation component of indebtedness arises when the benefactor's intentions are perceived to be strategic. Corresponding results and methods are updated on p. 13, p.18 and p. 42.

5. Discussion (line 493): *“One of the most notable contributions of this work is the development and validation of a computational model of indebtedness. The majority of research on emotions relies on self-reported subjective feelings, which has a number of limitations, such as its dependence on participants’ ability to introspect.” Given that the current study also used self reports to validate the two components (see Major concerns, point 3 above), this statement should be deleted or toned down accordingly.*

Response: We respectfully disagree with the reviewer and believe it is important to emphasize the notable accomplishment of this work in developing and validating a computational model of indebtedness. We think it is important to emphasize that the implementation of our computational model *does not require self-report*. Self-report was merely one of many types of data that were used to validate the model. Beyond self-report, we also validate our model with behavior, and also neuroimaging data, which is considerably more rigorous than most papers that propose a new model. We view this as a strength of the paper not a limitation of the model. We have attempted to clarify these issues throughout the manuscript and have also revised our discussion to better illustrate this point (p. 25, see also our responses to the reviewer’s major points *Query 3*).

6. *The description of the sample in Study 1 is unnecessarily complicated “In total, the data of 1,619 (812 females, 18.9 ± 2.0 (SD) years), 51 (33 females, 19.9 ± 1.6 years), 57 (45 females, 20.1 ± 1.8 years), and 53 (29 females, 20.9 ± 2.3 years) healthy graduate and undergraduate students were included for Study 1”. It should be explicitly stated how many participants have completed the questionnaire and how many have participated in the following procedures.*

Response: We agree with the reviewer and have made the suggested changes (p. 31, p. 32 and p. 37).

7. At page 16 of the SI it is stated that 6 LMMs were performed. Where these analyses controlled for multiple comparisons?

Response: We appreciate and are sympathetic to the reviewer's concerns for controlling false positives, but would like to emphasize that controlling for multiple comparisons is important when testing the same hypothesis multiple times. For example, if one was interested in knowing which brain region is involved in the comparison between faces and houses and we tested the hypothesis independently on 300k voxels, it is likely that many of these tests would be significant by chance and selecting voxels on the basis of these results would lead to overfitting to a specific dataset.

However, in this context, we have specific hypotheses about the relationships between experimental manipulations, appraisals, feelings and our computational model, and the testing of each hypothesis should have no relationship to increasing the false positive rate of a separate hypothesis. For example, testing the impact of our experimental design on subjective feelings of guilt should not increase the likelihood of whether our experimental design also impacted decisions to reject help. We do not believe that this constitutes testing the same hypothesis multiple times on the same dataset.

More importantly, we hope that our replications across multiple datasets and converging results across disparate types of data are more convincing of our efforts to minimize potential false positives than applying a simple heuristic correction like Bonferroni correction.

Reviewer #3 (Remarks to the Author):

There is a large body of psychological research on prosociality — people’s tendency to act (even altruistically at personal cost) to help others. Most of this research assumes that receiving help is an unproblematically good outcome for the beneficiary, so the puzzle is just why the benefactor is motivated to do it. However, as these authors note, receiving help is not a simple experience; indeed, the costs of receiving help are high enough that people often reject help altogether. The current project proposes that there are at least two distinct negative experiences associated with receiving help: feeling guilt for the burden on the benefactor, and feeling resentment for being coerced into reciprocation. They offer three rich and complementary sources of evidence for their claim: a survey; behaviour in a controlled interpersonal game, and patterns of neural activity.

The current paper tackles an important and under-explored topic with a rich and rigorous combination of methods. I think the work is an exciting contribution to the literature, and certainly deserved to be published. My comments below are intended to improve communication of the hypothesis and results.

Response: We thank the reviewer for the constructive suggestions, which we believe have helped us improve the communication of our hypotheses and results.

(1) Study 1 and the conceptual model: Distinguish guilt from gratitude

The overall concern of this paper is how people react to receiving help, depending on the intention that the beneficiary attributes to the benefactor. Here, the authors measure 4 psychological reactions (indebtedness, gratitude, guilt, and “obligation”). Although I generally agree with their view, I disagree with two choices in how they present and describe it.

First, and most importantly, their theoretical model conflates gratitude and guilt. However, both conceptually and empirically, it seems important to keep gratitude and guilt separate. Conceptually, gratitude is a positive emotion and guilt is a negative emotion. In study 1, the survey, gratitude has a very weak association with the ‘guilt’ component in people’s emotional descriptions; and gratitude strongly predicts accepting help whereas guilt quite strongly predicts rejecting help. Furthermore, in the real world, gratitude and guilt are likely to occur in different kind of relationships (in Alan Fiske’s scheme, gratitude might be more likely after receiving help in communal sharing relationships, whereas guilt might be more likely after receiving help in authority ranking relationships); and thus be associated with different subsequent behaviour.

When the authors formalize their model, they currently do not have a separate term for

gratitude versus guilt. As a result, they have an awkward workaround -- the effect of inferring the benefactor's caring is allowed to influence decisions to accept or reject help either positively (to capture the contribution of gratitude) or negatively (to capture the contribution of guilt). A more straightforward and conceptually clear model would have two separate terms for these distinct emotions -- even if patterns of game play does not allow for separate estimation of these terms.

Response: Our laboratory-based task was designed to test a key assumption in our theory that individuals trade-off feelings of communal concern and obligation when responding to receiving help. Although theoretical studies have proposed that both guilt and gratitude arise from altruistic favors and are part of a broader encompassing construct of communal concern^{44,45}, we agree with the reviewer that gratitude and guilt are two separate emotions with both conceptual and empirical differences. However, we were unable to clearly distinguish between these two emotions in the present study. Therefore, we make the following three revisions to clarify this point.

First, in Fig. 1, to distinguish gratitude and guilt, we included two separate arrows to indicate the effects of benefactor's altruistic intention on gratitude and guilt respectively. Moreover, we have added indebtedness in Fig. 1 to make it clear that indebtedness is a mixed negative feeling comprised of guilt and obligation, and gratitude is a distinct positive feeling that relates to indebtedness, but is independent.

Fig. 1 Conceptual model of indebtedness

Second, since in the current task, both gratitude and guilt were correlated with the perceived care from help, we acknowledge that our current task was unable to

distinguish between guilt and gratitude in the Discussion. Moreover, we have added discussions regarding the potential differences between gratitude and guilt (pp. 27 - 28):

“This work provides a substantial advance to our theoretical understanding of social emotions. First, we highlight the complex relationship between gratitude and indebtedness. We propose that feeling cared for by a benefactor, which we call communal concern^{44,45}, is comprised of both guilt and gratitude. Each emotion diverges in valence, with gratitude being positive⁶⁻⁹, and guilt being negative^{40-42,44,54}, but both promote reciprocity behavior. When faced with the offer of help, anticipated gratitude should motivate the beneficiary to accept help in order to establish or promote a relationship^{6,7}, whereas anticipated guilt should motivate the beneficiary to reject help out of concern to protect the benefactor from incurring a cost^{44,54,93}. Although we observed some evidence supporting this prediction, our interactive task was not designed to explicitly differentiate guilt from gratitude, which limits the ability of our computational model to capture the specific contributions of guilt and gratitude to communal concern and likely impacted identifiability of the parameters of the model for accepting/rejecting help (see *Computational Modeling in Materials and Methods*). Future work might continue to refine the relationship between these two aspects of communal concern both in terms of behaviors in experiments and computations in models^{54,62-64,67,68,90}.

Second, our model provides a framework to better understand the role of relationships and contexts in generating feelings of indebtedness within a single individual. Different types of relationships (see Clark and Mills’s theory of communal and exchange relationships^{4,5}, and Alan Fiske’s Relational Models Theory⁹⁴) have been theorized to emphasize different goals and social norms which can impact social emotions^{95,96}. For example, communal relationships prioritize the greater good of the community and are more conducive to altruistic sharing, which can be signaled by altruistic favors³⁻⁵. In contrast, exchange relationships are more transactional in nature^{2,4,5,10-12} and emphasize maintaining equity in the relationship, which can be signaled by strategic favors⁹⁴. Our model proposes that perceptions of the benefactor’s intentions directly impact the feelings experienced by the beneficiary (e.g., guilt & obligation). Although we deliberately attempted to minimize aspects of the relationship between the benefactor and beneficiary by making players anonymous to control for reputational effects, future work might experimentally manipulate these relationships to directly test the hypothesis that relationship types differentially moderate the responses of gratitude and subcomponents of indebtedness.”

We thank the reviewer for their suggestion of viewing gratitude and guilt from the perspective of Alan Fiske’s Relational Models Theory (Fiske, 1992). Therefore, as stated above, we propose that future work might explore manipulating these

relationships to more directly test the hypothesis that relationship types differentially moderate the responses of gratitude and subcomponents of indebtedness. However, whether the feeling of gratitude or guilt dominates in one relationship in comparison to others remain largely unclear. For example, while some evidence showed that benefits from help increase beneficiary’s gratitude when one applies a communal sharing model and not equality matching or authority ranking (Simão, 2013). In contrast, several researchers have independently postulated or described a process in which hierarchical authority ranking relationships derive from gratitude or obligation to reciprocate in equality matching relationships (Fiske, 1992; Fiske, 2002). Similarly, while some theories suggest that guilt appear to be strongest in the context of communal relationships, which supports communal relationships by fostering care and avoiding transgressions, the role of guilt in maintaining equity may reflect its role in equality matching relationships (Baumeister, 1994). Therefore, we did not develop specific hypotheses regarding how gratitude and guilt vary across different relationships, and leave the questions for future studies and now note this in the discussion.

Third, for the computational modeling, because the current task was unable to distinguish between guilt and gratitude mathematically, we were unable to compare different hypothesized models regarding the how much guilt and gratitude contribute to the communal concern. The reviewer astutely identifies one of the things we struggled with in the model. We appreciate the reviewer’s suggestion to include gratitude in the model, even if we never estimate a weight on gratitude. We agree with the reviewer that this helps articulate how we view the relationship between gratitude and indebtedness that we propose in the conceptual model. We have now proposed the potential model function capturing how guilt and gratitude contribute to the communal concern in the *Materials and Methods, Computational modeling* section (p. 44):

“Based on our conceptual model (Fig. 1), we define $U_{Communal}$ as a mixture of feelings of gratitude $U_{Gratitude}$ and guilt U_{Guilt} , in which the parameter δ_B ranges from [0,1] and specifies how much player B cares about gratitude relative to guilt. As the focus of this paper is on indebtedness, we set δ_B to zero and leave it to future work to build a model of gratitude $U_{Gratitude}$ and explore its relationship with guilt (see also *Discussion*). Thus, for this paper $U_{Communal}$ is synonymous with U_{Guilt} .

$$U_{Communal} = \delta_B * U_{Gratitude} + (1 - \delta_B) * U_{Guilt} \quad \text{Eq. 6}$$

We separately modeled the appraisals of second-order beliefs E_B'' of the benefactor’s expectation for repayment (Eq. 2) and perceived care ω_B (Eq. 3), and used them to capture guilt and obligation feelings (Eq. 7 and Eq. 8). We defined the appraisal/feelings of U_{Guilt} and $U_{Obligation}$ as:

$$U_{Guilt} = \begin{cases} -\left(\frac{\omega_B * \gamma_B - D_B}{\gamma_B}\right)^2 & \text{Reciprocity} \\ \omega_B & \text{Accept/Reject Help} \end{cases} \quad \text{Eq. 7}$$

$$U_{Obligation} = \begin{cases} -\left(\frac{E_B'' - D_B}{\gamma_B}\right)^2 & \text{Reciprocity} \\ \frac{E_B''}{\gamma_B} & \text{Accept/Reject help} \end{cases} \quad \text{Eq. 8''}$$

References:

- Simão, C. P. C. (2013). Communal Sharing and Gratitude: How They Interrelate. (Doctor of Psychology), Instituto Universitário de Lisboa, Lisbon, Portugal.
- Baumeister, R. F., Stillwell, A. M., & Heatherton, T. F. (1994). Guilt: an interpersonal approach. *Psychological Bulletin*, 115(2), 243-267.
- Fiske, A. P. (1992). The four elementary forms of sociality: framework for a unified theory of social relations. *Psychological Review*, 99(4), 689.
- Fiske, A. P. (2002). Socio-moral emotions motivate action to sustain relationships. *Self and Identity*, 1: 169-175.

(1a) A better word for obligation?

Relatedly, the authors describe three main emotional responses to receiving help. Guilt and gratitude, described above, and “obligation”. Of course, the actual materials were in Chinese, so I am not sure what word exactly the participants saw. But the emotional experience the authors describe -- in Fig s1: “a negative feeling of pressure to repay caused by the other’s ulterior intentions” -- seems to me a better fit for “resentment”. “Obligation” is not commonly used as an emotion word in English, and doesn’t have the negative valence that their theory seems to imply -- e.g. as the strongest motive for rejecting help. Indeed, feeling “obliged to reciprocate” does not predict rejecting help; whereas feeling “resentment” is a negative emotion and could lead to both rejecting help and reciprocating help, analogous to the argument that they make for “guilt”.

Response: We appreciate the reviewer’s suggestion of potential alternative words such as “resentment” instead of “the sense of obligation”. We think resentment might be secondary to obligation, meaning that people may resent feeling obligated. We don’t believe that the term works across all contexts to warrant replacing obligation throughout the text, but instead now we use “resentment” to help define our use of obligation (p. 4). We completely agree with the reviewer that, compared with guilt, “the sense of obligation” is not as commonly used to describe emotions in Chinese or English. However, the usage of “the sense of obligation” is consistent with previous work on indebtedness^{13,14,21,46,47}, and the recent discussions on “the sense of obligation” and “the sense of should”, showing that it is a internalized social pressure

or a motivational sentiment in response to social expectations, which emerge from a shared perspective with collaborative partners and in-group members, that motivates us to fulfill our cooperative duties⁴⁹⁻⁵¹. It has a demanding force, with a negative quality ‘I don’t want to, but I have to’, and is linked to negative emotional responses, such as pressure, burden, anxiety, and resentment⁴⁹⁻⁵¹. Throughout the manuscript we have tried to clarify these points and have added details to the Introduction (p. 4),

“In contrast, if the benefactors’ intentions are perceived to be strategic or even duplicitous, then the beneficiary is more likely to feel a sense of obligation^{13,14,21,46,47}. Obligation can also motivate the beneficiary to reciprocate^{13,14,21,46,47}, but unlike communal concern, it arises from external pressures, such as social expectations and reputational costs^{48,49} and has been linked to feelings of pressure, burden, anxiety, and resentment⁴⁹⁻⁵¹. In everyday life, inferences about a benefactor’s intentions are often mixed and we propose that indebtedness is a superordinate emotion that includes feelings of guilt for burdening the benefactor⁴⁰⁻⁴³ and also social obligation to repay the favor^{13,14,21,46,47}. ”

and the Discussion as well (p. 28).

“Third, we present new evidence exploring the relationship between indebtedness and guilt. Guilt and indebtedness are interesting emotions in that they are both negatively valenced, yet promote prosocial behaviors. In previous work, we have operationalized guilt as arising from disappointing a relationship partner’s expectations^{39,54,55,97}, which is conceptually related to the feeling of obligation in this paper. This feeling results from disappointing a relationship partner or violating a norm of reciprocity and is a motivational sentiment evoked by social expectations reflecting a “sense of should” that is associated with other negative affective responses such as feelings of pressure, burden, anxiety, and even resentment⁴⁹⁻⁵¹. In other work, we have investigated how guilt can arise from causing unintended harm to a relationship partner^{68,98}. This is conceptually more similar to how we frame guilt here, which arises from the feeling that one has unnecessarily burdened a relationship partner even though the help was never explicitly requested by the benefactor. We believe that continuing efforts to refine mathematical models of emotions across a range of contexts, will eventually allow the field to move beyond relying on the restrictive and imprecise semantics of linguistic labels to define emotions (e.g., guilt, gratitude, indebtedness, obligation, feeling, motivation, etc.).”

(1c) What is indebtedness?

Indebtedness does not appear in Fig 1, the theoretical model. The authors might think this is a fourth distinct emotional response (as suggested in Fig 2B); or is a cognitive appraisal of a factual situation (“I owe \$4”); or is a motivation to repay a perceived

debt (suggested in Fig 4A and most of the text). Each of these is a slightly different view. Their formal model is closest to the third of these, because indebtedness is the parameter that is minimized to define the optimal repayment.

In summary, redescribing their conceptual model for better clarity: I think their conceptual model should actually be described like this:

receiving help can cause people to feel at least three different emotions, gratitude, guilt, and/or resentment (“obligation”);

wanting to avoid feeling (the negative emotions) guilty or resentful can lead to rejecting help altogether;

if help is received, then gratitude, guilt, resentment, or a combination of them, can lead to a motive to repay or reciprocate the help, namely, indebtedness.

Response: As suggested, we have added indebtedness in Fig. 1 to make it clear that indebtedness is a mixed feeling comprised of guilt and obligation.

In the current study, based on previous studies on indebtedness, we propose the word “indebtedness” refers to the negative feeling after receiving help, which consists of at least two emotional components: “guilt”,⁴⁰⁻⁴³ referring to the beneficiary’s negative feeling of guilt for burdening the benefactor, and “obligation”^{13,14,21,46,47}, referring to the beneficiary’s negative feeling of the pressure to repay driven by the benefactor’s expectation for repayment. These two components are derived from appraisals about the benefactor’s intentions and can differentially impact the beneficiary’s reciprocal behaviors. We have now substantially revised the text to make this clearer (pp. 4 - 5).

Their formal model is simpler than this conceptual model in various ways. For example, in their formal model, the degree to which participants attribute strategic (i.e. second-order belief) versus caring (i.e. perceived care) motivations is constrained to trade off linearly (governed by one parameter that varies across individuals, kappa). Thus, the amount of guilt versus resentment a participant experiences is defined as anticorrelated. This over-simplification fits the two first principle components of individual differences in participants in the game, but seems unlikely in real life. That’s fine; the authors can just explicitly acknowledge how the model is simplified for tractability.

Response: According to the model terms for second-order belief (E_B'' , Eq. 2) and perceived care (ω_B , Eq. 3), when the benefactor’s cost is fixed, E_B'' and ω_B are anti-correlated, trading off by the parameter kappa. Critically however, our experimental design introduces a non-linearity which tampers this relationship: on some trials repayment was possible, while on others it was not. Therefore, when combining the data of all trials in the experiment, E_B'' and ω_B are actually uncorrelated (as shown in the figure below, i.e., Fig. S4). This is because in the Repayment impossible condition, E_B'' is always zero, while ω_B is positively correlated

with the benefactor's cost. In the Repayment possible condition, although E_B'' will reduce ω_B at one specific level of the benefactor's cost, both these two variables increase as the benefactor's cost increases; these two variables are positively correlated in this condition and the extent of correlation is defined by the parameter kappa. Therefore, these two variables are uncorrelated across all trials in the experiment. This is consistent with the relationships between self-reported ratings shown in Fig. S3 A-B and Tables S5 and S6. We include a short note about this when describing the model in the main text (p.15).

$$E_B'' = \begin{cases} 0 & \text{Repayment impossible condition} \\ D_A & \text{Repayment possible condition} \end{cases} \quad \text{Eq. 2}$$

$$\omega_B = \frac{D_A - \kappa_B * E_B''}{\gamma_A} \quad \text{Eq. 3}$$

Fig. S4 Relationships between model representations of second-order belief E_B'' and perceived care ω_B in the current study (A) and the simulated data with different levels of κ (B).

We acknowledge that our computational model to some extent simplifies the appraisal and emotion generating processes. Our model operationalizes the appraisals of perceived care and second-order belief using information available to each participant in the task (i.e., co-player's helping behavior and manipulation about the participants' ability to reciprocate). It is possible that these appraisals might not be orthogonal in other contexts; applying the model to other experimental contexts may require developing new ways to model these appraisals. Moreover, as pointed by the reviewer, there is some variability in participant self-reported second-order belief that could be captured by the current model. For example, some participants reported non-zero second-order belief in Repayment impossible condition (e.g., Fig. S5A). Since there is not enough evidence regarding how participants generated the second-order belief, we used the current simplified model term (Eq. 2) for second-order belief for tractability. Although our model performed well to capture the patterns of participants' reciprocity after receiving help and decisions of whether to accept help, we highlight the

importance of future studies to improve the quality of this model. Corresponding discussions are added on p. 29:

“Second, our computational model may oversimplify the appraisal and emotion generating processes. Our model operationalizes the appraisals of perceived care and second order belief using information available to each participant in the task (i.e., benefactor’s helping behavior and manipulation about the participants’ ability to reciprocate). These appraisals are likely context-dependent and our model may not generalize to other experimental contexts without significant modification to how these appraisals are operationalized. Although our model performed well capturing the patterns of participants’ reciprocity behaviors in this task, we believe it is important to continue to refine this model in future studies.”

(2) Distinguish experimental conditions from participant inferences

In the interactive game, benefactors paid money to relieve pain (from later electric shocks) for the participants (from the benefactors’ perspective: anonymous beneficiaries). To influence participants’ inferences of the benefactors intentions, there were two conditions: when the participants could versus could not “reciprocate” by returning money to the benefactor. The authors argue that when the participant could not reciprocate, donations must be perceived as ‘altruistic’, whereas when the participant could reciprocate, donations might be perceived as ‘strategic’ (to elicit money in return); the authors name the conditions according to these predicted inferences.

However, as their own data show (Fig 3), participants’ inferences are mixed. When returning money was possible, participants attributed a mixture of motives to benefactors -- both altruistic care and strategic coercion -- and actually felt more gratitude than obligation or guilt, on average (3a). And on the other hand, even when returning money was impossible, participants attributed non-zero expectations of reciprocation to the benefactor (3a), which increased with the benefactor’s cost (3b).

All of this is perfectly consistent with the authors’ theory -- but it is very confusing for a reader. So: please use different language to describe the experimental manipulation (e.g. \$ Return Allowed; \$ Return Impossible) and the resulting inferred intentions of the benefactor (e.g. Care, Coercion).

Response: We thank the reviewer for this suggestion. In the initial manuscript, the names of conditions were defined based on the inferred intentions in the two conditions. As pointed by the reviewer, since participants’ ratings of second-order belief and perceived care were mixed in both conditions, these inferred conditions of intentions might lead to misunderstanding. Therefore, as suggested, to show the actual

condition manipulation of our task rather than the inferred conditions of intentions, we have now replaced the original condition names of “Altruistic condition” and “Strategic condition” with “Repayment impossible condition” and “Repayment possible condition” in the manuscript.

(3) Study 2 and the formal model

In study 2, the interpersonal game, the benefactor can pay money to reduce the electric shocks that will be received by the participant. There are two key manipulations: either the participant will be allowed to repay the money, or the participant will not be allowed to repay it; and either the participant is allowed to reject the help, or the participant is not allowed to reject it. The game has three outcome variables:

Self-reported emotions: how much gratitude, guilt, and “obligation”, as well as indebtedness, the participant reported feeling on each trial.

Accept / reject decisions.

Repayment amounts.

Overall, I think the argument of this study is that all three of these outcomes can only be understood by taking into account participants’ inference about the intent of the benefactor: whether the benefactor expected to be repaid, or genuinely cared about the participant’s pain.

What sometimes this confusing is that the study uses three correlated but different proxies for participants’ inferences about benefactor intentions:

experimental condition: the benefactor’s intentions are caring in the “no-repayment-possible” condition and strictly about repayment the “repayment-possible” condition.

self-report: participants were asked to explicitly report their inferences of benefactor expectations and caring.

expressions in a formal model. The benefactor’s expectation of return is the money donated on the “repayment-possible” trials, and the perceived care is the money donated on “no-repayment-possible” trials; the formal model also includes some perceived care on the “repayment-possible” trials, by allowing that (variably across individuals) participants see donation on these trials as reflecting a mix of expectation of repayment and genuine care; this individual difference is captured by a free parameter $Kappa$, that varies across individuals, and is fit separately to the accept/reject and repayment decisions.

Overall, the hypothesis that the intent of the benefactor influences the three outcomes (emotions, rejections, repayments) is tested with respect to some of the proxies for inferred intent, but not in all or the most rigorous possible ways. So, the argument of the paper could be clarified, by making all of this structure more accessible to a reader

(this took days to figure out); and it could be improved by doing stronger versions of the tests.

Response: We apologize for the confusion and deeply appreciate the reviewer taking the time to make sense of all of this.

In the initial manuscript, the names of conditions were defined based on the inferred intentions in the two conditions, which might lead to misunderstanding and confusions. Therefore, as the reviewer suggested, we have now replaced the original condition names of “Altruistic condition” and “Strategic condition” with “Repayment impossible condition” and “Repayment possible condition” in the manuscript.

In Study 2, we aimed to manipulate participants’ appraisals, emotions and behaviors by condition manipulations, and test our theoretical model by examining appraisal-emotion associations and emotion-behavior associations. The main analyses consisted of three parts:

- 1) **Manipulation Check:** The effects of experimental conditions on participants' appraisals, emotions and behavioral responses. We examined the effects of experimental conditions on appraisals, emotions and behaviors to test whether our manipulations successfully impacted participants’ appraisals of the benefactor’s intentions behind the help, emotions and behaviors.
- 2) **Conceptual Model Test:** Relationships between appraisals and emotions. We conducted correlation analysis, factor analysis and mediation analysis on self-reported appraisals and self-reported emotions to investigate the relationships between appraisals and emotions. The two-factor results and results of mediation analysis supported our theoretical model at appraisal-emotion association level, showing that the guilt and the sense of obligation are derived from appraisals of the benefactor's altruistic intentions and strategic intentions respectively.
- 3) **Computational Model Test:** We built a formal computational model to capture the effects of the components of appraisals and emotions on behaviors. We demonstrate that the model-derived representations of communal concern and obligation can predict appraisals, emotions, and the two factor components derived from the factor analysis.

Other analyses are done as suggested by the specific points of this query (see below). We apologize for not clearly articulating the structure of our analyses, due to the concern of balancing methodological detail and readability. We have moved most part of our methods in the *Supplementary Materials* to the *Materials and Methods* section in the manuscript (pp. 31 - 53). To make the structure of the paper more accessible to readers, in the *Materials and Methods* section, 1) the methods of different studies are presented separately, 2) the order of analyses is re-organized based on the logic of

analyses, and 3) the aim of each analysis is clearly stated at the beginning of each section.

(3a) Validating the conceptual model in the self-reported emotions

In Fig 3a, the authors show that self reported emotions (outcome I) depend on experimental condition (proxy A) (and dependence on benefactor cost is shown in Fig S2). That is good!

But it would also be useful to test the conceptual model using the continuous data available in self-report (proxy B). Self-reports inference of benefactor intentions should predict, on a trial by trial basis, the reported level of gratitude, guilt, and 'obligation' on each trial. Probably the resulting correlations will be much stronger than the effects of condition shown here, because (according to the authors' own theory) there is lots of variability within conditions. (Note: in this analysis, make sure to carefully distinguish between variability across trials within participants, and variability across participants -- those two sources of variability are not distinguished in Fig 3 currently). Also, I recommend you replace the principle components analysis with the mediation models in Fig S3; or maybe a modified mediation model, combining across studies, and more fully testing the conceptual causal hypothesis: Experimental design -> two inferred intentions -> three emotional responses -> indebtedness.

Response: As suggested by the reviewer, we have now revised our analyses on relationships between self-reported appraisals and emotions.

First, to reveal the relationships between appraisals (second-order belief and perceived care) and emotions (indebtedness, guilt, sense of obligation and gratitude), we estimated the correlations between these variables at both within-participant and between-participant levels. For within-participant analysis, for each pair of these six variables, we estimated the pearson correlation for each participant, transformed the data using a fisher r to z transformation, and then conducted a one-sample test using z values of all participants to evaluate whether the two variables were significantly correlated at the group level. This analysis captured the variability of appraisals and emotions across trials within participants (Fig. S3A). For between-participant analysis, for each of the six variables, we computed the average value of the variable across all trials for each participant. We then estimated the correlations between each pair of variables based on variability across participants (Fig. S3B). These two lines of results were the similar as the original Fig. 3D. These results are now reported in Fig. S3 and Tables S5 and S6. Corresponding methods are added on pp. 41 - 42.

Second, as suggested by the Reviewer 2, we have now replaced principal component analysis with factor analysis, and the results remain the same (p. 13). We applied

exploratory factor analysis in Study 2 to identify the number of common factors and the relationships between appraisals and emotions purely from our data. Consistent with our hypothesis, results showed a two-factor solution (Fig. 4D and Fig. S2E). The Communal Factor reflected participants' perception that the benefactor cared about their welfare and resulted in emotions of gratitude and guilt, while the Obligation Factor reflected participants' second-order beliefs about the benefactor's expectation for repayment and the sense of obligation. Interestingly, indebtedness moderately loaded on both factors. Next, we conducted confirmatory factor analysis using the data of Study 3 to test the two-factor model built by Study 2 in an independent sample. Results showed that the fitness of this two-factor model is appropriate (RMSEA = 0.079, SRMR = 0.019, CFI = 0.986, TLI = 0.970; Fig. S7G) (p. 18).

Third, as suggested, instead of the original two separate mediation models, we conducted a modified mediation analysis by combining the data across the two experiments in Study 2. This model included the experimental variables as independent variables, the two appraisals (second-order belief and perceived care) as mediators, and the three emotions (gratitude, guilt and the sense of obligation) as dependent variables. Results remained the same, showing that second-order beliefs and perceived care appraisals differentially mediated the effects of the experimental manipulation on emotional responses (indirect effect = 0.59 ± 0.04 , $Z = 14.49$, $p < 0.001$; Fig. 4E and Fig. S3C). Second-order beliefs mediated the effects of the experimental manipulations on the sense of obligation (Indirect effect = 0.22 ± 0.03 , $Z = 7.18$, $p < 0.001$), while perceived care mediated the effects of the experimental manipulations on gratitude (Indirect effect = 0.19 ± 0.01 , $Z = 13.72$, $p < 0.001$) and guilt (Indirect effect = 0.17 ± 0.01 , $Z = 13.23$, $p < 0.001$) (p. 13). Corresponding methods are updated on pp. 42 - 43.

We have made several changes to Fig. 4 (the original Fig. 3) to better illustrate the results regarding the relationships between appraisals and emotions. First, as suggested by the reviewer, we decided to include the mediation model in Fig. 4. However, since this model is too complex with 18 coefficients in the figure, which may jeopardize the understanding of the structure of model and reduce the speed of reading, we now present the detailed results of the model in the *Supplementary Materials* (Fig. S3C), and include a simplified representation of this model in the main text (Fig. 4E). This simplified representation illustrates the total mediation effects for how appraisals mediated the effects of all experimental manipulations on emotions. Second, although both the figures of the correlation matrix and the factor analysis are aimed to demonstrate the relationship between variables, the factor analysis can better illustrate the two components of appraisals and emotions with statistical inferences. Therefore, we decided to include the figure of factor analysis in the Fig. 4D and moved the figures of the correlation matrix to *Supplementary Materials* Fig. S3 A and B.

(3b) Correlation of inferred intentions in self-report versus the formal model

As one measure of the validity of the formal model, the authors say that the terms for the inferred benefactor intentions in the formal model (proxy C) correlate with participants' self-report of benefactor intentions (proxy B). Although these correlations look very high, this is actually a weak test of the model, because the shared variance is determined by the experimental design.

For example: the expected repayment ("second order belief") in the formal is the amount the benefactor paid on trials when repayment is possible. Participants reports of expected repayment are correlated with this model -- participants don't report benefactors expecting to be repaid more than they paid (e.g. paid 1pt expects to be repaid 20pts), and on average report lower expectations for repayment when repayment is impossible (though there is tons of variability here, as noted above). A stronger and more conservative test of the model would be to show that it correctly captures differences between participants, in inferred intentions, after accounting for the experimental design. There is a lot of variability in participant reported benefactor intentions within each trial type (figure S4A). The formal model of expected repayment does not -- by definition -- capture any of this structure. On the other hand, the formal model of perceived care is fit to participant behaviour using a free parameter (κ); so maybe the formal model's quantity for perceived care would correlate with self-reported perceived care, within trial types but across participants? That would be a better test of convergence between the model and the self-reported inferences.

Response: As shown in Fig. 4 A and B, both extra information about the benefactor's intention and benefactor's cost significantly modulated participants' ratings of second-order belief. Since there is no other evidence regarding how participants generated the second-order belief, for tractability, in the computational model, participant's second-order beliefs (E_B'') of how much the benefactor expected in each trial were completely determined as the function of extra information about the benefactor's intention and benefactor's cost (D_A) (Eq. 2).

$$E_B'' = \begin{cases} 0 & \text{Repayment impossible condition} \\ D_A & \text{Repayment possible condition} \end{cases} \quad \text{Eq. 2}$$

We thank the reviewer for the suggestions of testing the correlation between model predictions and self-reported ratings from the view of individual difference. In the current study, although the order of trials was randomly determined for each participant, the trial set was the same for all participants (Table S2). Therefore, given that model term for second-order belief was completely determined by experimental conditions, there will be no variability across participants in the model term of second-order belief.

For the perceived care, we agree with the reviewer that, the model term for perceived care was fit to participant behavior using a free parameter (κ), which should capture the variability across participants. Thus, the model term for perceived care should be correlated with self-reported perceived care, within trial types and across participants. That is indeed what we observed. We found that the higher the average model prediction of perceived care for one participant, the higher the average self-reported perceived care for this participant ($r = 0.27, p = 0.004$). This result is added on p. 16.

We agree with the reviewer that, there is some variability in participant self-reported second-order belief that could not be captured by the current model. For example, some participants reported non-zero second-order belief in Repayment impossible condition (e.g., Fig. S5A). Since there is not enough evidence regarding how participants generated the second-order belief, we used the current simplified model term (Eq. 2) for second-order belief for tractability. Although our model performed well to capture the patterns of participants' reciprocity after receiving help and decisions of whether to accept help, we highlight the importance of future studies to improve the quality of this model. Corresponding discussions are added on p. 29.

(3c) Validating individual differences, across behaviours

The authors measure individual differences in three separate outcome variables, in the same individuals. It is not strictly necessary, but would be reassuring, to see that the estimates of individuals converge across these variables. For example, that a single person's kappa (which is related to estimates of caring motivations even in the presence of strategic incentives) is correlated, when modeled in accept/reject decisions, versus when modeled in reciprocity decisions.

Response: As suggested, we examined the relationships between the model parameters for reciprocity and the decisions of whether to accept or reject help. We observed no significant cross-behavior correlation for the three parameters (θ : $r = 0.12, p = 0.204$; ϕ : $r = -0.03, p = 0.738$; κ : $r = -0.01, p = 0.915$). However, we note that the parameters of the indebtedness model were highly identifiable for the reciprocity model (reciprocity $r = 0.94 \pm 0.07, p < 0.001$; Table S9), but to a lesser degree for the accept/reject help model ($r = 0.43 \pm 0.40, p < 0.001$; and Table S10). This may explain the observed non-significant correlations (pp. 16 - 17).

We have further investigated this issue with the accept/reject help model and have discovered the reason of instability in the parameters. For decisions of whether to accept/reject model,

$$\begin{cases} U(\text{Accept}) = \theta_B * \pi_B + (1 - \theta_B) * (\phi_B * U_{\text{Communal}} - (1 - |\phi_B|) * U_{\text{Obligation}}) \\ U(\text{Reject}) = 0 \end{cases}$$

Eq. 10 (Model 2.1)

U_{Communal} has a more complex influence on behavior, with guilt decreasing the likelihood of accepting help to avoid burdening a benefactor^{34,54}, and gratitude motivating accepting help to build a communal relationship^{6,7}. However, because $U_{\text{Communal}} = U_{\text{Gratitude}} = U_{\text{Guilt}} = \omega_B$ (i.e., perceived care appraisal) in this formulation, there is no variability in the design for the model to be able to disentangle the effect of gratitude from that of guilt. To address this complexity, we constrain ϕ to be within the interval of $[-1, 1]$, and explicitly divide up the parameter space such that $\phi > 0$ indicates a preference for gratitude and motives the participants to accept the help, while $\phi < 0$ indicates a preference for guilt and motives the participants to reject the help. Unfortunately, if participants are equally sensitive to gratitude and guilt, ϕ will reduce to zero and the weight on obligation increases, which decreases the model fit and leads to some instability in the parameters. Though this model is far from perfect, it does capture the unique predictions to accept/reject behavior proposed in our conceptual model and accurately predicts behavior with about 80% accuracy. We have added more details about all of this in the results section (pp. 16 - 17), methods section (p. 46), and acknowledge the limitations of the model and our task in being able to disentangle the effects of gratitude and guilt in the discussion (p. 27). We are currently planning future studies to explicitly examine the relationship between guilt and gratitude.

Finally, we note that from the perspective of psychological processes, our model for reciprocity captures how *experienced* emotions influence the beneficiary' behavioral responses. In contrast, the model for whether to accept help reflects the influences of *anticipatory* emotions on the beneficiary' behavioral responses. Whether there exist shared or differential cognitive bases underlying experienced and anticipatory emotions is still an open question and is beyond the scope of the present study. We are thinking about how to address this issue in future studies.

(4) What do the neural data add?

First analysis shows that training of 4/5ths of the data (fit to estimates of expectation of return and perceived care), can modestly but significantly decode these values in the left out fifth of the data, within participants. One big contribution to these trialwise estimates is the trial type: whether it is a return possible or a return impossible type. These two conditions lead to big differences in the mean of estimates of expectation of return (high when return is possible) and perceived care (high when return is impossible). It is thus possible that what is being decoded, and cross-validated, is just

this difference between conditions, rather than any more specific underlying representation. Maybe just the observation that these conditions can be decoded from the neural activity at the outcome phase is nevertheless an argument in favour of the authors' conceptual model.

The second model showed that these neural patterns could predict reciprocity behaviour -- here again, it would be nice to know whether these predictions are better than similar predictions made just in terms of the two conditions. Does the formal expression for expectation and perceived care fit the data (and predict behaviour) better than the qualitative condition difference?

Response: We thank the reviewer for suggesting analyses to rule out alternative explanations such as whether the multivariate brain models are simply capturing their differences between the Repayment possible and Repayment impossible conditions. We have run additional analyses to rule out this possibility, and have added these results to the paper (p. 21).

First, we tested our cross-validated predictions controlling for repayment condition, and found that the results were still significant after controlling for this experimental manipulation (communal concern: average $r = 0.18 \pm 0.02$, $p < 0.001$; obligation: average $r = 0.04 \pm 0.02$, $p < 0.024$), indicating that the experimental manipulation did not mediate the effects.

Second, we trained a whole-brain model to classify Repayment possible from Repayment impossible conditions. The overall cross-validated accuracy was fairly low, and not significant ($accuracy = 55.0 \pm 1.25\%$, $p = 0.746$). This model was unable to predict the trial-by-trial amount of reciprocity (average $r = -0.07 \pm 0.02$, $fisher-z = -0.07 \pm 0.02$, $p = 0.999$). These results indicated that the effects of our brain models for communal and obligation feelings were not simply driven by the differences between Repayment possible and Repayment impossible conditions.

For the third analysis, they look at individual differences. In principle, this ought to be the key analysis for testing the fit of their model. For example, if their model is capturing individual level variance then fits to the data would be better if using the model as fit to the participant's own responses, than the same structural form of the model but fit to a different participant's data.

In fact the analysis that they do present of individual differences is somewhat confusing. They show that the similarity between a participant's neural pattern predicted by feeling "obligation" (resentment) and the neural pattern predicted by behavioural reciprocity, predicts the level of the participant's reciprocity. This is a somewhat odd analysis, especially given that in these authors' conceptual model, reciprocity is likely to be caused by any combination of gratitude, guilt and

obligation/resentment. So why should only the pattern similarity of between obligation and reciprocity predict behaviour? Indeed, it seems plausible that this was not the only similarity analysis conducted by these authors, and thus reflects some experimenter degrees of freedom in the choice of analyses presented in the final paper, not entirely accounted for in the calculation of frequentist statistics. (If not, and these analyses were preregistered or defined a priori, the authors should be more clear about this).

Again, to be clear, I do agree that the authors' neural data are consistent with their hypothesis that in order to understand the pattern of neural responses that occur upon receiving a generous offer of help, it is necessary to consider the participant/recipient's inference of the beneficiary's intention and that this is the central contribution of the neural data. I just believe that the evidential basis of this claim, and it's strength could be communicated more effectively.

Response: We apologize for the confusion in the individual difference analysis in Study 3. Specifically, in this analysis, we were interested in whether we could tell purely from brain activity how much participants were motivated by feelings of communal concern or obligation when making their decisions. The logic was to determine which of the brain patterns representing each motivation was relatively more aligned to the brain pattern predicting how much money they chose to reciprocate. We hypothesized that the relative influence of a particular feeling on behavior should be reflected in the spatial alignment of their corresponding brain patterns⁷⁷. For example, if a participant weights obligation more than communal concern during reciprocity (higher $1 - \phi$ estimated from the behavioral model), then the spatial similarity between their obligation brain pattern and the pattern that directly predicts their reciprocity behavior (reciprocity brain pattern) should be relatively higher compared to the spatial similarity between their communal concern pattern and reciprocity brain pattern. Our results support this hypothesis. Participants who cared more about obligation relative to communal concern (higher behavioral $1 - \phi$) also exhibited greater spatial alignment between their obligation and reciprocity brain patterns relative to communal concern and reciprocity patterns, $r = 0.68$, $p < 0.001$ (Fig. 7C).

We have also run the analysis suggested by the reviewer. We assessed the importance of the participant-specific model parameters estimated from the neural utility model (i.e., ϕ) by generating a null distribution of predictions after permuting the estimated ϕ parameter across participants 5,000 times. We found that the participant-specific weightings were highly important in predicting behavior and that our neural utility model significantly outperformed a null model using parameters estimated for a different participant, $p < 0.001$ (Fig. 7B).

We have substantially rewritten this section of the paper and hope that it is more clear now (pp. 22-23).

Minor point: How does the efficiency manipulation affect emotions?

They measured feelings of obligation, guilt and gratitude after benefactors gave different amounts, with different efficiency. How does efficiency affect these emotions? Do recipients feel less guilt when a small gift had a larger effect? And more guilt when a larger gift had a smaller effect? Do donations of low efficiency create more gratitude? There is some hint of these effects on self-reported emotions Fig S3 -- and perhaps they need to stay there, because of how complicated this paper is already. But, well, they are interesting questions.

Response: Given that we observed no interaction effect between efficiency and the other variables, for brevity we combine the two behavioral datasets when reporting results in the main text and did not report the results of efficiency in the main text. We agree with the reviewer that how efficiency (or the benefit the participant obtained from help) influences participants' emotional responses are interesting questions. We have now added these results in the Supplementary Materials pp. 6 - 7.

In brief, we found that Efficiency contributed significantly and positively to participants' self-reported ratings of gratitude and indebtedness; these effects were not significant for self-reported ratings of guilt and obligation.

To be noted, since the effects of Efficiency (or the benefit the beneficiary obtained from help) on beneficiary's emotions are not the focus of the current study, the range of Efficiency was relatively small in the current study (i.e., 0.5, 1, 1.5). Therefore, it is possible that the current manipulation was not efficient to capture some relatively extreme situations, which resulted in the current non-significant effects of Efficiency on guilt and the sense of obligation. For example, will a beneficiary feel less guilt when the benefactor's small cost has a larger effect, or feel more guilt when the benefactor's large cost has a small effect? Future specially designed studies are needed to explore how the efficiency of help influences the beneficiary's emotional responses.

Surveys. This is not "experience sampling" (which implies repeated measurements). It's just a survey.

-- Rebecca Saxe

Response: We agree with the reviewer and have now replaced all "experience sampling" with "online questionnaire" to avoid misunderstanding.

Reviewers' Comments:

Reviewer #2:

Remarks to the Author:

I thank the authors for revising the paper. The manuscript has improved, but unfortunately not all of my concerns have been resolved.

1) In response to my first point (conceptual clarity) the authors claim that "both guilt and obligation are feelings/emotions ...". The cited literature suggests that guilt and obligation play a role in indebtedness, but "obligation" is not classified as an emotion. Instead, most researchers agree that obligations derive from norms that were established to maintain cooperative societies (see e.g., W Tomasello M. (2020). The moral psychology of obligation. Behavioral and Brain Sciences⁴³). I am sorry to say that the conceptual foundations of the manuscript are still unclear. As a result, the modelling results are hard to interpret and it is hard to say how the presented models broaden our understanding of indebtedness.

2) The authors use meta-analytic decoding to interpret their imaging results. They report that "communal feelings were related activity that was similar to the reciprocity results, but was additionally associated with Default mode term. Obligation activity was highly associated with terms related to "Social," Theory of mind (ToM), and Memory. These interpretations introduce yet different concepts. Are we looking at emotions as claimed in the introduction, or processes related to the default network or ToM or memory? Again, it remains unclear what we learn from the models using these neural patterns if it is unclear what these neural patterns actually represent and whether and how they relate to guilt and obligation that in the Introduction are presented as major drivers of indebtedness.

3) In reply to my third point (use of potentially biased self-reports) the authors state that the fMRI Study 3 overcomes the problems associated with self reports, because this study allows to objectively measure these constructs using brain activity. This makes the robustness and the interpretation of the neuroimaging results even more important. I thank the authors for highlighting that they used cluster-based FWE correction (which was actually well described in the original version of the manuscript), but I would still like to see the results based on voxel-based FWE correction, i.e., a more stringent statistical threshold that may also specify the network of involved brain regions. The papers the authors cite in their reply to this point emphasize the benefits of multivariate analyses (and I agree), but do not argue against applying voxel-based FWE correction on results derived from univariate analyses.

4) In their reply to my minor point 5, the authors state that "the implementation of our computational model does not require self-report." I acknowledge the authors' effort to validate their model with different types of data, but two of their three experiments contained self-reports, which, in Study 2, also entered the model along with the behavioral data. Using self-reports along with other data is no problem, but the claim that the model can do without these data seems too strong, especially because I am still not convinced about the validity and interpretation of the neuroimaging data (see my comments above).

Reviewer #3:

Remarks to the Author:

We thank the authors for their thorough and thoughtful revision.

We have just one minor concern.

In their original work, the coefficient of $U_{communal}$ could vary between $[-1,1]$ because it was covering both guilt and gratitude. In the revision, the authors wrote the $U_{communal}$ as a linear combination of U_{guilt} and $U_{gratitude}$ (page 22), and they said they will set the weight on $U_{gratitude}$ to zero because they want to study guilt in this paper. Therefore, they will have $U_{communal} = U_{guilt}$.

Based on pages 44, 45 and 46 in the main manuscript, they replaced the Ucommunal by Uguilt (by setting the coefficient of Ugratitude to zero), however it seems that in their accept/reject decision model fitting they considered Phi (the weight over Ucommunal=Uguilt) to vary in the range [-1,1] (also shown in Fig S6.D).

These two choices seem incompatible, unless we're missing something.

It is possible that the least bad choice here is describing treating gratitude and guilt separately as a future direction to pursue.

Response to reviews

Reviewer #2 (Remarks to the Author):

I thank the authors for revising the paper. The manuscript has improved, but unfortunately not all of my concerns have been resolved.

1) In response to my first point (conceptual clarity) the authors claim that “both guilt and obligation are feelings/emotions ...”. The cited literature suggests that guilt and obligation play a role in indebtedness, but “obligation” is not classified as an emotion. Instead, most researchers agree that obligations derive from norms that were established to maintain cooperative societies (see e.g., W Tomasello M. (2020). *The moral psychology of obligation. Behavioral and Brain Sciences*43). I am sorry to say that the conceptual foundations of the manuscript are still unclear. As a result, the modelling results are hard to interpret and it is hard to say how the presented models broaden our understanding of indebtedness.

Response: We strongly disagree with the reviewer about the novelty and importance of the conceptual foundations of this manuscript. In our previous work, we have argued that social emotions such as guilt and anger are the critical psychological mechanism for why we are motivated to adhere to and enforce social norms which in turn facilitate a cooperative society (Chang et al. 2011; Xiang et al. 2013; Chang and Sanfey 2013; Chang and Smith 2015; Chang and Jolly 2018; FeldmanHall and Chang 2018). The opinion paper cited by the reviewer (Tomasello 2019) provides a broad overview of the role of obligation in cooperation from a philosophical, evolutionary, and developmental psychology perspective. Tomasello’s perspective on “obligation” is entirely consistent with what we term “the obligation component of indebtedness” in the present manuscript. In his words, “*the human sense of obligation is best considered as a kind of self-conscious motivation, analogous to the self-conscious emotion of guilt. And, indeed, obligation is intimately related to guilt, as most often guilt is about not living up to one’s obligations.*” (pg 12). We believe our paper further clarifies Tomasello’s vague assertions with a mechanistic model grounded in social & cognitive psychology, economics, and brain science. In fact, many of issues raised with Tomasello’s argument in the published commentary accompanying his paper are directly addressed in the present manuscript, including: (1) that his perspective fails to clearly articulate the difference between communal concern and exchange relationships (Clark, Earp, & Crockett, 2020), (2) that he fails to acknowledge the role of affect and specifically gratitude in his perspective (Beeler-Duden, Yucel, & Vaish, 2020), and (3) that feelings are the critical missing aspect of his argument for actually making his opinion useful for predicting behavior (Rotella, Sparks, Barclay, 2020).

The present manuscript proposes a precise mechanistic model of the psychological processes involved in producing a behavioral response after receiving a favor or gift, which refines the more ethereal argument proposed by Tomasello (2020). We outline both a conceptual and computational version of our argument and validate each assumption of the model, providing a highly comprehensive and thorough empirical paper. Therefore, the present manuscript provides novel contributions to the understanding of indebtedness and obligation both theoretically and empirically.

2) The authors use meta-analytic decoding to interpret their imaging results. They report that “communal feelings were related activity that was similar to the reciprocity results, but was additionally associated with Default mode term. Obligation activity was highly associated with terms related to “Social,” Theory of mind (ToM), and Memory. These interpretations introduce yet different concepts. Are we looking at emotions as claimed in the introduction, or processes related to the default network or ToM or memory? Again, it remains unclear what we learn from the models using these neural patterns if it is unclear what these neural patterns actually represent and whether and how they relate to guilt and obligation that in the Introduction are presented as major drivers of indebtedness.

Response: We believe this comment is due to a misunderstanding of meta-analytic decoding, a technique that we developed almost 10 years ago and has been widely used in neuroimaging studies and cited over 800 times (Chang et al. 2013). The goal of our univariate neuroimaging analyses was to identify unique brain representations associated with each of the psychological processes captured by our computational model. This provides an additional layer of validation of the model beyond directly predicting participant’s behavior and accounting for this post-experiment self-reported ratings. Our model makes trial-to-trial predictions about each participant’s level of communal concern, which we believe reflect perceptions of how much a participant believes their co-player cares for them and its associated feelings of either guilt or gratitude. We find that this model-derived metric is associated with increased activation with the vmPFC, PCC, anterior insula, and DLPFC, regions which have all been previously associated with previous studies of guilt and gratitude. We also used our computational model to make trial-to-trial predictions about the degree to which the participant believes their co-player expected them to reciprocate money and the associated feeling of obligation. For this analysis, we observe increased activity in the dmPFC and TPJ, regions previously reported to be involved in mentalizing or representing another person’s mental state. The results from these univariate analyses provide additional validation of the psychological processes we purport to capture by our computational model.

However, this type of interpretative endeavor relies on reverse-inference and is a highly problematic logical fallacy associated with interpreting neuroimaging results (Poldrack 2011; Poldrack 2006). The reason why this inference is problematic is because voxel activations are typically cherry picked from many clusters of possible voxel activations and researchers tend to only pick the papers that are consistent with their preferred interpretation of the results rather than a comprehensive review of all published results of those regions. Instead, to aid in interpreting our univariate fMRI results, we employ meta-analytic decoding using the Neurosynth database (<http://neurosynth.org>). This analysis uses the results from over 10,000 published studies to identify psychological topics that are most likely implied by the entire pattern of results based on a comprehensive view of the published literature rather than cherry picking any single voxel or paper. The topics are determined by performing a latent variable analysis of the frequently used words in all 10,000 papers. This provides an unbiased approach to interpret results, *but is limited to the granularity of the available topics*. The reviewer appears concerned by why we are introducing “new concepts” into the paper, yet all we are doing is providing a quantitative approach to interpret our data in an unbiased fashion. The meta-analytic decoding analysis demonstrates that the communal concern imaging results are most associated with findings from papers that talk about the so-called “default-mode” network, while the obligation processing results are most associated with theory of mind and social processing. The default-mode network is thought to play a critical role in appraising value and the endogenous experience of feelings (Chang et al. 2021; Ashar et al. 2017; Roy et al. 2012), while mentalizing is a key psychological process involved in inferring another’s expectations. These are precisely the processes that are being modeled in our utility function for each of these processes. It is important to note that the

Neurosynth database has limited granularity of psychological topics. There are no topics of guilt, indebtedness, obligation or gratitude available in the Neurosynth database precisely because these concepts have not been systematically studied from a neuroscientific perspective, *which is why we believe our findings to be so novel*.

To improve the readability of our paper, we are open to moving these results to the supplementary materials to reduce the methodological complexity of our paper pending the recommendation of the reviewers and the editor.

3) In reply to my third point (use of potentially biased self-reports) the authors state that the fMRI Study 3 overcomes the problems associated with self reports, because this study allows to objectively measure these constructs using brain activity. This makes the robustness and the interpretation of the neuroimaging results even more important. I thank the authors for highlighting that they used cluster-based FWE correction (which was actually well described in the original version of the manuscript), but I would still like to the results based on voxel-based FWE correction, i.e., a more stringent statistical threshold that may also specify the network of involved brain regions. The papers the authors cite in their reply to this point emphasize the benefits of multivariate analyses (and I agree), but do not argue against applying voxel-based FWE correction on results derived from univariate analyses.

Response: We would like to clarify that both Study 2 and Study 3 use the same experimental design and we were able to validate each component of our model across both studies using the post-experiment subjective ratings. This is already a much more rigorous validation of a computational model than is common in the field. However, the neuroimaging results provide an even further validation of the processes accounted for by our computational model.

The reviewer appears adamant about us using a specific type of correction for the univariate analyses. However, we decided to use cluster-based FWE correction for the following reasons. In standard univariate neuroimaging analyses, it is important to control for the number of analyses to minimize the likelihood of falsely rejecting a null-hypothesis test at the voxel level (i.e., false positive). The voxel-based familywise error correction suggested by the reviewer, also known as “Bonferroni correction”, requires independently controlling for 328k voxel level hypothesis tests and is rarely used in whole-brain neuroimaging studies published over the past 20 years because it is overly conservative and does not accurately control for false positives when the independence assumption is violated (Nichols and Hayasaka 2003). To make this very concrete, according to the reviewer’s proposed correction, we would only be able to consider a voxel as being statistically significant at the alpha level of $p < 0.05$ if our results of our *t*-test exceeded a p-value threshold of 0.00000015 (i.e., $0.05/328,000$), which is widely believed to be too conservative (Nichols and Hayasaka 2003). We spatially smoothed our data as a preprocessing step and we know that the brain is functionally organized in regions larger than a single voxel, which violates the independence assumption of using Bonferroni correction. Instead, there are statistics that attempt to model the family-wise error to account for this spatial autocorrelation using Gaussian Random Field Theory and is precisely the approach that we employ to correct for multiple comparisons in our univariate analyses. There are other approaches that make different assumptions (e.g., FDR), but the cluster correction method is the most common method used in the field in approximately 80% of all published neuroimaging papers (Woo et al. 2014). We fully acknowledge that cluster correction has its own problems, but we believe these issues do not impact the interpretation of our results. It is important to note that our results do not hinge on this univariate analysis or multiple comparisons correction procedure. In fact, we include multiple types of analyses to

provide converging evidence and also to minimize potential methodological shortcomings and investigator bias (e.g., meta-analytic decoding & multivariate pattern analysis).

4) In their reply to my minor point 5, the authors state that “the implementation of our computational model does not require self-report.” I acknowledge the authors’ effort to validate their model with different types of data, but two of their three experiments contained self-reports, which, in Study 2, also entered the model along with the behavioral data. Using self-reports along with other data is no problem, but the claim that the model can do without these data seems too strong, especially because I am still not convinced about the validity and interpretation of the neuroimaging data (see my comments above).

Response: Although we have provided a lengthy response to this comment in the last round of review and have attempted to clarify our model in the manuscript, it seems that the reviewer continues to misunderstand our methods. To be clear, our computational model *does not* use any of the self-report measures to make predictions about the constructs it models. It uses the experimental design, several free parameters, and the hypothetical cognitive processes derived from our theoretical model to make quantitative predictions about how much the participants choose to reciprocate to the benefactor or whether they want to reject the help. We model the mentalizing process that we believe the participants undergo in the task, which produces latent feelings (e.g., guilt or obligation) that ultimately produce the behavioral actions we observe in the game (Studies 2 & 3). We collect self-reported feelings and appraisals *after* the experiment and use this data to *validate* that our model is accurately capturing the subjective appraisals and feelings based on our mathematical operationalization. In other words, if future studies only use our interactive task without collecting self-reported feelings, they can still apply our computational models to estimate each participant’s weight on communal concern and obligation during reciprocal behaviors upon receiving favors. We fully acknowledge that this is a dense and methodologically sophisticated paper. Beyond our mathematical notation and experimental design, we have also shared our analysis code (https://github.com/xiaoxuepsy/Indebtedness_Gao2021), which may help the reviewer better understand our computational modeling methods. We hope that clarifying that our model does not require or use any self-report data may help the reviewer appreciate the novelty and contribution of our work.

References

- Ashar, Y. K., Chang, L. J., & Wager, T. D. (2017). Brain Mechanisms of the Placebo Effect: An Affective Appraisal Account. *Annual Review of Clinical Psychology, 13*, 73–98.
- Beeler-Duden, S., Yucel, M., & Vaish, A. (2020). The role of affect in feelings of obligation [Review of *The role of affect in feelings of obligation*]. *The Behavioral and Brain Sciences, 43*, e60. academia.edu.
- Chang, L. J., & Jolly, E. (2018). Emotions as computational signals of goal error. In A. Fox, R. Lapate, A. Shackman, & R. Davidson (Eds.), *The nature of emotion: Fundamental questions* (pp. 343–348). Oxford University Press.
- Chang, L. J., Jolly, E., Cheong, J. H., Rapuano, K. M., Greenstein, N., Chen, P.-H. A., & Manning, J. R. (2021). Endogenous variation in ventromedial prefrontal cortex state dynamics during naturalistic viewing reflects affective experience. *Science Advances, 7*(17). <https://doi.org/10.1126/sciadv.abf7129>

- Chang, L. J., & Sanfey, A. G. (2013). Great expectations: neural computations underlying the use of social norms in decision-making. *Social Cognitive and Affective Neuroscience*, 8(3), 277–284.
- Chang, L. J., & Smith, A. (2015). Social emotions and psychological games. *Current Opinion in Behavioral Sciences*, 5, 133–140.
- Chang, L. J., Smith, A., Dufwenberg, M., & Sanfey, A. G. (2011). Triangulating the neural, psychological, and economic bases of guilt aversion. *Neuron*, 70(3), 560–572.
- Chang, L. J., Yarkoni, T., Khaw, M. W., & Sanfey, A. G. (2013). Decoding the role of the insula in human cognition: functional parcellation and large-scale reverse inference. *Cerebral Cortex*, 23(3), 739–749.
- Clark, M. S., Earp, B. D., & Crockett, M. J. (2020). Who are “we” and why are we cooperating? Insights from social psychology [Review of *Who are “we” and why are we cooperating? Insights from social psychology*]. *The Behavioral and Brain Sciences*, 43(e66), e66. Cambridge University Press (CUP).
- FeldmanHall, O., & Chang, L. J. (2018). Social Learning: Emotions aid in optimizing goal-directed social behavior. *Goal-Directed Decision Making*.
<https://www.sciencedirect.com/science/article/pii/B9780128120989000140>
- Nichols, T., & Hayasaka, S. (2003). Controlling the familywise error rate in functional neuroimaging: a comparative review. *Statistical Methods in Medical Research*, 12(5), 419–446.
- Poldrack, R. A. (2006). Can cognitive processes be inferred from neuroimaging data? *Trends in Cognitive Sciences*, 10(2), 59–63.
- Poldrack, R. A. (2011). Inferring mental states from neuroimaging data: from reverse inference to large-scale decoding. *Neuron*, 72(5), 692–697.
- Rotella, A., Sparks, A. M., & Barclay, P. (2020). Feelings of obligation are valuations of signaling-mediated social payoffs [Review of *Feelings of obligation are valuations of signaling-mediated social payoffs*]. *The Behavioral and Brain Sciences*, 43, e85. eprints.kingston.ac.uk.
- Roy, M., Shohamy, D., & Wager, T. D. (2012). Ventromedial prefrontal-subcortical systems and the generation of affective meaning. *Trends in Cognitive Sciences*, 16(3), 147–156.
- Tomasello, M. (2020). The moral psychology of obligation [Review of *The moral psychology of obligation*]. *The Behavioral and Brain Sciences*, 43, e56. cambridge.org.
- Woo, C.-W., Krishnan, A., & Wager, T. D. (2014). Cluster-extent based thresholding in fMRI analyses: pitfalls and recommendations. *NeuroImage*, 91, 412–419.
- Xiang, T., Lohrenz, T., & Montague, P. R. (2013). Computational substrates of norms and their violations during social exchange. *The Journal of Neuroscience: The Official Journal of the Society for Neuroscience*, 33(3), 1099–1108a.

Reviewer #3 (Remarks to the Author):

We thank the authors for their thorough and thoughtful revision.

We have just one minor concern.

In their original work, the coefficient of $U_{communal}$ could vary between $[-1,1]$ because it was covering both guilt and gratitude. In the revision, the authors wrote the $U_{communal}$ as a linear combination of U_{guilt} and $U_{gratitude}$ (page 22), and they said they will set the weight on $U_{gratitude}$ to zero because they want to study guilt in this paper. Therefore, they will have $U_{communal}=U_{guilt}$.

Based on pages 44, 45 and 46 in the main manuscript, they replaced the $U_{communal}$ by U_{guilt} (by setting the coefficient of $U_{gratitude}$ to zero), however it seems that in their accept/reject decision model fitting they considered Φ (the weight over $U_{communal}=U_{guilt}$) to vary in the range $[-1,1]$ (also shown in Fig S6.D).

These two choices seem incompatible, unless we're missing something.

It is possible that the least bad choice here is describing treating gratitude and guilt separately as a future direction to pursue.

Response: We thank the reviewer for the careful observation and deep understanding of the model presented in this paper. This discrepancy that the reviewer is raising is a very minor point and has to do with slightly different implementations of the model for predicting the amount of money being returned compared to predicting the decision to accept or reject help. Due to nuances in the formulation of the model, unlike the reciprocity model (Eq. 1), the accept/reject model (Eq. 10) is non-identifiable across the parameter range $\phi_B = [0,1]$. This means that there are multiple combinations of the parameters that will predict the same behavior. To address this issue and also to reflect the predictions of the model that indebtedness will lead to rejecting help while gratitude will lead to accepting help (See Figure 1 & 2), we provide a slightly different formulation of the model, in which we carve up the parameter space of ϕ_B to increase the identifiability of the model. Values greater than 0 indicate a motivation of gratitude, while values less than 0 indicate a motivation of guilt. By dividing up the parameter space in this manner, we are able to increase the identifiability of the model and estimate the parameters. However, we note in the paper that approximately 50% of the parameter space will still be unidentifiable, which is precisely reflected in the parameter recovery analyses ($r = 0.43 \pm 0.40$, $p < 0.001$). This is not a key point in the paper and is something we are planning to address in future work, which is well beyond the scope of the current paper and will require a new experimental design.

We explain this issue in more detail in the methods section on pages 44-45:

“In this model, $U(\text{Reject})$ was set to zero, because the participant’s emotional responses would not change if the participant did not accept help. Increased obligation reduces the likelihood of accepting help to avoid being in the benefactor’s debt^{13,14,113}. In contrast, $U_{Communal}$ has a more complex influence on behavior, with guilt decreasing the likelihood of accepting help to avoid burdening a benefactor^{34,54}, and gratitude motivating accepting help to build a communal relationship^{6,7}. However, because $U_{Communal} = U_{Guilt} = U_{Gratitude} = \omega_B$ in this formulation, there is no variability in the design for the model to be able to disentangle the effect of gratitude from that of guilt. To address this complexity, we

constrain ϕ to be within the interval of $[-1, 1]$, and explicitly divide up the parameter space such that $\phi > 0$ indicates a preference for gratitude and motives the participants to accept the help, while $\phi < 0$ indicates a preference for guilt and motives the participants to reject the help.

$$\begin{cases} \phi_B > 0 & \textit{Gratitude} \\ \phi_B < 0 & \textit{Guilt} \end{cases} \quad \textbf{Eq. 11}$$

Regardless of whether the participant is motivated primarily by guilt or gratitude, participants can still have a mixture of obligation captured by $1 - |\phi|$, which ranges from $[0, 1]$. Unfortunately, if participants are equally sensitive to gratitude and guilt, ϕ will reduce to zero and the weight on obligation increases, which decreases the model fit and leads to some instability in the parameters (see *Results and Discussion*). ”

Reviewers' Comments:

Reviewer #2:

Remarks to the Author:

In this second, substantial revision the authors present the refined and more detailed conceptual framework I was asking for, and clarify how their modelling approach may advance our understanding of indebtedness. The paper is now more accessible for a broader readership, because the authors make an effort to explain the rationale behind the analyses (including meta-analytic decoding that is familiar to me, but may not be familiar to other readers). So overall, the paper has much improved.

However, there is still one point that needs to be clarified. In the previous version of the manuscript, the performance of the utility model was worse than the performance of the control model (see Results below), which was one of the reasons why I was concerned about the validity of the "neural utility model". Checking this part of the Results section again in the current version, I found that the performance of the utility model now "numerically outperformed" (cited from line 639 of the Discussion section) the control model (see Results below). Given that the analyses were performed with the same data, this is unsettling and I would like the authors to explain this change in results.

Previous Results section, line 420: "We were able to reliably predict reciprocity behavior with our computational model informed only by predictions of communal and obligation feelings derived purely from brain responses (average $r = 0.10 \pm 0.01$, fisher- $z = 0.10 \pm 0.01$, permutation $p = 0.013$, AIC = 324.04 ± 4.93). [...] As a benchmark, this model performed slightly worse than our overall ability to directly predict reciprocity behavior from multivariate patterns of brain activity (Fig. 6A, reciprocity pattern: average $r = 0.18 \pm 0.03$, fisher- $z = 0.17 \pm 0.03$, permutation $p < 0.001$, AIC = 321.07 ± 4.81 ; paired t test for AIC, $t_{52} = 5.26$, $p < 0.001$) [...]"

Current Results section, line 514: We were able to reliably predict reciprocity behavior with our computational model informed only by predictions of communal and obligation feelings derived purely from brain responses (average $r = 0.19 \pm 0.02$, $p < 0.001$, AIC = 317.70 ± 5.00). As a benchmark, this model numerically outperformed a whole-brain model trained to directly predict reciprocity (average $r = 0.18 \pm 0.03$, $p < 0.001$, AIC = 317.54 ± 5.00 ; Fig. 7A), but this difference only approached statistical significance, $t_{52} = 1.64$..."

Minor comment:

Ruff, C. C. & Fehr, E. The neurobiology of rewards and values in social decision making. *Nat. Rev. Neurosci.* 15, 549-562 (2014).

is cited two times with a different number (118 and 66)

The authors may consider citing a paper from the same group showing that hidden states (motives) can be predicted from patterns of functional connectivity
DOI: 10.1126/science.aac7992

Reviewer #4:

Remarks to the Author:

In this manuscript, the authors address the important point of how people value, perceive, and react to help by others. In particular, the authors try to show and numerically describe the influence of a number of concepts (including guilt, indebtedness, beliefs of expected reciprocity, etc.). To do so, the authors use three studies: 1) an online study in which participants were asked about a real-life event, in which they received (or rejected) help, 2) a behavioral study, in which participants were led to believe that interaction partners could spend money to reduce the pain administered to the participants, and 3) an fMRI study with the same design as study 2. The authors used an impressive multitude of analytic approaches (including regressions, factor analyses, comparison of specific mathematical models, GLMs for fMRI data, meta-analytical decoding, etc.).

In my view, this manuscript provides an interesting contribution to the current literature. Overall, the experiments and the analyses seem well-conducted. I am in favor of seeing this manuscript published. However, I have to admit that while reading I had to struggle quite a bit because the

relevant concepts, models, methods, and results were not always clear to me (especially in the main text).

Please find my suggestions below. They are rather lengthy. This does not necessarily mean that I have a lot to criticize, often I am rather concerned with the clarity of the manuscript and I wanted to be as specific as possible.

Major:

1. Concepts & clarity regarding the components of the models: I really appreciate that the authors aim to give mathematical precision to many concepts/terms that are mentioned in everyday life by including these terms as (free) parameters in their model. The overall idea was easy to follow but I found it difficult to get the details: The following suggestions could help:
 - a. You can include all components of Eq. 1 in Fig. 1; especially "self-interest" and "greed" are missing.
 - b. "Gratitude" is included in Fig. 1, assessed in all studies, included in several analyses, and mentioned at many points in the manuscript. But "gratitude" is not included in the models (even though the authors also show how it would be included in their models). The authors say explicitly that they focus on guilt rather than gratitude but they never say why.
 - c. Indebtedness seems to be an umbrella term for guilt and sense of obligation (e.g., in Fig 1). However, in Fig. 2 and the regression analyses shown in these figures all three terms seem to have the same status (along with gratitude). Also, in Studies 2 & 3 ratings for all four terms were given. Please explain and motivate this discrepancy in the use of the three terms.
 - d. Is "guilt" a component of "communal concern" and of "indebtedness?"
 - e. Was the conceptual model developed before the results of Study 1 were known or after? Did Study 1 refine the model(s)? If model 1 cannot be directly used to test the model(s), why was it included here at all? I think that you can easily address this point with a few more sentences that motivate the transition between Study 1 and Study 2.

2. Details of the tasks are missing

3. Model:

- a. When I first saw Eq. 1, I was wondering whether this "computational model" is actually not "just" a regression. After all, one could just use three free parameters for weighing π_B , $U_{Communal}$, and $U_{Obligation}$ (plus an intercept). I do see why the authors use free parameters, which are restricted between 0 and 1, to balance the relative contribution of the three input parameters but I still wonder whether this overcomplicates things unnecessarily (especially because ϕ_B is nested within θ_B). As far as I can see, the authors did individually parameterize ϕ_B in model 1.5 but they did not test the rather straightforward model suggested above.
- b. The models listed with a-d in the main text (lines 382-387) should be labelled in the same way as the detailed explanations of the models in the Supplementary Materials.
- c. Model comparisons should be mentioned more explicitly in the main text (not just in the Supplementary Materials).
- d. When reading the main text, I was really confused about the relationship between Eq. 1 and Eq. 2/3. None of the parameters in Eq. 2/3 actually appears in Eq. 1. This only becomes clear in the methods but even then one needs to follow up how the different parameters relate to each other. At some point, Eq. 1 needs to be spelled out fully and the number of free parameters needs to be made evident.
- e. Related to the previous point, I was really surprised to read in the methods that U_{Guilt} and $U_{Obligation}$ are defined differently for the "reciprocity" and the "accept/reject" models.
- f. The previous point relates to one of main points. The authors write: "To validate our computational model, we tested whether it accurately captured each proposed component process of our conceptual model and successfully predicted participant's behavior" - Which type of behavior? The ratings? The allocation decisions? The rejection rate? It was initially completely unclear to me on which part of the participants' data the model was fitted. The authors go on to report correlations but testing for correlations are not how models are usually fitted or evaluated. From reading the methods, I gather that the authors fitted "reciprocity" and "accept/reject" decisions separately. In any case, this should be much clearer in the results section.
- g. The authors never explain their definition of U_{Guilt} and $U_{Obligation}$ (in Eq. 7 & 8). Maybe I missed it but this is completely intransparent to me. Why are the differences squared? Also, ω_B , which is included in U_{Guilt} depends on E_B and at the same time E_B appears in $U_{Obligation}$. Why?
- h. Eq. 2 & 3: It also seems confusing to me that ω_B has D_A twice in the numerator. E_B simply is D_A for the "repayment possible" condition. The authors write: "This creates a nonlinear relationship between ω_B and E_B such that the relationship is negative when κ is high, positive when κ is low, and uncorrelated in the current dataset with $\kappa = 0.32$ [...]." Doesn't this imply that "high" means above 1 and "low" means between 0 and 1. At $\kappa_B = 1$ the term D_A

disappears for the "repayment impossible" condition.

i. In the results, the authors report "that each term of our model was able to accurately capture trial-to-trial variations in self-reported appraisals of second-order belief of the benefactor's expectation for repayment [...] and perceived care [...]." What does "each term" mean here? Doesn't this mean that these terms are not separable?

j. Okay, after writing all these points above, I realized that these models use D_A and D_B as input to determine the utility of D_B (or better some form of D_A and D_B that are normalized by the money to be allocated). Is this correct? For the "accept/reject" models D_B is not available because if participants reject, D_B cannot be indicated. If I understand this correctly, then I am even more confused because then the objective function (Eq. 8) includes the difference between D_B and the utility of D_B ... but then the utility of D_B depends on D_B . Sorry, I am a bit lost here.

4. Study 1:

a. When reading the manuscript I had the following questions in mind: Can the different events for which participants received help be clustered? How severe were these events (e.g., help when moving to a new apartment vs. help during a period of sickness)? How many events entailed pain, i.e., could some events be conceptually related to Study 2? Was the received help truly helpful for resolving the challenges posed by the events? How many did participants report on average ... or were they asked to provide a specific number of events? ... Appendix S1 shows that no questions about the specific events were asked. This should be mentioned in the main text. On the other hand, it would be really helpful to know which of the questions from Appendix S1 were used for the analyses in which were not.

b. Were regressors normalized? The beta values do suggest this.

5. Study 2:

a. Please specify in the number of trials with the two repayment conditions. How many benefactors were there? I could only find answers to these two questions in Table S2.

b. Did participants believe that the benefactors were in a neighboring room? It's clear from the methods but could be very briefly mentioned in the results.

c. The differences between study 2a & 2b could be mentioned more explicitly in 1-2 additional sentences in the main text.

d. How was perceived care assessed? Were participants always asked for the other person's expectations AND for perceived care? Or is perceived care "simply" equal to ω_B and thus derived from D_A and E_B ?

e. What components of the interactive tasks were shown for the post-task rating? The repayment condition, the spent money, the participant's expectation, and their decisions (to accept and to allocate)? Were the four ratings always given in the same order?

f. Fig 4B & C: Are these the averages of individual regression coefficients (i.e., data points for all trials per participant) or of one regression (i.e. data points for averages across participants)?

After reading the methods. I think that these are derived from mixed effects regressions (lmer in R). Correct?

g. In the section "Behavioral responses to help are influenced by the benefactor's intentions," did the authors compare the magnitudes of the beta coefficients? It seems to me that they just report the magnitude in the text but not simple t-tests (or contrasts) between the betas for the "repayment possible" and the "repayment impossible" conditions. Fig 5B only shows the rejection rate but not the "reciprocity as a function of the amount of received help."

h. Fig 5E: What is the unit on the axis for "predicted reciprocity?" Is it the percentage? If so, the maximum percentage across this parameter space is below 16% but in Fig 5C the values exceed 16%.

6. Study 3:

a. fMRI: Why three separate GLMs? Shouldn't it be one GLM with (1) reciprocity, (2) communal concern, and (3) obligation as parametric modulators (with or without orthogonalization depending on the question asked)?

b. Which time points were used? This should be clear from reading the results.

c. Otherwise, the neural utility model was much clearer to me than the behavioral model.

Minor:

1. The first sentence of the abstract is "Receiving help or a favor from another person can sometimes have a hidden cost." This may be nitty-gritty but I would add "... for the recipient/beneficiary"

2. The abstract could be a bit more specific; in particular, it would be nice to know immediately how many different types of tasks the authors used.

3. Related to the previous comment: I appreciate that the introduction is quite general. Nevertheless, it would be nice to get an idea of the used tasks (i.e., the situations in which help can be received such as a reduction of pain as in Studies 2 & 3).

4. It is an open question how much the current results generalize to different situations, in which help can be received. This should be mentioned in 1-2 sentences in the discussion.
5. It is unclear what the Tables S3 and S4 show. What is the general verbal conclusion?
6. In the discussion, the relationship to game theory is rather indirect. The task may bear some similarities to the trust game (but even these rather loose connections are not explicitly mentioned).

Response to reviews

We thank the reviewers for their constructive suggestions, which we believed have helped us to greatly improve the clarity of the manuscript. Below we detail our responses to each of the reviewers' points. The revised parts are highlighted in blue in the Manuscript File. The reference numbers are consistent with those in the main text.

Reviewer #2

Comments:

In this second, substantial revision the authors present the refined and more detailed conceptual framework I was asking for, and clarify how their modelling approach may advance our understanding of indebtedness. The paper is now more accessible for a broader readership, because the authors make an effort to explain the rationale behind the analyses (including meta-analytic decoding that is familiar to me, but may not be familiar to other readers). So overall, the paper has much improved.

Response: We are grateful for your careful reading and thoughtful feedback, which have helped us to make the paper more accessible for a broader readership.

However, there is still one point that needs to be clarified. In the previous version of the manuscript, the performance of the utility model was worse than the performance of the control model (see Results below), which was one of the reasons why I was concerned about the validity of the “neural utility model”. Checking this part of the Results section again in the current version, I found that the performance of the utility model now “numerically outperformed” (cited from line 639 of the Discussion section) the control model (see Results below). Given that the analyses were performed with the same data, this is unsettling and I would like the authors to explain this change in results.

Previous Results section, line 420: “We were able to reliably predict reciprocity behavior with our computational model informed only by predictions of communal and obligation feelings derived purely from brain responses (average $r = 0.10 \pm 0.01$, fisher-z = 0.10 ± 0.01 , permutation $p = 0.013$, AIC = 324.04 ± 4.93). [...] As a benchmark, this model performed slightly worse than our overall ability to directly predict reciprocity behavior from multivariate patterns of brain activity (Fig. 6A, reciprocity pattern: average $r = 0.18 \pm 0.03$, fisher-z = 0.17 ± 0.03 , permutation $p < 0.001$, AIC = 321.07 ± 4.81 ; paired t test for AIC, $t_{52} = 5.26$, $p < 0.001$) [...]

Current Results section, line 514: We were able to reliably predict reciprocity behavior with our computational model informed only by predictions of communal

and obligation feelings derived purely from brain responses (average $r = 0.19 \pm 0.02$, $p < 0.001$, $AIC = 317.70 \pm 5.00$). As a benchmark, this model numerically outperformed a whole-brain model trained to directly predict reciprocity (average $r = 0.18 \pm 0.03$, $p < 0.001$, $AIC = 317.54 \pm 5.00$; Fig. 7A), but this difference only approached statistical significance, $t_{52} = 1.64$...”

Response: We thank the reviewer for this observation and greatly appreciate their attention to detail throughout the review process. We made substantial changes to the neural utility model section based on comments from several previous reviewers (particularly Reviewer #3) and re-ran all of our analyses. Through this process, we discovered that we were using different methods to evaluate the reciprocity brain prediction model and our neural utility model. In the previous version, after estimating the model parameters, we evaluated the performance of neural utility model by combining the data of all participants and using a linear mixed model with participant as a random intercept and slope¹⁰⁶. This approach inadvertently added some additional parameters to the model and also likely resulted in some statistical shrinkage in the overall model performance. In our revised manuscript, we also made sure that we evaluated the neural utility model within each participant as we did for training and testing the reciprocity brain prediction model. This is cleaner as it ensures that we are using the same trials to test both models for each participant. We were pleased to see that this procedure actually improved the performance of our neural utility model to be comparable to the best we could do predicting behavior directly from brain activity. We have shared our analysis code of the neural utility model on github (https://github.com/xiaoxuepsy/Indebtedness_Gao2021).

Minor comment:

Ruff, C. C. & Fehr, E. The neurobiology of rewards and values in social decision making. Nat. Rev. Neurosci. 15, 549-562 (2014).

is cited two times with a different number (118 and 66)

Response: We thank the reviewer for this careful observation and have replaced the previous Ref. 118 with Ref. 66 to avoid duplication (p. 56).

*The authors may consider citing a paper from the same group showing that hidden states (motives) can be predicted from patterns of functional connectivity
DOI: 10.1126/science.aac7992*

Response: As suggested, we have added this paper in the Discussion as Ref. 91 (p. 27):

“We provide an even stronger test of our ability to characterize the neural processes associated with indebtedness by deriving a “neural utility” model. Previous work has demonstrated that it is possible to build brain models of preferences that can predict behaviors^{89,90} and the hidden motives behind the behaviors⁹¹.”

Reviewer #4

Comments:

In this manuscript, the authors address the important point of how people value, perceive, and react to help by others. In particular, the authors try to show and numerically describe the influence of a number of concepts (including guilt, indebtedness, beliefs of expected reciprocity, etc.). To do so, the authors use three studies: 1) an online study in which participants were asked about a real-life event, in which they received (or rejected) help, 2) a behavioral study, in which participants were led to believe that interaction partners could spend money to reduce the pain administered to the participants, and 3) an fMRI study with the same design as study 2. The authors used an impressive multitude of analytic approaches (including regressions, factor analyses, comparison of specific mathematical models, GLMs for fMRI data, meta-analytical decoding, etc.).

In my view, this manuscript provides an interesting contribution to the current literature. Overall, the experiments and the analyses seem well-conducted. I am in favor of seeing this manuscript published. However, I have to admit that while reading I had to struggle quite a bit because the relevant concepts, models, methods, and results were not always clear to me (especially in the main text).

Please find my suggestions below. They are rather lengthy. This does not necessarily mean that I have a lot to criticize, often I am rather concerned with the clarity of the manuscript and I wanted to be as specific as possible.

Response: We thank the reviewer for their constructive comments, which we believe has helped in improving our paper. As the reviewer points out, we have included a number of datasets and a multitude of analyses to support our argument. This has resulted in difficulty finding the best balance between providing sufficient details of the methods while still maintaining readability. Based on feedback from previous reviewers, we opted to move many of the details from the main text to the *Methods* section and the *Supplementary Information*. As suggested, we have now attempted to further clarify these details beyond the specific points highlighted by the reviewer throughout the manuscript, primarily including the details of the interpersonal task,

linear mixed model, computational modeling, and GLM, among others, in order to make our methods more accessible to readers. To be noted, due to the word limit in the main text (up to 6000 words required by the journal), we could only present the main analyses and corresponding results in the *Results* section, and thus include the methodological and analytical details in the *Methods* section, especially for the computational modelling part. Nevertheless, we have made substantial revisions on the *Results* section to enable the reader to better understand our methods and results. Please see detailed responses below.

Major:

1. Concepts & clarity regarding the components of the models: I really appreciate that the authors aim to give mathematical precision to many concepts/terms that are mentioned in everyday life by including these terms as (free) parameters in their model. The overall idea was easy to follow but I found it difficult to get the details: The following suggestions could help:

a. You can include all components of Eq. 1 in Fig. 1; especially “self-interest” and “greed” are missing.

Response: We appreciate the reviewer’s suggestion and used this as an opportunity to further clarify the differences between the conceptual model of indebtedness and the computational model used to model behaviors in the interpersonal task by substantially restructuring the *Introduction* section (pp. 4 - 6).

Specifically, by proposing a conceptual model, we aim to establish and validate a theoretical framework of indebtedness, which captures how the feelings of indebtedness (guilt and obligation) arise from appraisals and impacts subsequent decision behavior. We also have included gratitude in this conceptual model, because gratitude is a very common feeling in the beneficiary. It is important to make it easy to understand how gratitude and indebtedness relate to each other as prior work has tended to emphasize the importance of gratitude versus indebtedness. Our conceptual model provides a general framework that integrates the results of previous studies on indebtedness and can be applied to understand and investigate indebtedness in various favor-receiving contexts. We then use computational models in Study 2 to model specific decision behaviors in the interpersonal task (e.g., reciprocity & help-acceptance decisions) in order to quantitatively validate our conceptual model.

While self-interest or greed is an important concern for individuals making reciprocal decisions, it frequently appears in various types of decision-making processes and has no direct relationship to the feeling of indebtedness. We have some concerns that adding self-interest into the conceptual model could potentially add further confusion for readers to understand the generation and influences of indebtedness. Though we

appreciated the reviewer's suggestion to more tightly integrate the conceptual model with the computational model, after careful thought and creating such a figure, we have ultimately decided to not add self-interest into Fig. 1.

Instead, we attempted to address this concern with a more substantial refactor of the entire paper. To this end, we have substantially restructured the *Introduction* section (pp. 4 - 6) to introduce the central idea of our conceptual model, and clarify that our computational models are specifically designed to model reciprocity and help-acceptance decisions in our interpersonal task in Study 2. We have also moved the methods of computational model and Eq. 1 from the *Introduction* section to the *Results* section of Study 2 (pp. 15 - 16). In this way, it is also easier for the readers to understand our methods of computational modeling. We hope that this makes everything clearer based on the reviewer's thoughtful suggestion.

b. "Gratitude" is included in Fig. 1, assessed in all studies, included in several analyses, and mentioned at many points in the manuscript. But "gratitude" is not included in the models (even though the authors also show how it would be included in their models). The authors say explicitly that they focus on guilt rather than gratitude but they never say why.

Response: We thank the reviewer for this observation and apologize for any confusion in the methods of computational modeling. Based on the feedback from previous reviewers we have added gratitude to our models and have acknowledged that our interpersonal task was not designed to explicitly differentiate guilt from gratitude (p. 28). The reciprocity model (Model 1.1), in particular, cannot differentiate between the effects of gratitude and guilt, because both positively contribute to reciprocity. Therefore, we collectively call this term "communal concern". The help-acceptance model (Model 2.1) can somewhat differentiate between gratitude and guilt based on dividing up the parameter space such that increased gratitude leads to increased utility when taking action to accept help and guilt leads to increased utility when taking action to reject help.

We agreed with previous reviewers that is important for us to articulate how we believe guilt and gratitude could be incorporated into our modeling framework, but leave it to future work to specifically address the relationship between these two constructs using different experimental designs. Therefore, we use the concept of "communal concern" instead throughout the paper until the relationship between guilt and gratitude can be better differentiated. We realize that this may be inadvertently complicating things and adding unnecessary confusion, but we believe that it is important to be upfront about the complicated nature of modeling emotions so that future work is aware of the issues we encountered in this project. Because so little research has attempted to develop models of social emotions and test them, we feel it

is important to set a strong precedent about not oversimplifying the complexity of this work. We have attempted to clarify this point in multiple places in the revision, for example:

Results section (p. 16)

“The *reciprocity model* (Model 1.1) predicts the participant’s amount of money reciprocated to the benefactor, while the *help-acceptance model* (Model 2.1) predicts binary decisions to accept or reject help. Though the two models are conceptually similar, the values of $U_{Communal}$ and $U_{Obligation}$ are computed slightly differently due to differences in the types of data (i.e., continuous vs. binary decisions) and how appraisals are inferred. It is important to note that in the reciprocity model, we are unable to distinguish between the separate motivations of guilt and gratitude because both positively contribute to reciprocity. In contrast, based on the findings from Study1, we divide up the parameter space for the help-acceptance model such that $\phi > 0$ indicates a preference for gratitude and motives accepting the help, while $\phi < 0$ indicates a preference for guilt and motives rejecting the help.”

Methods section (pp. 48- 51):

“*Predicting reciprocity decisions*

...

Based on our conceptual model (Fig. 1), we defined $U_{Communal}$ as a mixture of feelings of gratitude $U_{Gratitude}$ and guilt U_{Guilt} , in which the parameter δ_B ranged from [0,1] and reflected how much gratitude contributed to communal concern in comparison to guilt.

$$U_{Communal} = \delta_B * U_{Gratitude} + (1 - \delta_B) * U_{Guilt} \quad \text{Eq. 8}$$

We note that both guilt and gratitude positively contribute to reciprocity and our interpersonal task was not designed to explicitly differentiate the effects of guilt and gratitude, which precluded our ability to estimate the specific value of δ_B for predicting reciprocity decisions due to a lack of identifiability. Thus, in this paper we can only make inferences about the broader $U_{Communal}$, which may reflect guilt and/or gratitude. However, our help-acceptance model does attempt to differentiate the contributions of guilt and gratitude to decisions of whether or not to accept help as discussed below.

...

Predicting help-acceptance decisions

...

Increased obligation reduces the likelihood of accepting help to avoid being in the benefactor’s debt^{13,14,114}. In contrast, $U_{Communal}$ has a more nuanced influence on behavior, with guilt decreasing the likelihood of accepting help to avoid burdening a benefactor^{34,54}, and gratitude motivating accepting help to build a communal relationship^{6,7}. However, because $U_{Communal} = U_{Guilt} = U_{Gratitude} = \omega_B$

in this formulation, there is no variability in the design for the model to be able to disentangle the effect of gratitude from that of guilt. To address this complexity, we constrain ϕ to be within the interval of $[-1, 1]$, and explicitly divide up the parameter space such that $\phi > 0$ indicates a preference for gratitude and motives the participants to accept the help, while $\phi < 0$ indicates a preference for guilt and motives the participants to reject the help.

$$\begin{cases} \phi_B > 0 & \textit{Gratitude} \\ \phi_B < 0 & \textit{Guilt} \end{cases} \quad \text{Eq. 12}$$

Regardless of whether the participant is motivated primarily by guilt or gratitude, participants can still have a mixture of obligation captured by $1 - |\phi|$, which ranges from $[0,1]$. Unfortunately, if participants are equally sensitive to gratitude and guilt, ϕ will reduce to zero and the weight on obligation increases, which decreases the model fit and leads to some instability in the parameters.”

Discussion (p. 28):

“Although we observed support for this prediction, our interpersonal task was not designed to explicitly differentiate guilt from gratitude, which limited the ability of our reciprocity model to capture the specific contributions of guilt and gratitude to communal concern and likely impacted identifiability of the parameters of the help-acceptance model. Future work might continue to refine the relationship between these two aspects of communal concern both in terms of behaviors in experiments and computations in models ^{54,62-64,67,68,88}.”

c. Indebtedness seems to be an umbrella term for guilt and sense of obligation (e.g., in Fig 1). However, in Fig. 2 and the regression analyses shown in these figures all three terms seem to have the same status (along with gratitude). Also, in Studies 2 & 3 ratings for all four terms were given. Please explain and motivate this discrepancy in the use of the three terms.

Response: Indebtedness has often been considered a unitary construct, defined singularly as either the feeling of guilt for personally burdening the benefactor ⁴⁰⁻⁴³ or as a sense of obligation to repay ^{13,14,21,46,47}. In the current study, Study 1 empirically demonstrate that indebtedness is a mixed feeling comprised of guilt and obligation based on evidence from (I) emotion ratings in the daily event recalling, (II) attribution of guilt and obligation as source of indebtedness, and (III) topic modeling of the emotional words in self-reported definition of indebtedness (Fig. 2A). These findings refine the understanding on the definition of indebtedness, which is one of the main contributions of the current study. This point has now been explicitly stated in the *Introduction (p. 4)*.

Given that indebtedness is an umbrella term for guilt and sense of obligation, we agree with the reviewer that, we should focus the separate effects of guilt and obligation, rather than the combined effect of indebtedness. That is exactly what we did in the mediation analysis in Study 2 (Fig. 4E) and all the following analyses. However, we still provide the results of indebtedness ratings in Fig. 2B of Study 1, and Fig. 4A and Fig. 4D of Study 2 for the following reasons.

- 1) These results of indebtedness ratings can help further support our findings in Fig. 2A that indebtedness is a mixed feeling comprised of guilt and obligation. For example, the effects of indebtedness on behaviors are consistent with those of guilt and obligation in the regression analysis of Study 1 (Fig. 2B). For factor analysis on ratings of Study 2, indebtedness moderately loaded on both Communal and Obligation factors (Fig. 4).
- 2) Second, by providing these results, readers can compare them to the results of previous studies and understand the relationship between our findings and previous findings on indebtedness^{13,14,21,40-43,46,47}.

We have added sentences into the *Results* section to clarify this point (p. 13):

“The Communal Factor reflected participants' perception that the benefactor cared about their welfare and resulted in emotions of guilt and gratitude, while the Obligation Factor reflected participants' second-order beliefs about the benefactor's expectation for repayment and the sense of obligation. Interestingly, indebtedness moderately loaded on both factors supporting its mixed relationship to guilt and obligation.”

d. Is “guilt” a component of “communal concern” and of “indebtedness?”

Response: We thank the reviewer for this question. As shown in Study 1, from the perspective of linguistic labeling and definition, guilt is a component of indebtedness in self-reported definitions. As shown in Study 2 and previous literature^{44,45}, from the perspective of social functions, similar to gratitude, guilt motivates reciprocity out of concern for the benefactor, which is a type of communal concern. These findings suggest that human linguistic labeling of emotions may not necessarily correspond to the social functions they serve. This complication highlights the importance of studying and comprehending social emotions from various perspectives, such as emotion classification, linguistic naming, emotional responses, and social functions, among others. As stated in the *Discussion* (p. 30), we believe that continuing efforts to refine mathematical models of emotions across a range of contexts, will eventually allow the field to move beyond relying on the restrictive and imprecise semantics of

linguistic labels to define emotion categories (e.g., guilt, gratitude, indebtedness, obligation, feeling, motivation, etc.).

e. Was the conceptual model developed before the results of Study 1 were known or after? Did Study 1 refine the model(s)? If model 1 cannot be directly used to test the model(s), why was it included here at all? I think that you can easily address this point with a few more sentences that motivate the transition between Study 1 and Study 2.

Response: We thank the reviewer for this suggestion. The conceptual model was developed before the results of Study 1 were known. We examine support for this conceptual model in Study 1, in which participants describe memories of past emotional experiences in a large-scale online questionnaire using regression analysis and topic modeling. The computational models attempt to implement the conceptual model to predict the two different types of decisions (reciprocity & help-acceptance decisions) in the interpersonal task of Study 2. Based on the reviewer's comments we now realize that these two types of models have been inadvertently conflated throughout the paper, which we believe has contributed to some of the confusion. We have now substantially restructured the introduction to explain the aim of each study and clarify the relationship between the conceptual and computational models. Please see paragraphs 3-5 in the *Introduction* section (pp. 4 - 6).

2. Details of the tasks are missing

Response: We thank the reviewer for pointing this out.

First, we have added more detailed descriptions of the task in the *Results* section (pp. 10 - 11) and the caption of Fig. 3, so that the reader can rapidly understand the main idea of our task through concise statements in the main text.

Second, we have added all the details into the *Methods* section (pp. 33 - 39) so that researchers have a deeper understanding of our task and can replicate our study.

3. Model:

a. When I first saw Eq. 1, I was wondering whether this “computational model” is actually not “just” a regression. After all, one could just use three free parameters for weighing π_B , $U_{Communal}$, and $U_{Obligation}$ (plus an intercept). I do see why

the authors use free parameters, which are restricted between 0 and 1, to balance the relative contribution of the three input parameters but I still wonder whether this overcomplicates things unnecessarily (especially because “phi_B is nested within theta_B”). As far as I can see, the authors did individually parameterize phi_B in model 1.5 but they did not test the rather straightforward model suggested above.

Response: We have struggled throughout the review process for this paper to make the models more accessible to a broader audience, but realize we may have overshot this slightly by leading the reviewer to believe the model is simpler than it actually is. As noted above, given the word limitation in the main text and to increase the readability, we have substantially refactored the modeling part in the *Results* section to clarify how the modeling adds to the paper and also to make the high-level details clearer (pp. 15 - 20). We still relegate the majority of the methodological details to the *Methods* section (pp. 46 -53), but we hope that interested readers will now have enough details to understand our methods.

Based on the reviewer’s observation, we have now added a model with three separate and unweighted free parameters (i.e., Model 1.6 for reciprocity and Model 2.5 for help acceptance decisions) as an additional comparison to compare our computational models. Results of model comparison revealed that the previous winning models (Model 1.1 and Model 2.1) still outperformed the alternative models (Tables S7 - S10). This suggests that individual’s reciprocal behaviors are more likely a process of integrating and weighing different psychological components rather than simply a linear combination of all potential factors. These new results of model comparison have been added on pp. 17- 20 and Tables S7 - S10. Corresponding methods have been added on pp. 52 - 53 and to the *SI Methods*

b. The models listed with a-d in the main text (lines 382-387) should be labelled in the same way as the detailed explanations of the models in the Supplementary Materials.

Response: We thank the reviewer for this suggestion. We have now revised these paragraphs on pp. 17 - 20 to ensure the model labels are consistent in the *Methods* (pp. 52 - 53) section and in the *SI Methods*.

c. Model comparisons should be mentioned more explicitly in the main text (not just in the Supplementary Materials).

Response: As suggested, to make the part of model comparison more explicit, we have now presented the main ideas of model comparisons and descriptions of alternative models for reciprocity decisions and help-acceptance decisions as separate

parts in both the *Results* (pp. 17 - 20) and *Methods* (pp. 52 - 53) sections. However, due to the space restrictions, the details and equations of alternative models have been kept in the *SI Methods*.

d. When reading the main text, I was really confused about the relationship between Eq. 1 and Eq. 2/3. None of the parameters in Eq. 2/3 actually appears in Eq. 1. This only becomes clear in the methods but even then one needs to follow up how the different parameters relate to each other. At some point, Eq. 1 needs to be spelled out fully and the number of free parameters needs to be made evident.

Response: We thank the reviewer for pointing this out. Based on this feedback, we have now rewritten the computational modeling part in the *Results* section and included the equations for $U_{Obligation}$ and $U_{Communal}$ to facilitate understanding on our model (pp. 15 - 17). We have also made it clear that there is only one conceptual model, but *two* computational models of different types of behavior. We have named each of these two computational models as the “reciprocity model” and the “help-acceptance model”. However, we have retained the abbreviated Eq. 1 to make it clear how we validate the same conceptual model using the reciprocity decision model, help-acceptance decision model, as well as the neural utility model. We hope this is now clearer in the manuscript.

e. Related to the previous point, I was really surprised to read in the methods that U_{Guilt} and $U_{Obligation}$ are defined differently for the “reciprocity” and the “accept/reject” models.

Response: As stated above we have now attempted to further clarify this by separately referring to the four proposed models in the paper: (a) conceptual model, (b) computational model of reciprocity decisions, (c) computational model of help-acceptance decisions, & (d) neural utility model. We have clarified the differences between the reciprocity model and the help-acceptance model by adding additional high-level explanations in the *Results* section (p. 16):

“The *reciprocity model* (Model 1.1) predicts the participant’s amount of money reciprocated to the benefactor, while the *help-acceptance model* (Model 2.1) predicts binary decisions to accept or reject help. Though the two models are conceptually similar, the values of $U_{Communal}$ and $U_{Obligation}$ are computed slightly differently due to differences in the types of data (i.e., continuous vs. binary decisions) and how appraisals are inferred. It is important to note that in the reciprocity model, we are unable to distinguish between the separate motivations of guilt and gratitude because both positively contribute to reciprocity. In

contrast, based on the findings from Study1, we divide up the parameter space for the help-acceptance model such that $\phi > 0$ indicates a preference for gratitude and motives accepting the help, while $\phi < 0$ indicates a preference for guilt and motives rejecting the help.”

We have also added detailed explanations for the modeling of $U_{Obligation}$ and $U_{Communal}$ for reciprocity decisions and help-acceptance decisions separately in the *Methods* section. Please see our revisions on pp. 46 - 53.

f. The previous point relates to one of main points. The authors write: “To validate our computational model, we tested whether it accurately captured each proposed component process of our conceptual model and successfully predicted participant’s behavior” - Which type of behavior? The ratings? The allocation decisions? The rejection rate? It was initially completely unclear to me on which part of the participants’ data the model was fitted. The authors go on to report correlations but testing for correlations are not how models are usually fitted or evaluated. From reading the methods, I gather that the authors fitted “reciprocity” and “accept/reject” decisions separately. In any case, this should be much clearer in the results section.

Response: We apologize for the previous ambiguous statements. As explained above, we have now refactored the descriptions and validations of computational modeling in the *Results* section, and attempted to clarify that there are two computational models for reciprocity decisions and help-acceptance decisions separately. We have also attempted to be more explicit about which part of the models we validate with each result by adding subheadings and transitional sentences (pp. 17 - 20). We hope this is now clearer.

g. The authors never explain their definition of U_{Guilt} and $U_{Obligation}$ (in Eq. 7 & 8). Maybe I missed it but this is completely intransparent to me. Why are the differences squared? Also, ω_B , which is included in U_{Guilt} depends on E_B ” and at the same time E_B ” appears in $U_{Obligation}$. Why?

Response: As stated above we have now restructured the paper and rewritten the modeling sections in the *Results* (pp. 15 - 20) and *Methods* (pp. 46 -53) sections. We have tried to organize our notation around concepts that are shared across models for reciprocity (Model 1.1) and help-acceptance decisions (Model 2.1) and also that differ across these two models. We note that we have also fully written out and explained the two models. The non-linear formulations for the reciprocity model are a standard way to model social preferences in utility functions^{55,113} by adding convexity to the

utility function via exponential error signals resulting from failing to meet perceived social standards (the benefactor's expectation or the benefactor's perceived care). In terms of model performance, this non-linear model for reciprocity outperformed a model with linear formulations of utilities for self-interest, communal concern, and obligation (Model 1.2; Tables S7 and S9; pp. 48 - 49).

h. Eq. 2 & 3: It also seems confusing to me that ω_B has D_A twice in the numerator. E_B simply is D_A for the “repayment possible” condition. The authors write: “This creates a nonlinear relationship between ω_B and E_B ” such that the relationship is negative when κ is high, positive when κ is low, and uncorrelated in the current dataset with $\kappa = 0.32$ [...].” Doesn't this imply that “high” means above 1 and “low” means between 0 and 1. At $\kappa_B = 1$ the term D_A disappears for the “repayment impossible” condition.

Response: For the detailed explanations of Eq. 2 for second-order-belief E_B (the Eq. 3 now) and Eq. 3 for perceived care ω_B (the Eq. 5 now), please see *Methods* on pp. 46 - 48. In Eq. 3 for perceived care (ω_B) (the Eq. 5 now), the parameter κ ranges from [0, 1] and represents the degree to which the perceived strategic intention E_B reduces the perceived altruistic intention ω_B . It seems that when the benefactor's cost is fixed, E_B and ω_B are anti-correlated, trading off by the parameter κ . Critically however, our experimental design introduces a non-linearity which tampers this relationship: on some trials, repayment was possible, while on others it was not. Therefore, when combining the data of all trials in the experiment, E_B and ω_B are actually uncorrelated (as shown in the figure below, i.e., Fig. S4). This is because in the Repayment impossible condition, E_B is always zero, while ω_B is positively correlated with the benefactor's cost. In the Repayment possible condition, although E_B will reduce ω_B at one specific level of the benefactor's cost, both these two variables increase as the benefactor's cost increases; these two variables are positively correlated in this condition and the extent of correlation is defined by the parameter κ . Therefore, these two variables are uncorrelated across all trials in the experiment. This is consistent with the relationships between self-reported ratings shown in Fig. S3 A-B and Tables S5 and S6. This point was raised by Reviewer 3 in the first round of revision, and we include this short note regarding this point when describing the model in the main text. To avoid the potential misunderstanding on the range of parameter κ , as pointed out by the reviewer, we have revised this note as:

“Here, the parameter κ ranges from [0, 1] and represents the degree to which the perceived strategic intention E_B reduces the perceived altruistic intention ω_B . This creates a nonlinear relationship between ω_B and E_B such that the relationship is negative when κ is close to one, positive when κ is close to zero, and uncorrelated in the current dataset with $\kappa = 0.32 \pm 0.01$, $\beta = -0.03 \pm 0.03$, $t = -1.23$, $p = 0.222$ (Fig. S4).”

Given the word limitation and to increase the readability of the main text, this sentence has now been moved to the *Methods* section (p. 47).

Fig. S4 Relationships between model representations of second-order belief E_B'' and perceived care ω_B in the current study (A) and the simulated data with different levels of κ (B).

We acknowledge that our computational models may oversimplify the appraisal and emotion generating processes. These models operationalize the appraisals of perceived care and second-order belief using information available to each participant in the task (i.e., benefactor’s helping behavior and manipulation about the participants’ ability to reciprocate), which may not generalize to other experimental contexts without modification. Although our computational models performed well in capturing participants' decision behaviors in this task, we emphasize the importance of continued refinement. Corresponding discussions have been added on p. 30 during the first round of revision.

i. In the results, the authors report “that each term of our model was able to accurately capture trial-to-trial variations in self-reported appraisals of second-order belief of the benefactor’s expectation for repayment [...] and perceived care [...].” What does “each term” mean here? Doesn’t this mean that these terms are not separable?

Response: As responded in 3. *Model, Query f.*, we have now refactored the descriptions and validations of computational modeling in the *Results* section (pp. 15 - 20).

j. Okay, after writing all these points above, I realized that these models use D_A and D_B as input to determine the utility of D_B (or better some form of D_A and D_B that are normalized by the money to be allocated). Is this correct? For the

“accept/reject” models D_B is not available because if participants reject, D_B cannot be indicated. If I understand this correctly, then I am even more confused because then the objective function (Eq. 8) includes the difference between D_B and the utility of D_B ... but then the utility of D_B depends on D_B . Sorry, I am a bit lost here.

Response: We apologize for the confusion and appreciate the Reviewer’s efforts in attempting to work through this. We hope that our substantial refactoring of the presentation of the models in the *Results* (pp. 15 - 20) and *Methods* (pp. 46 -53) sections helps make everything clearer in this revision.

As stated by the reviewer, for the reciprocity model, since the utility of D_B depends on the value of D_B , we did not use the maximum likelihood estimation to estimate the model parameters as we did for binary choice of help-acceptance decisions. We estimated the model parameters for Model 1.1 by minimizing the sum of squared errors of the percentages that the model’s behavioral predictions deviated from actual behaviors across all the trials that participants had to passively accept help⁵⁵. Specifically, for each parameterization, we select the maximum $U(D_B)$ across the range of D_B (i.e., $\max(U(D_B))$) to predict each participant’s reciprocity decision for each trial. We calculate the difference between the participant’s actual choice and the model predicted choice, yielding SSE, or the residual sum of squares. We used Matlab’s `fmincon` routine to identify parameters that minimized the sum of SSE of all trials separately for each participant (Eq. 10).

$$SSE = \sum_{t=1}^n \left(\frac{D_B(t) - \max(U(D_B(t)))}{\gamma_B} * 100 \right)^2 \quad \text{Eq. 10}$$

with t indicating trial number. To avoid ending the fitting procedure at a local minimum, the model-fitting algorithm was initialized at 1000 random points in the three-dimensional theta-phi-kappa parameter space for each participant (p. 50).

4. Study 1:

a. *When reading the manuscript I had the following questions in mind: Can the different events for which participants received help be clustered? How severe were these events (e.g., help when moving to a new apartment vs. help during a period of sickness)? How many events entailed pain, i.e., could some events be conceptually related to Study 2? Was the received help truly helpful for resolving the challenges posed by the events? How many did participants report on average ... or were they asked to provide a specific number of events? ... Appendix S1 shows that no questions about the specific events were asked. This should be mentioned in the main text. On*

the other hand, it would be really helpful to know which of the questions from Appendix S1 were used for the analyses in which were not.

Response: We thank the reviewer for this suggestion. In the questionnaire of Study 1, we asked the participants to think carefully about an event in which they received/rejected help from others that impressed them the most in the past one year and happened recently, and answer questions regarding the appraisals, emotions, behaviors, and event-related other factors (e.g., benefactor's cost, the participant's benefit and the social distance between the participant and the benefactor) of this event. At the end of the questionnaire, participants were asked to write a story describing the event in detail in the form of a short story (Q. 28; see *Appendix S1*). While 98.7% participants reported the events of receiving help, 24.4% participants reported the events of rejecting help within the past one year, resulting in 1,991 effective daily events (already reported on pp. 32 - 33).

Regarding “*how severe were these events*”, we did not include a direct question about the severity of event, but we believe this question is closely related to Q. 6 in the questionnaire “6. How helpful was the help? (0 is useless, 100 is very helpful)”. This Q. 6 was used as the indicator for the participant's benefit in our analysis. When examining the relationships between guilt, obligation and indebtedness, to rule out the possibility that these emotion ratings might covary with other related factors in Study 1 (e.g., benefactor's cost, the participant's benefit and the social distance between the participant and the benefactor), we re-estimated the regression models with these factors as control variables, which did not appreciably change the results (Table S1; pp. 40 - 41).

We agree with the reviewer that it is an intriguing question that whether the events can be clustered and whether there are different response patterns in different clusters. However, there was no direct question regarding the classification of the events in the questionnaire (e.g., help when moving to a new apartment vs. help during a period of sickness). It might be possible to conduct text-analysis based on the text descriptions in Q. 28; this point is a valuable future direction but beyond the scope of the current study. Corresponding discussion are added on pp. 30 - 31:

“Future research is needed to extend our conceptual model by differentiating different types of help-receiving events (e.g., help when moving to a new apartment vs. help during a period of sickness) and manipulating other related contexts, such as gift-receiving²³ and help-seeking¹⁷.”

As suggested, we have added corresponding question numbers in the methods of Study 1 (pp. 32 - 33) and labeled the questions included in final analysis in bold in *Appendix S1*.

b. Were regressors normalized? The beta values do suggest this.

Response: We thank the reviewer to point this out. Yes. All variables were normalized before the regression analysis. This information is now added in *Methods* section on p. 40, p. 43, p. 46, and p. 53.

5. Study 2:

a. Please specify in the number of trials with the two repayment conditions. How many benefactors were there? I could only find answers to these two questions in Table S2.

Response: We have added this information in the *Methods* section on p. 37:

“In Study 2a, we manipulated the participant’s beliefs about the benefactor’s intentions by providing additional information about benefactor's intention (condition: Repayment possible vs. Repayment impossible) and benefactor's cost (12 levels of 5, 7, 8, 9, 10, 11, 12, 14, 15, 16, 18, 20). We included one trial for each condition-benefactor’s cost combination for free-choice and force-accept situations, totaling 24 trials in each situation. As a result, there were a total of 48 trials with different anonymous co-players in Study 2a.

Study 2b also included information about benefactor's intention and benefactor's cost. In addition, to disentangle the effect of the benefactor's cost and participant's benefit (i.e., pain reduction), we manipulated the exchange rate between the co-player’s cost and participant’s pain reduction (i.e., efficiency, 0.5, 1.0, and 1.5; the efficiency was always 1.0 in Study 2a). Thus, the participant’s pain reduction was calculated as follows: $\text{Pain reduction} = \text{co-player's cost} / \text{co-player's endowment} \times \text{efficiency} \times \text{maximum pain reduction (16s)}$. In Study 2b, we only included 5 levels for the benefactor’s cost (i.e., 4, 8, 12, 16, 20). Participants were informed that the co-player could only choose discrete amounts of money to spend in this experiment. Furthermore, because the pain duration when benefactor's cost = 20 and efficiency = 1.5 exceeds the maximum pain reduction, this combination was eliminated, leaving 14 benefactor's cost-efficiency combinations. We included one trial for each condition-benefactor’s cost-efficiency combination for free-choice and force-accept situations, totaling 28 trials in each situation. As a result, there were 56 trials with different anonymous co-players in Study 2b. See Table S2 for more details about experimental settings.”

b. Did participants believe that the benefactors were in a neighboring room? It's clear from the methods but could be very briefly mentioned in the results.

Response: The participants were informed that the benefactors made their decisions during a separate visit to ensure that there were enough trials for the study. We have made this clearer in the *Results* (p. 10) and *Methods* (p. 35) sections.

c. The differences between study 2a & 2b could be mentioned more explicitly in 1-2 additional sentences in the main text.

Response: We agree with the reviewer and have updated the text accordingly on p. 11.

d. How was perceived care assessed? Were participants always asked for the other person's expectations AND for perceived care? Or is perceived care "simply" equal to omega_B and thus derived from D_A and E_B"?

Response: We thank the reviewer for these questions.

In response to the reviewer's first half of the question, during the main task, the participant reported trial-by-trial second-order belief of the benefactor's expectation for repayment after the benefactor's decision was revealed and before the participant made decisions of whether to accept help and reciprocity. After the task, all trials were displayed again in a random order and participants recalled how much they believed the benefactor cared for them (i.e., perceived care), as well as their feelings of indebtedness, obligation, guilt, and gratitude in response to the help they received for each trial. These details about the assessments of perceived care and second-order belief have now been added in the *Results* section on pp. 10 - 11, Fig. 3 and corresponding caption, and the *Methods* section on pp. 34 - 39.

Given the limited duration of the task, we only included the ratings second-order belief in the main task, because this is an important indicator of manipulation check that validated our manipulation of Repayment possible and Repayment impossible conditions. Ratings of perceived care and feelings of indebtedness, obligation, guilt, and gratitude were included in the post-task rating session.

We used these self-reported ratings of appraisals and emotions to:

- (1) reveal participants' appraisals and emotional responses to benefactor's help with different intentions (Fig. 4, A - C);

- (2) investigate the relationships between appraisals and emotions to test the hypothesis of conceptual model that feelings of communal concern and obligation are derived from the appraisals of benefactor's altruistic and strategic intentions respectively (Fig. 4, D - E);
- (3) test whether our computational model terms of second-order belief (E_B'') and perceived care (ω_B), developed based on experimental manipulations and hypothetical cognitive processes derived from our conceptual model, can predict self-reported ratings of second-order belief and perceived care (Fig. S5). This analysis validated our computational modeling.

In response to the reviewer's last half of the question, in our computational model, the model term of perceived care (ω_B) was defined as a function of the benefactor's cost D_A and second-order belief E_B'' (see the current Eq. 3 and Eq. 5). We validate this model term by conducting regression analysis between model predictions and self-reported ratings. As responded to 3. *Model, Query f*, to make the computational part clearer, we have now refactored the descriptions and validations of computational modeling in the *Results* section (pp. 15 - 20).

e. What components of the interactive tasks were shown for the post-task rating? The repayment condition, the spent money, the participant's expectation, and their decisions (to accept and to allocate)? Were the four ratings always given in the same order?

Response: We appreciate the reviewer's eye for detail. During Session 2 of the interpersonal game, all of the trials from session 1 were displayed again in a random order. The participant was shown the co-player's information (Information period with a blurred picture, co-player's ID and extra information about benefactor's intention, i.e., the benefactor knew *Repayment possible* or *Repayment impossible*; 4 sec) and their decision (Outcome period with benefactor's cost and the duration of pain reduction for the participant; 5 sec). Then the participant was asked to recall how much they believed the benefactor cared about them, as well as their feelings of indebtedness, obligation, guilt, and gratitude at the time point when they had received the help of the co-player, but had not indicated decisions of accept/reject help and the amount of reciprocity. Ratings were conducted on a scale from 0 to 100, with 0 represented "not at all" and 100 represented "extremely intense". The order of these ratings was counter-balanced across trials. The questions for self-reported ratings on guilt⁴⁰⁻⁴³ and obligation^{13,14,21,46,47} were based on previous research. Notably, the participant's second-order belief of how much the benefactor expected, decisions of whether to accept or reject help, and the amount of reciprocity were not shown in this

session to minimize the influence of their prior behaviors on their reported feelings. These details have now been added in the *Methods* section on pp. 37 - 38.

f. Fig 4B & C: Are these the averages of individual regression coefficients (i.e., data points for all trials per participant) or of one regression (i.e. data points for averages across participants)? After reading the methods. I think that these are derived from mixed effects regressions (lmer in R). Correct?

Response: Visualizations of the regression analyses (Fig. 4, B - C) were created using the `lmer` function of `seaborn` 0.9.0 (<https://seaborn.pydata.org/index.html>) in IPython/Jupyter Notebook (Python 3.6.8)¹⁰⁷ and using the data of all trials for all participants. We note that the confidence intervals were created via bootstrapping that respected the repeated measurements within a participant. We have added more descriptions about this point in the *Methods* section (p. 43).

g. In the section “Behavioral responses to help are influenced by the benefactor's intentions,” did the authors compare the magnitudes of the beta coefficients? It seems to me that they just report the magnitude in the text but not simple t-tests (or contrasts) between the betas for the “repayment possible” and the “repayment impossible” conditions. Fig 5B only shows the rejection rate but not the “reciprocity as a function of the amount of received help.”

Response: To test the effects of experimental manipulations on beneficiary's appraisals (i.e., second-order belief and perceived care), emotions (i.e., gratitude, indebtedness, guilt, and obligation) and behaviors (reciprocity and help-acceptance decisions), in Study 2a we conducted LMM analyses for each dependent variable separately with the benefactor's cost, extra information about benefactor's intention (Repayment possible vs. Repayment impossible) and their interaction effect as fixed effects and the participant ID as a random intercept and slope¹⁰⁶ (p. 43).

As suggested, for decisions of whether to accept help, we re-analyze the data and complement the analysis on the interaction between condition (Repayment possible vs. Repayment impossible) and benefactor's cost (p. 14):

“A logistic regression revealed that when given the chance, participants were more likely to reject help in the Repayment possible condition when they reported more obligation (rejection rate = 0.37 ± 0.10), compared to the Repayment impossible condition (rejection rate = 0.30 ± 0.03), $\beta = 0.27 \pm 0.08$, $z = 3.64$, $p < 0.001$ (Fig. 5B). Moreover, as the benefactor's cost increased, participants were less likely to reject the help ($\beta = -0.65 \pm 0.13$, $z = -5.16$, $p < 0.001$). No significant interaction effect

between condition and benefactor’s cost was observed ($\beta = 0.07 \pm 0.07$, $z = 1.08$, $p = 0.279$).”

We have also revised Fig. 5B to show both the effect of condition (Repayment possible vs. Repayment impossible) and benefactor’s cost on decisions of whether to reject help.

Fig. 5 Computational models of indebtedness. ... (B) Participants' decisions of accept or reject help in each trial plotted as a logistic function of extra information about benefactor's intention and benefactor's cost.

h. Fig 5E: What is the unit on the axis for “predicted reciprocity?” Is it the percentage? If so, the maximum percentage across this parameter space is below 16% but in Fig 5C the values exceed 16%.

Response: The unit for “predicted reciprocity” is yuan (¥), which is the same as Fig. 5A & 5C. As shown in Fig. 5C, the participant’s amount of reciprocity and the resulting predicted reciprocity varied as a function of the benefactor’s cost. To show the general pattern of how predicted reciprocity varied at different parameterizations across different levels of benefactor’s cost, we used the average values of the predicted reciprocity across all levels of benefactor’s cost as the y axis variable. This information has been added to the caption of Fig. 5:

“(E) Model simulations for predicted reciprocity behavior in Repayment impossible and Repayment possible conditions at different parameterizations. The y axis shows the average values of the predicted amount of reciprocity across all levels of benefactor’s cost.”

The unit of reciprocity has now been added in Fig. 5A, C and E.

6. Study 3:

a. *fMRI*: Why three separate GLMs? Shouldn't it be one GLM with (1) reciprocity, (2) communal concern, and (3) obligation as parametric modulators (with or without orthogonalization depending on the question asked)?

Response: For these analyses, we were concerned about the potential impact of multicollinearity. As the reviewer undoubtedly knows, the orthogonalization procedure effectively runs a step-wise regression procedure where the target regressor is first used to explain variance in the predicted variable y . The residual of this model is then used as y for the remaining regressors. This effectively privileges any shared variance between the regressors to the target regressor and is one solution to address multicollinearity. Thus, this procedure actually requires running three separate models. Rather than running this complicated procedure, we simply tested each hypothesis separately, which should provide identical beta estimates as the orthogonalization procedure for each regressor. We appreciate the discussion of this complicated issue in Mumford et al., 2015. This point is clarified in the *Methods* section on p. 54.

One additional complication to note that we test a very specific hypothesis for GLM 3, which imposes a slight change to how we represent second order beliefs E_B'' . Because of the non-normal distribution of E_B'' imposed by our model (i.e., zero in Repayment impossible condition and linear increase in Repayment possible condition, Eq. 3), we constructed a piecewise linear contrast in GLM3, instead of linear parametric analysis (pp. 55 - 56). This entailed creating four separate regressors modeling different parts of the function during the Outcome period: (1) Repayment impossible, (2) Repayment possible and low benefactor's cost (i.e., 4, 6, or 8), (3) Repayment possible and medium benefactor's cost (i.e., 10, 12, or 14), (4) Repayment possible and high benefactor's cost (i.e., 16, 18, or 20). Subsequently, for each participant, we constructed a contrast vector of $c = [-6, 1, 2, 3]$. This piecewise linear contrast ensures that brain responses to the Repayment impossible trials are lower than all of the Repayment possible trials. We have successfully used this approach in

previous work modeling guilt using similar Psychological Game Theoretic utility models⁵⁴. This makes combining tests for this hypothesis more complicated in a single combined GLM with the other hypotheses.

Mumford, J. A., Poline, J. B., & Poldrack, R. A. (2015). Orthogonalization of regressors in fMRI models. *PloS one*, 10(4), e0126255.

b. Which time points were used? This should be clear from reading the results.

Response: This point has been explicitly elaborated in the *Results* section, for example:

“Second, we performed univariate analyses to identify brain processes during the Outcome period (Fig. 3B), where participants learned about the benefactor's decision to help.” (p. 20)

“Using brain activity during the Outcome period of the task (Fig. 3B), we trained two whole-brain models using principal components regression with 5-fold cross-validation⁷²⁻⁷⁴ to predict the appraisals associated with communal concern (ω_B) and obligation (E_B') separately for each participant.” (p. 22)

c. Otherwise, the neural utility model was much clearer to me than the behavioral model.

Response: We apologize for not clearly articulating the details of the computational modeling. We have now rewritten the corresponding parts in *Results* and *Methods* sections to clarify the methods of computational modeling beyond the specific points raised by the reviewer. Please see pp. 15 - 20 in the *Results* section and pp. 46 - 53 in the *Methods* section.

Minor:

1. The first sentence of the abstract is “Receiving help or a favor from another person can sometimes have a hidden cost.” This may be nitty-gritty but I would add “... for the recipient/beneficiary”

Response: As suggested, we have now added this information in the sentence of the Abstract:

“Receiving help or a favor from another person can sometimes have a hidden cost for the beneficiary.”

2. *The abstract could be a bit more specific; in particular, it would be nice to know immediately how many different types of tasks the authors used.*

Response: We thank the reviewer for this suggestion. Given the word restriction, we can only generally summarize the main tasks and techniques used in the current study in the *Abstract* in one sentence, “In this study, we explore these hidden costs by developing and validating a conceptual model of indebtedness across three studies that combine a large-scale online questionnaire, an interpersonal game, computational modeling, and neuroimaging.” As suggested by the reviewer’s next query, we have added more details of the task we used in each study in the *Introduction* (pp. 4 - 6), which will enable the reader to rapidly understand the aim and task of each study.

3. *Related to the previous comment: I appreciate that the introduction is quite general. Nevertheless, it would be nice to get an idea of the used tasks (i.e., the situations in which help can be received such as a reduction of pain as in Studies 2 & 3).*

Response: As suggested, we have added this information in the *Introduction* (p. 6):

“In Study 2, we move beyond self-report and focus specifically on how the guilt and obligation components of indebtedness arise and influence behaviors in the context of an interpersonal game. In this study, participants receive electrical shocks and anonymous benefactors (co-players) can choose to provide aid to the participants by spending money to reduce the duration of their pain experience. The participants, in turn, have the opportunity to accept or reject this help and also to reciprocate the benefactor’s help by sharing some of their own money back. We experimentally manipulate the participants’ beliefs about the benefactors’ intentions by providing information about whether or not the co-players are aware that the participants have the opportunity to repay after receiving help. We test the hypothesis that appraisals of altruistic intentions produce guilt as well as gratitude (i.e., communal concern) while appraisals of strategic intentions lead to obligation. Building on previous models of other-regarding preferences^{37,38,52}, we develop computational models to predict reciprocity and help-acceptance decisions respectively in this interpersonal task by quantifying the tradeoff between the latent motivations of self-interest, communal concern (consisting of guilt & gratitude), and obligation based on appraisals induced by the task (Eq. 1).

In Study 3, we provide further validations of the conceptual model by examining the brain processes associated with the two components of indebtedness by scanning an additional cohort of participants playing the interpersonal game while undergoing functional magnetic resonance imaging (fMRI). Finally, we construct a neural utility model of indebtedness by applying our computational model directly to multivariate brain patterns to demonstrate that neural signals reflect the tradeoff between these feelings and can be used to predict participants' trial-to-trial reciprocity behavior.”

4. It is an open question how much the current results generalize to different situations, in which help can be received. This should be mentioned in 1-2 sentences in the discussion.

Response: As suggested, related discussions are added on pp. 30 - 31:

“Future research is needed to extend our conceptual model by differentiating different types of help-receiving events (e.g., help when moving to a new apartment vs. help during a period of sickness) and manipulating other related contexts, such as gift-receiving²³ and help-seeking¹⁷.”

5. It is unclear what the Tables S3 and S4 show. What is the general verbal conclusion?

Response: Tables S3 and S4 reported the statistics of how benefactor's cost, the extra information about benefactor's intention (Repayment possible vs. Repayment impossible), and their interaction influence appraisals, emotions and behaviors in the combining data of Studies 2a and 2b (Table S3-1), Study 2a (Table S3-2), Study 2b (Table S3-3) and Study 3 (Table S4) respectively. The corresponding verbal conclusions have been already included in the main text. For statistics of the effect of efficiency reported in Table S3-4, which was not the main focus of the current study, we wrote up a summary of the results in the *SI Results*. We have now added references to these points in the main text (p. 11) and paste them below.

“Results of efficiency manipulation

In Study 2b, we further manipulated the participant's benefit from help by varying the exchange rate between the co-player's cost and participant's pain reduction (i.e., **Efficiency**, 0.5, 1, and 1.5) on the basis of Study 2a where Efficiency was 1. The higher the Efficiency, the more benefit the participant would obtain from each amount of the co-player's cost. Results are presented in Table S3-4. Specifically, first, in line with previous studies on gratitude showing that the beneficiary's benefit contributes positively to the feeling of gratitude⁶⁻⁸, we found that the higher the Efficiency (i.e., the more benefit the participant obtained), the higher the

participant's self-reported feeling of gratitude ($\beta = 0.05 \pm 0.02, t = 3.27, p < 0.001$). Similarly, participants' self-reported ratings of indebtedness were positively correlated with the size of Efficiency ($\beta = 0.04 \pm 0.01, t = 2.65, p = 0.008$). In contrary, the Efficiency did not contribute significantly to participants' rating of guilt and the sense of obligation (guilt: $\beta = 0.01 \pm 0.01, t = 0.74, p = 0.458$; obligation: $\beta = 0.00 \pm 0.02, t = -0.03, p = 0.975$) (Table S3-4).

To be noted, since the effects of Efficiency (or the benefit the beneficiary obtained from help) on beneficiary's emotions are not the focus of the current study, the range of Efficiency was relatively small in the current study (i.e., 0.5, 1, 1.5). Therefore, it is possible that the current manipulation was not efficient to capture some relatively extreme situations, which resulted in the current non-significant effects of Efficiency on guilt and the sense of obligation. For example, will a beneficiary feel less guilt when the benefactor's small cost has a larger effect, or feel more guilt when the benefactor's large cost has a small effect? Future specially designed studies are needed to explore how the efficiency of help influences the beneficiary's emotional responses."

6. In the discussion, the relationship to game theory is rather indirect. The task may bear some similarities to the trust game (but even these rather loose connections are not explicitly mentioned).

Response: We apologize for the confusion. Game theory refers to a framework in economics used to model specific interactions between players based on the information available, the possible actions available to each player, and their associated payoffs. Game theorists use this mathematical framework to identify equilibriums using solution concepts. The Trust Game is simply a mathematical object from game theory that is often used as a behavioral assay to probe behavior in experimental economics. Psychological Game Theory is a further refinement to game theory that additionally incorporates player's beliefs into the modeling toolkit. This addition has been instrumental to model more complex cognitive and affective processes into players' utility functions. We and others have successfully used this framework in prior work to model players' intentions, and also emotions such as guilt and anger in games. We further extend these game theoretical models to structural models, in which we identify parameters that allow the models to directly predict participant behavior, which is more akin to how models are used in psychology and neuroscience research. Unfortunately, due to word limitations we are not able to add too much detail explaining this, but we have now tried to add a few sentences in both the *Results* and *Discussion* sections.

Results, p. 15:

“Building on our conceptual model of indebtedness, we developed two computational models using a Psychological Game Theoretic framework ³⁴⁻³⁶ to predict reciprocity and help-acceptance decisions that maximize the beneficiary’s expected utility based on the competing latent motivations of self-interest, communal concern (i.e., guilt and gratitude), and obligation (Eq. 1). ”

Discussion, pp. 26 - 27:

“Previous research has had success modeling belief-dependent utility using Psychological Game Theory ^{36,37} in interactive social contexts. Building on these works, we model participants’ appraisals and emotions ²⁸⁻³³ based on the state of the game to predict two different types of decisions ³⁹. The current work contributes to a growing family of game theoretic models of social emotions such as guilt ^{34,54}, gratitude ⁸⁵, and anger ^{86,87}, and can be used to infer feelings in the absence of self-report, providing new avenues for investigating other social emotions.”

Reviewers' Comments:

Reviewer #2:

Remarks to the Author:

My concerns have been addressed. I have no further comments.

Reviewer #4:

Remarks to the Author:

The authors have put a lot of effort into carefully addressing all my comments. I congratulate them for this wonderful paper.

I have only 4 very minor comments. I do not need to see the responses to these 4 tiny comments.

Fig 5E: I did not quite get why the authors show two plots with different kappa. Please clarify in the caption.

Fig 5: My preference would be to order A-F with A-D in the upper half and E-F in the lower half. But changing this is not necessary.

Line 365: Could you also briefly mention in the beginning of this section that D_A links (indirectly) to U_communal and U_obligation?

Discussion: It would be nice to read a few more - very high-level - sentences on how the two reciprocity and the help-acceptance models can be combined

The earlier comment by Reviewer #3 is indeed a very insightful comment. As far as I understand, the authors have addressed this comment.

Response to reviews

Reviewer #2 (Remarks to the Author):

My concerns have been addressed. I have no further comments.

Response: We thank the reviewer for their comments and suggestions on this article during the review process.

Reviewer #4 (Remarks to the Author):

The authors have put a lot of effort into carefully addressing all my comments. I congratulate them for this wonderful paper. I have only 4 very minor comments. I do not need to see the responses to these 4 tiny comments.

Response: We thank the reviewer for their constructive suggestions during the review process, which we believed have helped us to greatly improve the clarity of the manuscript.

Fig 5E: I did not quite get why the authors show two plots with different kappa. Please clarify in the caption.

Response: We show two plots with different κ to illustrate the how the model predicted reciprocity changes as a function of the tradeoff between communal and obligation feelings based on ϕ and interacts with the intention inference parameter κ . As suggested, we have clarified this point in the Fig. 5E caption.

“e Model simulations for predicted reciprocity behavior in Repayment impossible and Repayment possible conditions at different parameterizations. The y-axis shows the average values of the predicted amount of reciprocity across all levels of benefactor’s cost. The model predicted reciprocity changes as a function of the tradeoff between communal and obligation feelings based on ϕ and interacts with the intention inference parameter κ . Increased emphasis on obligation corresponds to increased reciprocity to favors in the Repayment possible condition, but decreased reciprocity in the Repayment impossible condition; this effect is amplified as κ increases.”

Fig 5: My preference would be to order A-F with A-D in the upper half and E-F in the lower half. But changing this is not necessary.

Response: We appreciate the reviewer's suggestion. We attempted to rearrange the sub-figures as indicated. This would, however, increase the gap between the sub-figures for actual behavior data and the corresponding sub-figures for model predictions, making it more difficult for readers to compare these two lines of results. As stated by the reviewer, changing this is not necessary. Therefore, we decide to keep Fig. 5 as it is.

Line 365: Could you also briefly mention in the beginning of this section that D_A links (indirectly) to $U_{communal}$ and $U_{obligation}$?

Response: We thank the reviewer for this suggestion. We have revised the corresponding parts to briefly mention that $U_{Communal}$ and $U_{Obligation}$ link indirectly to D_A (p. 12):

“The central idea is that upon receiving a favor D_A from a benefactor A , the beneficiary B chooses an action D_B that maximizes his/her overall utility U . This utility is comprised of a mixture of values arising from self-interest π weighted by a greed parameter θ , and feelings of communal concern $U_{Communal}$ and obligation $U_{Obligation}$, which are inferred from the appraisals of D_A and weighted by the parameter ϕ . Larger ϕ values reflect the beneficiary's higher sensitivity to feelings of communal concern relative to obligation. $U_{Communal}$ reflects a linear combination of guilt and gratitude components (see Methods).”

Discussion: It would be nice to read a few more - very high-level - sentences on how the two reciprocity and the help-acceptance models can be combined

Response: We thank the reviewer for this suggestion, but have ultimately decided that this particular nuance of our modeling choices was a little too detailed for our high-level discussion of the models. The two models predicting reciprocity and help-acceptance decisions are built on the same conceptual model and both use very similar representations of the appraisal and emotion generating processes. We could have combined the models and simultaneously predicted both behaviors by minimizing a joint loss function, but this would have made it complicated to systematically evaluate each component of the models and compare them to control models, which seemed more important for validating the models.

A high-level conceptual point about extending these models to other behaviors was added at the suggestion of the reviewer in the previous revision on p. 26:

“Third, future research is needed to extend our conceptual model by differentiating different types of help-receiving events (e.g., help when moving to a new apartment vs. help during a period of sickness) and manipulating other related contexts, such as gift-receiving²³ and help-seeking¹⁷.”

The earlier comment by Reviewer #3 is indeed a very insightful comment. As far as I understand, the authors have addressed this comment.

Response: We agree with the reviewer, and have already addressed this comment during last round of revision.